# Pregnancy-induced maternal microchimerism shapes neurodevelopment and behavior in mice

Steven Schepanski [1,2], Mattia Chini [2], Veronika Sternemann[1,2], Christopher Urbschat [1], Kristin Thiele[1], Ting Sun[3,4], Yu Zhao [3], Mareike Poburski[1,2], Anna Woestemeier[5], Marie-Theres Thieme [1], Dimitra E. Zazara[1], Malik Alawi [6], Nicole Fischer[7], Joerg Heeren [8], Nikita Vladimirov [9], Andrew Woehler [9], Victor G. Puelles[10], Stefan Bonn [3], Nicola Gagliani [5], Ileana L. Hanganu-Opatz[2,11] ✉ & Petra C. Arck [1,11] ✉

Life-long brain function and mental health are critically determined by developmental processes occurring before birth. During mammalian pregnancy, maternal cells are transferred to the fetus. They are referred to as maternal microchimeric cells (MMc). Among other organs, MMc seed into the fetal brain, where their function is unknown. Here, we show that, in the offspring's developing brain in mice, MMc express a unique signature of sensome markers, control microglia homeostasis and prevent excessive presynaptic elimination. Further, MMc facilitate the oscillatory entrainment of developing prefrontal-hippocampal circuits and support the maturation of behavioral abilities. Our findings highlight that MMc are not a mere placental leak out, but rather a functional mechanism that shapes optimal conditions for healthy brain function later in life.

Fetal development is a critical period in life and highly dependent on the supply of maternal resources[1–7]. Besides nutrients, hormones and other factors, maternal cells are vertically transferred to the fetus during mammalian pregnancy[8–12]. As they occur in low numbers in the offspring, these cells have been termed maternal microchimeric cells (MMc)[6,8,10,13]. The transfer of MMc from mother to fetus commences with maturing placentation, hence, with the onset of the second trimester in humans and around mid-gestation in mice[8]. Remarkably, MMc are not rejected by the fetal immune system[14]. In fact, the genetically discordant MMc can even show a long-term persistence in offspring's organs until adulthood[14–16]. During fetal development, MMc seed into a number of fetal organs, including primary and secondary immune organs as well as non-immune organs[17,18]. MMc have also been detected in the offspring's brain[9,19], yet their phenotype, location and impact on brain-resident immune and non-immune cells in the fetus and brain function is still unknown.

Higher cognitive abilities, the most complex brain functions, rely on limbic circuits centered on the prefrontal cortex (PFC) and the

[1]Division of Experimental Feto-Maternal Medicine, Department of Obstetrics and Fetal Medicine, University Medical Center Hamburg-Eppendorf, Hamburg, Germany. [2]Institute of Developmental Neurophysiology, Center for Molecular Neurobiology Hamburg (ZMNH), University Medical Center Hamburg-Eppendorf, Hamburg, Germany. [3]Institute of Medical Systems Biology, Center for Molecular Neurobiology Hamburg (ZMNH), University Medical Center Hamburg-Eppendorf, Hamburg, Germany. [4]Department of Neurogenetics, Max Planck Institute for Multidisciplinary Sciences, Göttingen, Germany. [5]Department of General, Visceral and Thoracic Surgery, University Medical Center Hamburg-Eppendorf, Hamburg, Germany. [6]Bioinformatics Service Facility, University Medical Center Hamburg-Eppendorf, Hamburg, Germany. [7]Institute of Medical Microbiology, Virology and Hygiene, University Medical Center Hamburg-Eppendorf, Hamburg, Germany. [8]Department of Biochemistry and Molecular Cell Biology, University Medical Center Hamburg-Eppendorf, Hamburg, Germany. [9]Berlin Institute for Medical Systems Biology, Max Delbrück Center for Molecular Medicine, Berlin, Germany. [10]III Department of Medicine, University Medical Center Hamburg-Eppendorf, Hamburg, Germany. [11]These authors contributed equally: Ileana L. Hanganu-Opatz, Petra C. Arck. ✉e-mail: hangop@zmnh.uni-hamburg.de; p.arck@uke.de

hippocampus (HP). During development, the HP drives the formation of neuronal ensembles in the PFC[20,21]. The maturation of brain function is not only controlled by patterns of coordinated electrical activity emerging within these neuronal ensembles[22], but also regulated by microglia. Microglia refine the neuronal connectivity via synapse elimination[23,24], if equipped with relevant transcriptional patterns[25,26], such as the so-called microglial sensome[27,28]. Whether MMc interact with limbic circuits or offspring's microglia during development is still unknown.

In this work, we utilized preclinical models to investigate molecular and cellular characteristics of MMc in combination with in vivo electrophysiology and behavioral experiments to decipher the functional role of MMc in the developing brain. We show that MMc promote microglia homeostasis, early brain wiring and behavioral development of mice.

## Results

### Detection and location of MMc in the offspring's brain

We first phenotyped MMc in offspring's brain by flow cytometry, using two complementary experimental approaches. MMc were identified at the end of fetal development, at embryonic day (E) 18.5, as well as at postnatal day (P) 8 and adulthood (P60). MMc detection was either based on the maternal expression of CD45.2 and H-2D[b/b], or—exclusively assessed on E18.5—by tdTomato[+]-fluorescence (Fig. 1a–d). Both approaches yielded to similar numbers of MMc in the fetal brain on E18.5 (Fig. 1b–d). MMc in brain declined with increasing offspring's age. Within the MMc leukocyte subset in the offspring's brain at E18.5, P8, and P60, MMc could be identified to comprise largely of microglia, T and B cells (Fig. 1b, Supplementary

Fig. 1a–g). Sex-specific differences of MMc total numbers or within distinct phenotypic subsets were not detectable. Utilizing a whole organ clearing pipeline[29], we located tdTomato[+] MMc mainly in the PFC and HP at E18.5, but also in other limbic areas, the cerebellum, and—albeit sparsely—in other brain regions (Supplementary Fig. 1h, Supplementary Movie 1).

### ScRNA-sequencing-based identification of cellular diversity among MMc in fetal brain

For further MMc characterization, we isolated brain-resident MMc from fetal cells based on their unique surface expression of CD45.2 and H-2D[b/b] and performed scRNA-sequencing (scRNA-seq) (Supplementary Fig. 2a, Supplementary Table 1). Subsequent analysis of the scRNA-seq data of the brain-resident MMc via dimensionality reduction and unsupervised clustering revealed nine transcriptionally distinct clusters among the MMc (Fig. 1e). Five of the nine MMc clusters were characterized as classical microglia due to their consistently high expression of C-X3-C motif chemokine receptor 1 (Cx3cr1), purinergic receptor P2ry12 (P2ry12), secreted protein acidic and cysteine rich (Sparc), and transmembrane protein 119 (Tmem119) (Fig. 1f). The remaining four MMc clusters formed spatially distinct subsets and were identified as B, T, endothelial, and neuron-like cells (Fig. 1e, f). We excluded that the MMc microglia were border-associated or perivascular macrophages due to their low expression of platelet factor 4 (Pf4) and lymphatic vessel endothelial hyaluronan receptor 1 (Lyve1) (Supplementary Fig. 2b, Supplementary Table 2). Taken together, our findings underpin that MMc in the fetal brain are mostly leukocytes, but also include non-immune subsets, e.g., endothelial and neuron-like cells.

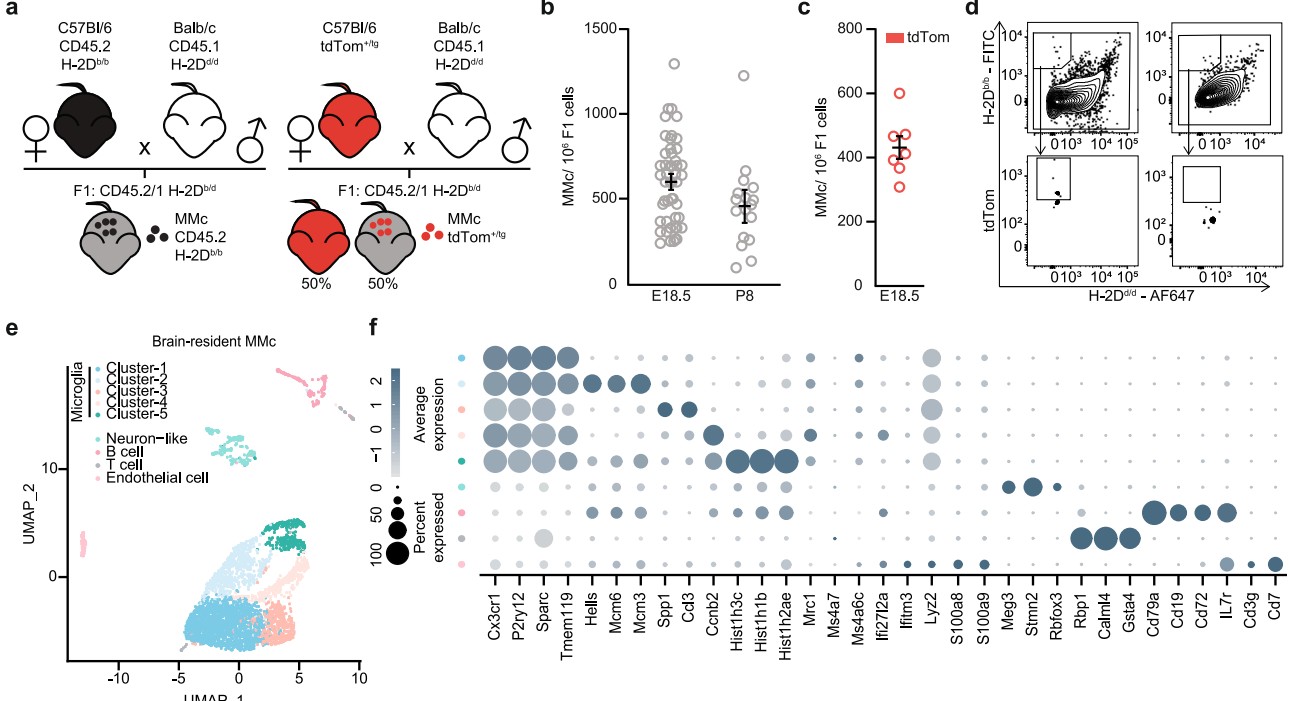

**Fig. 1 | Cellular and molecular characterization of MMc in the offspring's brain.** **a** Experimental set-up of the mating strategies developed to identify MMc in brain based on surface expression of CD45.2 and H-2D[b/b] in CD45.2/1 and H-2D[b/d] offspring (left) and based on tdTomato-fluorescent reporter-expression (right). **b** Number of CD45.2 and H-2D[b/b] MMc among fetal brain cells on E18.5 (n = 43) and P8 (n = 17), using the gating strategy shown in Supplementary Fig. 1a. **c** Number of tdTomato-expressing MMc among fetal brain cells on E18.5 (n = 7). **d** Representative contour plots used to identify tdTomato[+] MMc in fetal brain by flow cytometry (left) and tdTomato negative control (right). **e** UMAP

dimensionality reduction plot of scRNA-seq data set from 4305 MMc isolated from fetal brains (n = 21, pooled from 4 litters). The phenotypes of the nine transcriptionally distinct MMc clusters are marked by the color keys. **f** Dot plots revealing the expression of cluster-specific genes in the nine MMc clusters identified in (**e**), using the same color keys. The dot size indicates the percentage of expression, the shade represents the average expression within the respective cluster. Data are displayed as mean ± SEM in (**b**) and (**c**), each circle corresponds to one pup.

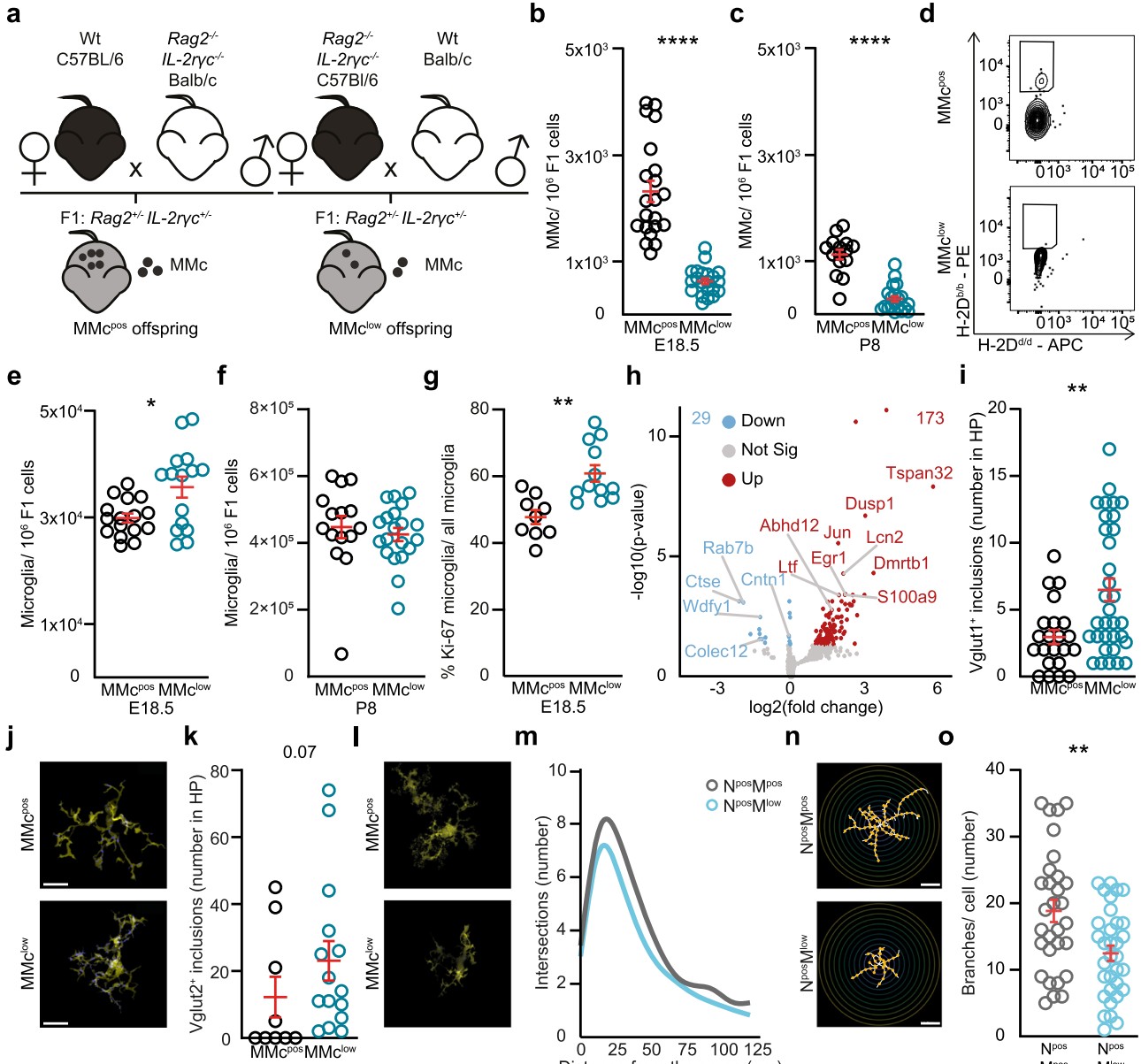

**Fig. 2 | Reduction of MMc in the offspring's brain induces excessive microglia-mediated neuronal refinement in neonatal mice. a** Reciprocal mating strategies used to generate MMc$^{low}$ and MMc$^{pos}$ offspring. **b** Number of MMc among brain cells from MMc$^{pos}$ ($n = 21$) and MMc$^{low}$ ($n = 21$) offspring isolated at E18.5 and identified by flow cytometry. **c** Same as (**b**) but for P8 (MMc$^{pos}$ $n = 15$, MMc$^{low}$ $n = 20$). **d** Representative contour plots used to identify MMc in fetal brain. **e** Number of fetal microglial cells, identified by flow cytometry in MMc$^{pos}$ ($n = 16$) and MMc$^{low}$ ($n = 15$) brains at E18.5 ($p = 0.0366$). **f** same as (**e**) but for P8 (MMc$^{pos}$ $n = 15$, MMc$^{low}$ $n = 20$). **g** Frequency of Ki-67 co-expression on fetal microglia, identified by flow cytometry at E18.5 (MMc$^{pos}$ $n = 9$, MMc$^{low}$ $n = 12$; $p = 0.0007$). **h** Volcano plot depicting comparative gene expression changes within fetal microglial cells isolated from MMc$^{pos}$ and MMc$^{low}$ mice at E18.5. Upregulated genes ($n = 173$) are marked in red, down-regulated genes ($n = 29$) in blue. Gene names are given when concordant expression was observed in two independent experiments; using transcriptome data from 2 to 3 biological replicates isolated each from 3 to 4 pooled litters. **i** Number of Vglut1$^+$ presynaptic vesicles engulfed by microglial cells in the CA1 area of the hippocampus (HP) in MMc$^{pos}$ ($n = 22$) and MMc$^{low}$ ($n = 33$) offspring at P8 ($p = 0.0048$). **j** Representative photomicrograph of data shown in (**i**), microglial cells appear in yellow, engulfed Vglut1$^+$ presynaptic vesicles as blue dots. Scale bar, 200 µm. **k, l** Same as (**i**), (**j**) but for Vglut2$^+$ presynaptic vesicles (MMc$^{pos}$ $n = 9$, MMc$^{low}$ $n = 15$; $p = 0.0693$). **m** Number of dendritic intersections of co-cultured neurons and microglia isolated from MMc$^{pos}$ embryos (N$^{pos}$M$^{pos}$, gray) and neurons from MMc$^{pos}$ embryos co-cultured with microglia from MMc$^{low}$ embryos (N$^{pos}$M$^{low}$, light blue). Data were generated in three independent experiments. **n** Representative photomicrographs of data shown in (**m**). Scale bar, 50 µm. **o** Number of branches per cell in co-culture conditions (N$^{pos}$M$^{pos}$ $n = 30$; N$^{pos}$M$^{low}$ $n = 33$, $p = 0.0065$). Significance was determined by a two-sided Mann−Whitney U test, *$p < 0.05$, **$p < 0.01$, ****$p < 0.0001$. Data are displayed as mean ± SEM and each circle denotes one pup (**b, c, e–g, i, k**) or neuron (**n**).

## Mouse model of experimental reduction of MMc in fetal brain

Given the distinct phenotypic and transcriptional signature of MMc in fetal brain, we used a mouse model in which MMc were experimentally reduced in order to gain insights into the functional role of MMc in the offspring's brain. This reduction of MMc was achieved by reciprocal mating of *Rag2$^{-/-}$IL-2rγc$^{-/-}$* female or male mice respectively with wild-type (wt) mice. *Rag2$^{-/-}$IL-2rγc$^{-/-}$* mice are immunodeficient and lack T, B and—to a lesser extent—innate lymphoid cells[30]. The offspring of these reciprocal mating combinations all express a *Rag2$^{+/-}$IL-2rγc$^{+/-}$* genotype (Fig. 2a). Noteworthy, since the γc gene is encoded by the X-chromosome, male offspring born to *Rag2$^{-/-}$IL-2rγc$^{-/-}$* females are γc deficient, while male offspring born to wt females carry one copy of the γc gene.

To control for this hemizygosity, only female offspring were included in the respective experiments.

We first excluded that *Rag2⁻/⁻IL-2rγc⁻/⁻* and wt pregnant females differ in parameters that may confound fetal development and hence, subsequent brain function. We analyzed systemic immune markers indicative for maternal inflammation or immune activation (MIA). Comparative analyses between *Rag2⁻/⁻IL-2rγc⁻/⁻* and wt pregnant females revealed no significant differences in plasma protein or RNA levels of peptide hormones, cytokines and chemokines (Supplementary Fig. 3a–d). We also assessed the number of peripheral immune cell subsets in *Rag2⁻/⁻IL-2rγc⁻/⁻* and wt pregnant females and observed no differences in residual subsets, such as myeloid cells, between groups of dams, whilst the expected lack of T and B cells in *Rag2⁻/⁻IL-2rγc⁻/⁻* females could be confirmed (Supplementary Fig. 3e–g). Since maternal infections, for which especially *Rag2⁻/⁻IL-2rγc⁻/⁻* mice are susceptible, could lead to MIA, we also ruled out differences in various pathogen encounter between wt and *Rag2⁻/⁻IL-2rγc⁻/⁻* females over the course of our experiments (Supplementary Table 3). We also excluded that *Rag2⁻/⁻IL-2rγc⁻/⁻* and wt dams deviate in gestational parameter, such as maternal weight gain, or in postnatal caring behavior (Supplementary Fig. 4a–e). Similarly, offspring born to *Rag2⁻/⁻IL-2rγc⁻/⁻* or wt dams respectively showed no significant differences in early postnatal developmental trajectories, such as early life weight gain, surface righting, vibrissae reaction, cliff avoidance, or grasping reflex (Supplementary Fig. 4f–j). Prenatal parameter that may affect fetal development and intestinal microbiome between offspring born to *Rag2⁻/⁻IL-2rγc⁻/⁻* or wt dams have previously been ruled out[17]. Moreover, we used 16S rRNA sequencing and Gram staining in offspring at P8 to exclude the presence and difference of the microbiome in the developing brain, which may have resulted from blood-brain barrier disruptions (Supplementary Fig. 5a–d).

Upon exclusion of these confounding factors, we analyzed MMc numbers in offspring's brain at E18.5 and P8. Offspring born to *Rag2⁻/⁻IL-2rγc⁻/⁻* dams harbored significantly fewer MMc in the brain and were termed 'MMc^low', compared to offspring from wt dams. These offspring were termed 'MMc^pos' (Fig. 2a–d). MMc immune phenotyping in MMc^low and MMc^pos offspring was performed upon magnetic-activated cell sorting (MACS)-based MMc enrichment to improve signal-to-noise ratio during flow cytometric analyses and revealed that not only T and B cells among MMc were significantly lower in MMc^low offspring, but also microglia (Supplementary Fig. 6a, b).

## MMc maintain fetal microglia homeostasis and suppress excessive presynaptic elimination

Next, we analyzed fetal microglia as the key immune cell population in the brain and detected significantly higher numbers of microglia in MMc^low compared to MMc^pos offspring at E18.5 (Fig. 2e). At P8, the number of fetal microglia was overall higher and significant differences between MMc^low and MMc^pos offspring could no longer be detected (Fig. 2f). In depth assessment of microglia at E18.5 revealed that the proportion of microglia co-expressing the proliferation marker Ki-67 was significantly higher in MMc^low offspring (Fig. 2g), suggesting an overshooting microglia proliferation when MMc are low. Conversely, both, fetal T and B cells, were significantly lower in number among brain cells in MMc^low offspring at E18.5. With increasing offspring's age (P8), the number of T and B cells was higher compared to E18.5 and differences between groups were no longer present (Supplementary Fig. 6c–f). Transcriptome analysis of fetal microglia revealed that 202 genes were differentially regulated between MMc^pos and MMc^low offspring (Supplementary Table 4). Here, genes suppressing inflammation, such as ras-related protein (*Rab-7b*), responsible for suppressing tumor necrosis factor (*Tnf*), interleukin-6 (*Il-6*), and interferon β (*Inf-β*) production in macrophages[31], were down-regulated in microglia from MMc^low offspring (Fig. 2h). Along this line, an increased expression of *Tnf* and—to a lesser extent—methyl CpG binding protein 2 (*Mecp2*) and

*Ifn-β* in fetal MMc^low microglia were confirmed by qPCR (Supplementary Fig. 6g). These findings are highly indicative for a hyperactivation and altered function of fetal microglia in MMc^low offspring. This may result in an enhanced disruption of homeostasis in the brain and dismantling of synaptic connections by phagocytosis of presynaptic vesicles[23,24,32,33].

In order to provide evidence for this notion, we investigated microglia engulfment of synaptic terminals in the prelimbic subdivision (PL) of PFC and HP, since these areas are the core of neuronal networks accounting for complex cognitive abilities, such as memory, learning, and flexibility[34]. The engulfment of terminals from short-range projections stained by Vglut1⁺ and from long-range projections stained by Vglut2⁺ augmented in MMc^low offspring ex vivo, yet the increase reached significance level only for Vglut1⁺ (Fig. 2i–l, Supplementary Fig. 7a–n). More specifically, we observed fewer Vglut1⁺ and Vglut2⁺ synaptic puncta in MMc^low PL and HP, compared to MMc^pos offspring, indicating fewer Vglut1⁺ and Vglut2⁺ synapses and therefore fewer short- and long-range projections (Supplementary Fig. 7b, e, i, l). Co-culture of neurons with microglia from MMc^pos and MMc^low offspring supported these findings, as significantly reduced neuronal branching and growth was detected when microglia were derived from MMc^low offspring (Fig. 2m–o, Supplementary Fig. 8a–g). These results indicate that MMc promote fetal microglia homeostasis in the developing brain via suppression of hyperactivation and inflammation of fetal microglia.

## MMc boost functional communication within the developing prefrontal-hippocampal circuits

To test if the excessive microglia engulfment of presynaptic vesicles in MMc^low offspring leads to abnormal neuronal and network activity in developing PL and intermediate/ventral HP, we performed multisite extracellular recordings of local field potential and spiking activity in P8 mice in vivo (Fig. 3a). At this age, discontinuous patterns of activity that consist of oscillatory discharges, alternating with "silent" periods lacking coordinated activity[20,21] were recorded in PL (Fig. 3b top) and HP (Fig. 3b bottom) of both MMc^pos and MMc^low offspring. Oscillatory broadband power and amplitude in PL and HP were both significantly lower in MMc^low than MMc^pos (Fig. 3c–f, Supplementary Fig. 9a–f). While only some additional oscillatory properties, such as duration and number of active oscillatory periods, were significantly affected by the MMc reduction (Supplementary Fig. 9g–l), the entropy of the spiking patterns showed a slight decrease in the HP of MMc^low mice (Supplementary Fig. 9m, n). Although levels of significance were not reached, this decrease suggests a possible disturbance of structured spiking networks (Supplementary Fig. 9m, n).

Since oscillatory activity in HP drives the activation of PL circuits during neonatal development[20,21], we next tested whether MMc influence the interactions within prefrontal-hippocampal circuits. While the tight prefrontal-hippocampal synchrony in theta band was similar in MMc^pos and MMc^low offspring (Supplementary Fig. 9o–q), the phase-amplitude coupling within 20–30 Hz was significantly weaker in MMc^low offspring (Fig. 3g–i). Accordingly, also the spike-spike communication between PL and HP, as measured by the mutual information between the spike trains of the two brain areas, was significantly reduced in absence of MMc (Fig. 3j). These data show that MMc impact network activity and functional communication within neonatal prefrontal-hippocampal circuits.

## MMc contribute to the development of behavioral abilities in neonatal and pre-juvenile offspring

Next, we tested whether the dysfunction of prefrontal-hippocampal circuits in neonatal MMc^low mice leads to behavioral deficits already at the early developmental stage. First, we investigated isolation-induced ultrasonic vocalizations as a proxy for emotional dysregulation in neonatal offspring[35]. We observed significant differences with regard

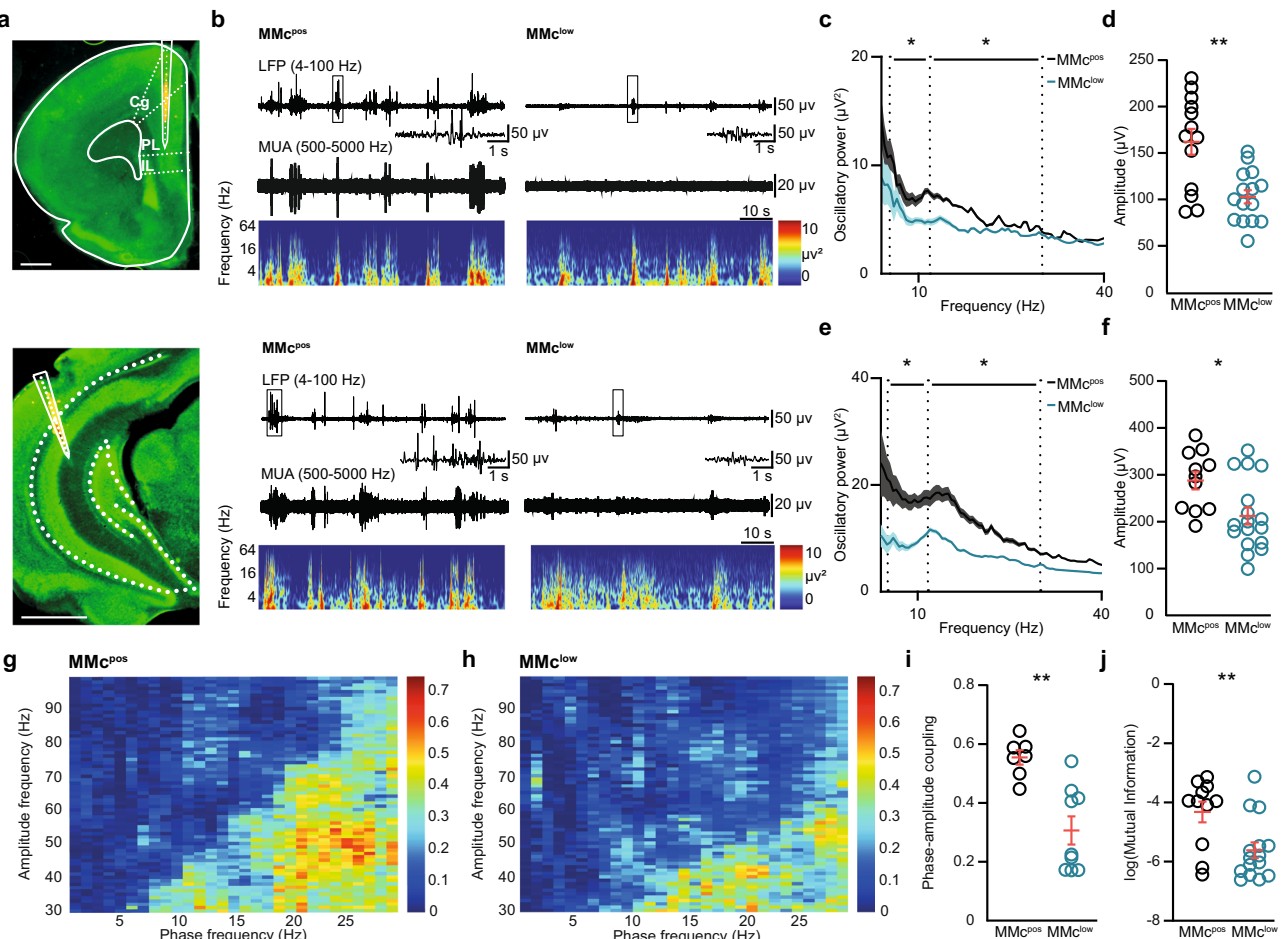

**Fig. 3 | Reduction of MMc in the offspring's brain impairs prefrontal-hippocampal circuits in neonatal mice. a** Digital photomontage reconstructing the position of a one-shank DiI-labeled electrode in a Nissl-stained 100 μm-thick coronal section containing the prefrontal cortex (top) sub-divided into cingulate cortex area 1 (Cg), prelimbic cortex (PL), and infralimbic cortex (IL) from a P8 mouse. Scale bar, 500 μm. The bottom digital photomontage shows the same but for the CA1 of the hippocampus (HP). Scale bar, 1000 μm. Photomicrographs are representative of three independent experiments. **b** (top) Extracellular recordings of the local field potential (LFP) in the PL of a P8 MMc$^{pos}$ (left) and MMc$^{low}$ (right) pup after band pass filtering (4–100 Hz) together with the corresponding MUA (500–5000 Hz). (below) Color-wavelet spectra of LFP displayed at identical time scale. (bottom). Same as (top) for HP. **c** Spectra of absolute power of discontinuous oscillations averaged for all investigated MMc$^{pos}$ (θ $n = 11$, β $n = 10$, γ $n = 11$) and

MMc$^{low}$ (θ $n = 12$, β $n = 12$, γ $n = 11$) mice. Dotted lines mark the borders of frequency bands theta (θ) (4–12 Hz), beta (β) (12–30 Hz), and gamma (γ) (30–40 Hz). **d** Amplitude of oscillations detected in the PL of P8 mice (MMc$^{pos}$ $n = 13$, MMc$^{low}$ $n = 15$; $p = 0.0022$). **e**, **f** Same as (**c**), (**d**) for the HP (θ MMc$^{pos}$ $n = 9$, MMc$^{low}$ $n = 13$; β MMc$^{pos}$ $n = 10$, MMc$^{low}$ $n = 13$; γ MMc$^{pos}$ $n = 10$, MMc$^{low}$ $n = 13$; Amplitude MMc$^{pos}$ $n = 11$, MMc$^{low}$ $n = 17$; $p = 0.0109$). **g** Heatmap of coupling between phase and amplitude in MMc$^{pos}$ offspring at P8. **h** same as (**g**) for MMc$^{low}$. **i** Average phase-amplitude coupling between 20–30 Hz (MMc$^{pos}$ $n = 7$, MMc$^{low}$ $n = 9$, $p = 0.0007$), as shown in (**g**) and (**h**). **j** Log of mutual information between HP and PL (MMc$^{pos}$ $n = 11$, MMc$^{low}$ $n = 15$, $p = 0.0077$). Significance was determined by a two-sided Mann–Whitney U test, *$p < 0.05$, **$p < 0.01$. Data in (**c**) and (**e**) are displayed as median ± 1.5 IQR. Data in (**d**), (**f**), (**i**), (**j**) are displayed as mean ± SEM and each circle denotes one pup.

to the quality of the vocalizations between groups, as the length of simple calls was lower in the MMc$^{low}$ offspring, whereas the frequency jumps and the complex calls lasted longer (Fig. 4a, b, Supplementary Fig. 10a–c). These behavioral features might indicate emotional distress and disruption of social communication between mother and MMc$^{low}$ pups. Intriguingly, a similar cry pattern has been observed in autistic children[36].

Second, we monitored the emergence of cognitive abilities requiring prefrontal-hippocampal communication that can be tested starting from the second to third postnatal week. Recognition memory is one of the earliest cognitive abilities that develops during pre-juvenile age and can be easily tested, since it relies on the mouse's intrinsic exploratory drive to investigate novel stimuli. MMc$^{low}$ offspring showed significantly shorter interaction time and fewer interactions with novel or less-recent objects when compared to MMc$^{pos}$ (Fig. 4c–f), which indicates greater discrimination compared to MMc$^{pos}$ animals.

General anxiety, monitored by the general running distance and the time spent in the center vs. border of the arena were excluded as confounder of the behavioral test (Supplementary Fig. 10d, e). Assessment of other abilities, such as spatial recognition, working memory, and repetitive-like behavior, revealed no significant differences between MMc$^{pos}$ and MMc$^{low}$ offspring (Supplementary Fig. 10f–h).

**Restoration of MMc rescues brain wiring and cognitive abilities**
Next, we tested if immune cells adoptively transferred into *Rag2$^{-/-}$IL-2ryc$^{-/-}$* pregnant females can restore MMc and reach fetal brains. First, we injected tdTomato$^+$ cells and successfully traced them in the fetal brain (Supplementary Fig. 11a, b). In order to provide proof of concept that the observations made in MMc$^{low}$ offspring with regard to alterations in fetal microglia number and function as well as the behavioral disturbances are causally linked to lower number of MMc, we then utilized an approach of adoptive transfer (AT) and injected $1 \times 10^7$

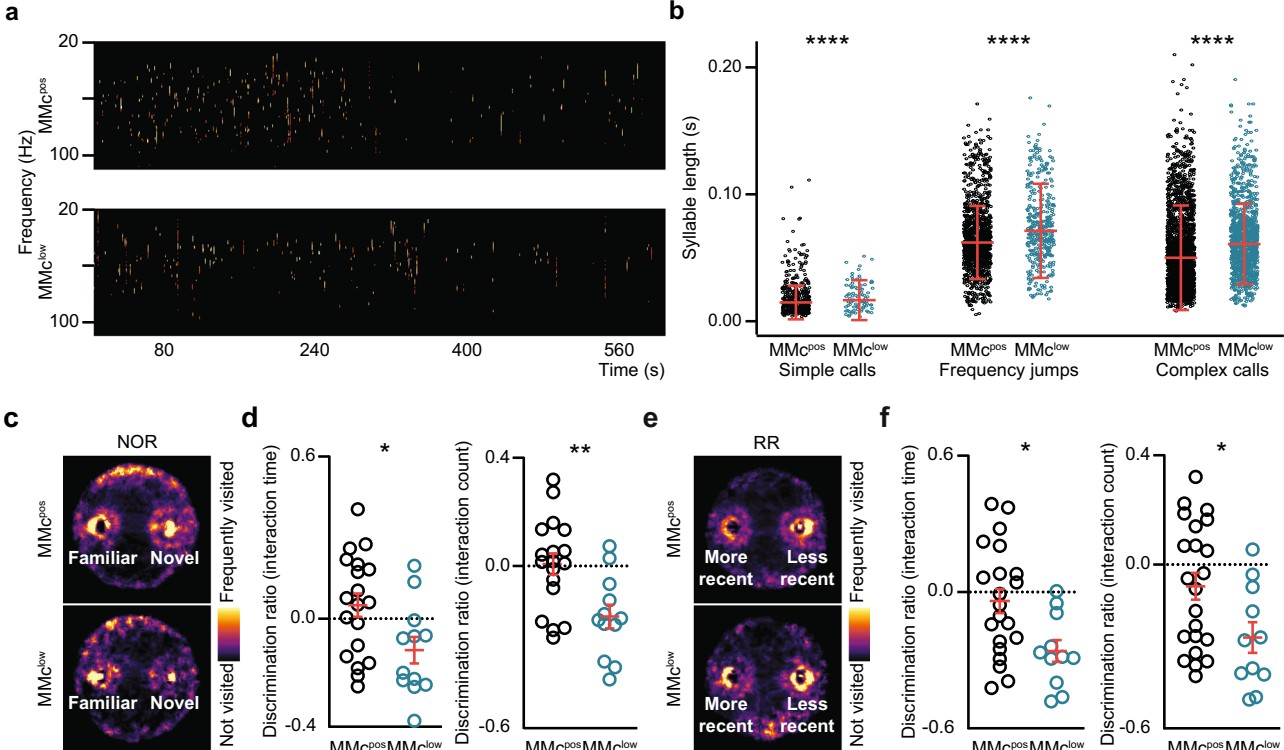

**Fig. 4 | MMc modulate behavioral performance of neonatal and juvenile MMc^low and MMc^pos offspring. a** Spectrogram of maternal-separation induced ultrasonic vocalizations of MMc^pos and MMc^low pups. **b** Scatter plot displaying syllable length of the three types of calls emitted by MMc^pos ($n = 11$) and MMc^low ($n = 8$) pups at P8. **c** Heatmap tracking MMc^pos and MMc^low offspring in a novel object recognition task (Arena size: D: 34 cm; H: 30 cm). Areas of frequent visits appear in bright yellow. **d** Scatter plots of discrimination ratio for the time of interaction with the objects (left, $p = 0.0183$) and the number of interactions (right, $p = 0.0031$) of MMc^pos (left $n = 19$, right $n = 18$) and MMc^low (left $n = 12$, right $n = 12$) offspring. **e** Heat maps

tracking MMc^pos and MMc^low offspring when performing a recency recognition (RR) task (Arena size: D: 34 cm; H: 30 cm). **f** Same as (**d**), but depicting interaction time ($p = 0.0110$) and interaction counts ($p = 0.0257$) in a RR task performed by MMc^pos (left $n = 21$, right $n = 11$) and MMc^low (left $n = 22$, right $n = 11$) offspring. Significance in (**b**) was determined by the Friedman test followed by Dunn's multiple comparison test, ****$p < 0.0001$. Significance in (**d**) and (**f**) was determined by a two-sided Mann–Whitney U test, *$p < 0.05$, **$p < 0.01$. Data in (**b**), (**d**), (**f**) are displayed as mean ± SEM and each circle denotes one call or one pup.

immune cells into *Rag2^−/−IL-2rγc^−/−* pregnant female at E12.5 (Supplementary Fig. 11c). We termed the resulting group of *Rag2^+/−IL-2rγc^+/−* offspring 'MMc^low+AT' (Fig. 5a). In the MMc^low+AT offspring, the AT restored the absolute number of MMc in fetal and neonatal brain (Fig. 5b–d). MMc subset populations in offspring's brain were similarly to the distribution observed in MMc^pos offspring at E18 and P8 (Supplementary Fig. 11d, e). Moreover, the number of fetal and neonatal microglia was restored in MMc^low+AT offspring to the levels observed in MMc^pos offspring (Fig. 5e, f). Similarly, in the MMc^low+AT offspring, the enhanced presynaptic terminal elimination detected in MMc^low offspring was restored to levels comparable to those seen in MMc^pos offspring (Fig. 5g).

Concomitantly, the broadband oscillatory power in PL and HP was moderately recovered (Supplementary Fig. 11f–m). The discrimination between novel and old stimuli was similar to control animals upon restoring MMc and pre-juvenile MMc^low+AT offspring did not show the behavioral performance observed in MMc^low mice (Fig. 5h–j).

## Discussion

Here we investigated if and how MMc in the offspring's brain regulate fetal brain development and promote brain function and behavior. The use of an MMc reduction model allowed us to provide causal evidence for the importance of MMc in maintaining fetal microglia homeostasis, suppression of excessive presynaptic elimination, promotion of functional communication within the developing prefrontal-hippocampal circuits, and the advancement of early behavioral performance. These insights highlight that the unique population of MMc is not an

evolutionary leftover or accidental leak out present in eutherian mammals, but promotes brain development and function later in life. Intriguingly, fetal microglia increase when MMc are low. Our functional assessments highlight that MMc abrogate microglia hyperactivation, e.g., overshooting microglia proliferation and phagocytosis. This conclusion is supported by the excessive engulfment of presynaptic terminals ex vivo as well as the reduced neuronal branching and growth in vitro when MMc are scarce. Consequently, MMc play an important role in the regulation of tissue homeostasis in the developing brain.

The experimental approach we here chose was primarily geared towards the identification of the yet unknown physiological role of MMc on fetal brain development and later function. However, our focus on E18.5, the time point during fetal development closest to birth, precludes from detecting possible fluctuations of MMc in fetal brain at various stages of neurodevelopment throughout gestation, once MMc transfer occurs upon completion of placentation. Based on the insights presented here on the relevance of MMc for offspring's neurodevelopment and behavior, a focus on distinct time points during neurodevelopment will likely be highly relevant when assessing the impact of prenatal adverse events. These events can occur at certain days of fetal development, are well known to interfere with neurodevelopment and hence, likely also interfere with MMc phenotype and function[6,8,37].

Fetal brain development shows a high degree of sexual dimorphism, which is especially obvious in the context of adverse prenatal events[38,39]. Surprisingly, when identifying the number of MMc

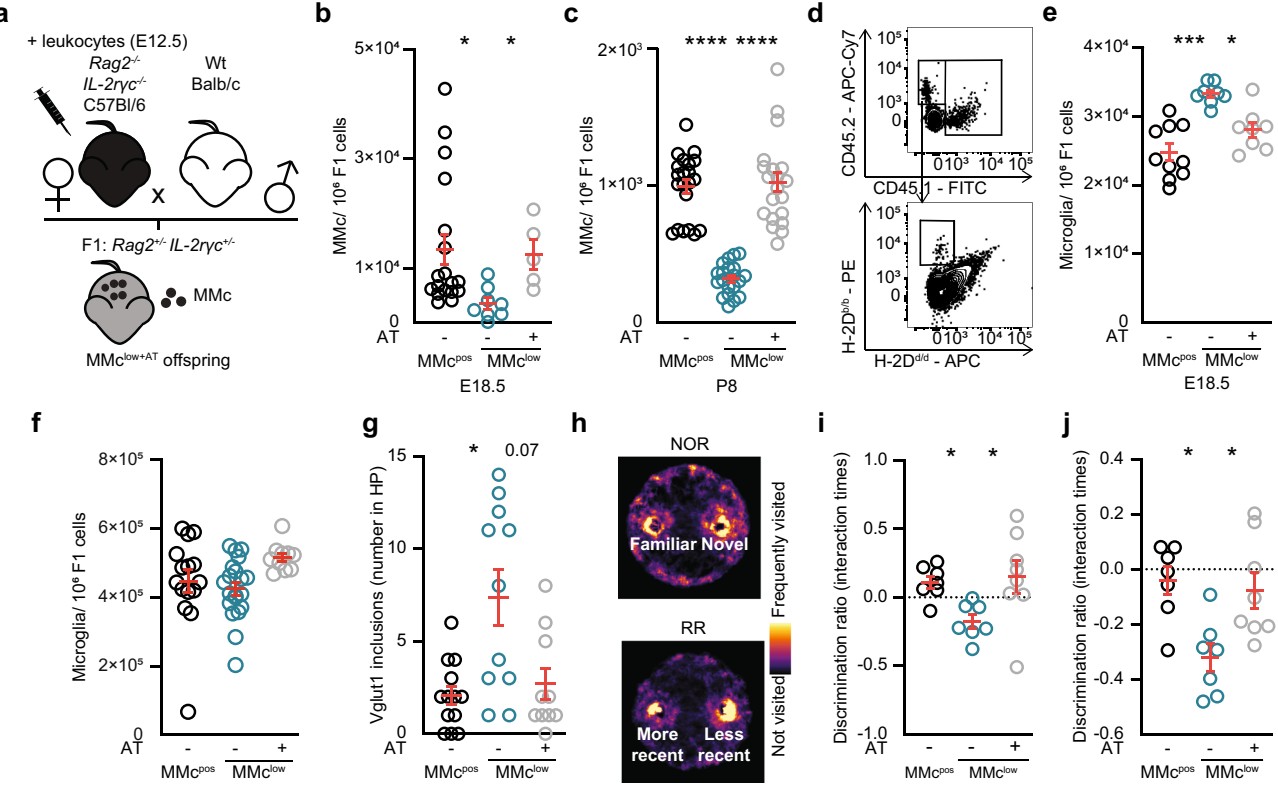

**Fig. 5 | MMc recovery in fetal brain restores microglial-mediated neuronal refinement and prevents behavioral disturbances. a** Experimental design used to restore the MMc level in MMc[low] offspring. Immune cells isolated from wt dams were adoptively transferred into *Rag2[-/-] IL-2rγc[-/-]* dams on E12.5. The resulting offspring were termed MMc[low+AT]. **b** Number of MMc in brain of MMc[low+AT] compared to MMc[low] and MMc[pos] offspring (MMc[pos] *n* = 18, MMc[low] *n* = 8, MMc[low+AT] *n* = 5), as identified by flow cytometry and assessed at E18.5 (MMc[pos] vs. MMc[low] *p* = 0.0137; MMc[low] vs. MMc[low+AT] *p* = 0.0204). For generation of MMc[low] and MMc[pos], see Fig. 2a. Offspring of all three groups were generated within the same experimental time frame. **c** Same as (**b**), but assessed at P8 (MMc[pos] *n* = 21, MMc[low] *n* = 21, MMc[low+AT] *n* = 21). **d** Representative contour plots of MMc detection in MMc[low+AT] offspring. **e** Number of fetal microglia, identified by flow cytometry in brain cells of MMc[pos] (*n* = 10), MMc[low] (*n* = 9) and MMc[low+AT] (*n* = 8) offspring at E18.5 (MMc[pos] vs. MMc[low] *p* = 0.0003; MMc[low] vs. MMc[low+AT] *p* = 0.0417). **f** Same as (**e**) but for neonatal (P8)

mice (MMc[pos] *n* = 15, MMc[low] *n* = 20, MMc[low+AT] *n* = 10). **g** Number of inclusions of presynaptic vesicles engulfed by microglial cells in the CA1 area of the hippocampus in MMc[pos] (*n* = 13), MMc[low] (*n* = 11), and MMc[low+AT] (*n* = 10) P8 offspring (MMc[pos] vs. MMc[low] *p* = 0.0251; MMc[low] vs. MMc[low+AT] *p* = 0.0745). **h** Heatmap tracking MMc[low+AT] in a novel object recognition (top) and recency recognition (bottom) task (Arena size: D: 34 cm; H: 30 cm). Areas of frequent visits appear in bright yellow. **i** Discrimination ratio for number of interactions (MMc[pos] vs. MMc[low] *p* = 0.0406; MMc[low] vs. MMc[low+AT] *p* = 0.0186) with the novel and **j** less recent object (MMc[pos] vs. MMc[low] *p* = 0.0224; MMc[low] vs. MMc[low+AT] *p* = 0.0449) of MMc[pos] (*n* = 7), MMc[low] (*n* = 7), and MMc[low+AT] (*n* = 8) offspring. Significance was determined by a Kruskal–Wallis one-way ANOVA followed by Dunn's multiple comparison test, **p* < 0.05, ****p* < 0.001, *****p* < 0.0001. Data in **b, c, e, f, g, i, j** are displayed as mean ± SEM and each circle denotes one pup.

in brain in pregnancies unchallenged by adverse events, we did not observe significant differences between male and female fetuses in wt mice, which suggests that the vertical transfer of MMc is not affected by the sexual dimorphism at the placental level. Due to hemizygosity of the γc gene in male MMc[low] offspring, we had to exclud male offspring from the assessments in the MMc[pos]/MMc[low] model. Future investigations are needed to assess the possibility of sex-specific MMc effects in the developing brain.

Since the fetal blood-brain barrier becomes functional during fetal development at E15.5[40], one may assume that MMc migration into the fetal brain discontinues as of this developmental milestone. Interestingly, published evidence reveals that breast milk-derived MMc may also be transferred to the offspring's brain postnatally [41], which would support a continuous MMc migration upon blood-brain barrier establishment. Clearly, our assessment of MMc in the fetal brain does not provide insights on the origin or differentiation fate of MMc prior to entering the fetal brain. We here observed that a large number of MMc in the fetal brain are microglia, but also T and B cells were present. In a previous study, we focused on the role of MMc in fetal bone marrow and could identify that the high frequency of

MMc are T cells, whilst the frequency of myeloid cells, e.g., macrophages, was low[17].

Similar to reports in other organs[17,42], we observed a decrease of MMc with increasing age, whereby MMc were still detectable at low numbers in offspring's brain at P60. To date, insights on pathways supporting such longevity of MMc–including organ-specific longevity of MMc–as well as the mechanisms leading to the decline of MMc with increasing offspring's age are still mostly unknown. Possible pathways that may explain the observed MMc decline over time may include a limited potential for self-renewal, or the death of MMc due to cellular exhaustion or absence of growth factors in the organ-specific microenvironment, e.g., the offspring's brain. Remarkably, MMc longevity may also be explained by the different MMc phenotypes that can be detected in the respective offspring's organs, such as the aforementioned T-MMc in bone marrow[17] or the microglia-MMc in the brain. These observations suggest either a preferential recruitment of these MMc subsets to the different fetal organs, or a disparate differentiation of progenitor-like MMc, dependent of the tissue microenvironment in which they seeded. This may subsequently also affect the lifespan of MMc in the different offspring's organs.

Tissue-resident macrophages like microglia in brain originate from erythromyeloid progenitors (EMP), which develop in the yolk sac, then migrate and seed into the fetal liver and subsequently colonize embryonic organs as EMP-derived macrophages. In the brain, these progenitors complete their differentiation into microglia, which are self-renewing throughout life and are only minimally replenished by circuiting macrophages[43,44]. In our study, initial screening experiments revealed that the number of progenitor cells did not differ in yolk sac of MMc[pos] and MMc[low] offspring at E9.5 (Supplementary Fig. 12a, b). However, since the overall number of such progenitor cells was extremely low, the validity and biological significance of these observations may still be limited. Future studies should aim at in-depth investigations of the microglia progenitor cells and their development in presence and absence of MMc infiltration in order to determine their potential interaction already in the yolk sac.

Given the importance of microglia for brain wiring, the present study focuses on the identification and functional consequences of the altered fetal microglia number seen in offspring with reduced or restored MMc. The observed decrease of T and B cells in brain of fetal MMc[low] offspring will be subject of future investigations, especially taking into account the functional role of T cells in autoimmune diseases affecting the brain.

The microglia engulfment of presynaptic terminals was used as readout of diverse neuronal interactions that control the development of circuits. In the present study, we were able to detect MMc in different brain regions, including the cerebellum. The in-depth functional investigation focused only on MMc in PFC and HP due to the role of these areas for cognitive processing in health and abnormal memory and cognitive flexibility in neurological and neuropsychiatric disorders[45–49]. Within the prelimbic-hippocampal networks investigated in the present study, we detected not only lower broadband power of oscillatory activity but also decreased communication between HP and PL. We recently identified that interference of the patterns of electrical activity within prefrontal-hippocampal circuits at neonatal age perturbs the network function and cognitive performance in adult mice[22]. Therefore, the lower oscillatory power and reduced communication within neonatal prefrontal-hippocampal networks in mice with low MMc might account for the observed recognition memory impairment. Further, the disrupted vocalization mirrors a global dysfunction of brain circuits already at neonatal age. Of note, the performance of pre-juvenile MMc[pos] offspring in recognition memory tasks was often below chance level. This may be explained by the mixed strain background of the MMc[pos]/MMc[low] offspring, which resulted from the allogenic mating combination of C57BL/6 females and Balb/c males. In fact, Balb/c mice have been shown to be more anxious and less explorative[50,51]. In addition, the MMc[pos] and MMc[low] offspring are also heterozygous for the *Rag2/IL-2rγc* genes. The genetic manipulation of *IL-2rγc* may result in an impaired behavioral competence[52]. However, these deficits of individual groups do not bias the robust differences observed between MMc[low] and MMc[pos] animals, since they all shared mixed strain background and are heterozygous for the *Rag2/IL-2rγc* genes.

In the present study, we focused on the emergence of cognitive abilities along neonatal and pre-juvenile development and did not extend the investigation of behavioral phenotype until adult age. The long-term effects of reduced number of MMc might be either milder and (partially) compensated or persistent, leading to life-long deficits. We recently identified critical time windows of cognitive development during which transient manipulation of electrical activity causes permanent reduction of network function and behavioral performance in memory tasks[22]. Similar processes may occur also in MMc[low] mice. The number of retained MMc declines with age, although low numbers are still detectable in mature offspring.

Another limitation is the monitoring of MMc by scRNA-seq solely on E18.5. Clearly, it would have been desirable to survey the gene expression in brain MMc throughout life. However, technologies enabling to isolate and assess cells of extremely low numbers at reasonable costs must still be optimized. Since our results strongly support an MMc-dependent suppression of fetal microglia activation and related synaptic pruning, they raise the question of how MMc may interact with fetal microglia in the developing brain. The outcome of our scRNA-seq analyses provides pivotal hints toward understanding how the interaction between MMc and fetal microglia is operational. We identified an upregulation of sensome genes in MMc, including *Cd47, Selplg, Cd37*, and *Il-6ra*, along with a downregulation of inflammatory genes. It has been shown that the microglia sensome conveys neuroprotection and is involved in host defense[27]. More specifically, CD47 protects synapses from excess microglia-mediated pruning during development[33]. This provides an explanation for the observation we made in MMc[low] offspring, where the reduction of MMc and hence, *Cd47*, was linked to an increased microglia-dependent pruning. Another gene expressed by MMc, the *Il-6r*, has similar beneficial functions, as repopulation of the brain with microglia is dependent on IL-6r pathways[53]. In addition, only a very low number of MMc expressed inflammatory genes, which may skew the microenvironment in the fetal brain toward homeostatic balance. Indeed, when MMc are low, we observed an upregulation of inflammatory genes in fetal microglia, e.g., *Tnf-α*, and *Ifn-β*, along with the downregulation of *Rab-7b*, which suppresses inflammation. Taken together, the data suggest a broad spectrum of possible pathways for interaction between MMc and fetal microglia, and, very likely, the entire microenvironment in the developing brain.

According to the present data, MMc seemed to cluster in the PFC. This unique profile of homeostatic and sensome genes may account for the observed neuronal refinement in specific brain areas. The course of pregnancy can be divided into immunologically distinct stages, including a brief inflammatory surge around the time of blastocysts implantation, followed by the long gestational period of anti-inflammation and immune tolerance to ensure that the fetus is not rejected. Parturition is then initiated by progesterone withdrawal and inflammation[54]. Especially the inflammatory period related to the onset of parturition may affect the transfer of MMc from mother to fetus. Moreover, adverse events occurring during the period of anti-inflammation during pregnancy in mice and humans, e.g., infection or related proxy, as well as trauma, skew maternal cells toward a pro-inflammatory phenotype. This has been shown to enhance the transfer rate of MMc from mother to fetus[55–58] and may possibly alter the function of MMc in various fetal organs. The here presorted data on the functional role of MMc during normally progressing pregnancies will enable to address the inflammation-induced alterations of MMc upon adverse prenatal events and the related consequences for offspring's brain and other offspring's organ development and function.

Taken together, our findings show that MMc promote the offspring's microglia homeostasis, brain wiring, and behavioral performance. Future studies should aim to assess MMc in human brain development as well as to establish MMc as a biomarker for the early recognition of neurodevelopmental disorders.

## Methods

### Experimental mouse model and times pregnancies

All procedures were approved by the University Medical Center Hamburg-Eppendorf institutional guidelines and institutional animal welfare officer. The procedures conform with the requirements of the German Animal Welfare Act. Approvals were obtained from the State Authority of Hamburg (Behörde für Justiz und Verbraucherschutz, Amt für Verbraucherschutz, Lebensmittelsicherheit und Veterinärwesen), Germany (G17/049, N18/111, ORG_927, ORG_1005). C57BL/6 *Rag2[-/-] IL-2rγc[-/-]* (*B6.129-Rag2[tm1Cgn]Il2rg[tm1Cgn]*), Balb/c CD45.1 *Rag2[-/-] IL-2rγc[-/-]*, and C57BL/6-Gt(ROSA)26Sor[tm4(ACTB-tdTomato,-EGFP)Luo]/J (Jax Stock No: 7676) mice were obtained from the animal facility of University Medical

Center Hamburg-Eppendorf. Balb/c CD45.1 (*CByJ.SJL(B6)-Ptprc^a^/J*, Jax Stock No: 6584) mice were purchased from The Jackson Laboratory and C57BL/6 mice from Charles River Laboratories. Mice were single-housed (males) or maintained in groups (females) with food and water provided ad libitum in a 12-h light/12-h dark cycle at a room temperature of 21 °C and controlled humidity at 43%. Males were used for mating from fertile age up until ~1 year of age. Eight- to ten-week-old virgin female mice were mated according to mating combinations shown in Figs. 1a, 2a, and 5a. The presence of a vaginal plug in the morning was considered as E0.5.

## Tissue processing

On E18.5, fetuses were sacrificed by decapitation. To obtain single-cell suspensions, fetal brains were disrupted mechanically and washed through a 40 μm cell strainer. At P8 and P60, offspring were sacrificed, brains were mashed and filtered through a 40 μm cell strainer. Cell pellets were resuspended in 30% Percoll, containing RPMI, and gently over-layered on 35% Percoll and PBS. Using 10% ketamine/2% xylazine in 0.9% NaCl solution (10 μg/g body weight), some P8 mice were anesthetized intraperitoneally and transcardially perfused with PBS followed by Histofix containing 4% paraformaldehyde. After decapitating, brains were stored in Histofix containing 4% paraformaldehyde until further processing by immunohistochemistry or optical clearing. Maternal PBMCs were collected in EDTA-coated tubes after scarifying the animals. After centrifugation, plasma was stored at −80 °C until further processing. Lysis of erythrocytes was performed using 1x RBC lysis buffer for 5 min at room temperature. Afterward, single-cell suspensions were used for flow cytometric analysis.

## Maternal microchimeric cell enrichment and flow cytometric quantification

For antibody staining, $1 \times 10^6$ cells were used. In order to block unspecific FcγRII/III binding, cells were incubated in normal rat serum and TruStain fcX (anti-mouse CD16/32, BioLegend, Cat. No. 101320). Afterward, the below mentioned antibodies were added in pre-defined dilutions. Simultaneously, either Fixable viability dye eFluor506™ or Zombie Yellow™ viability kit was added and incubated for 30 min at 4 °C on ice in the dark.

Next, fetal brains were stained with an anti-H-2K^d/d^/H-2D^d/d^ antibody (1:100, BioLegend, Cat No. 114714) conjugated to APC. Stained cells were then resuspended in 80 μl per $10^7$ cells. Then cells were mixed with 20 μl anti-APC antibody-conjugated beads and incubated for 15 min, followed by washing with 2 ml of buffer. The cells were then resuspended in 500 μl of buffer and passed over a magnetized LS column (Miltenyi Biotech). The column was washed with 3 ml of buffer three times and then removed from the magnetic field. Afterward, the bound cells were eluted by 5 ml of buffer through the column with a plunger. The resulting enriched cell suspensions were stained with the following fluorescent dye-conjugated antibodies against surface antigens were used to identify MMc among the offspring's cells: anti-CD45.1 (1:400, BioLegend, Cat. No. 110705), anti-CD45.2 (1:100, BioLegend, Cat. No. 109823), anti-H-2D^b/b^ (1:20, BioLegend, Cat. No. 111507), anti-H-2D^d/d^ (1:200, BioLegend, Cat. No. 114712)[59,60].

The following fluorescent dye-conjugated antibodies against surface antigens were used to phenotype cells from maternal and fetal origin: anti-CD11b (1:200, BD, Cat. No. 562128), anti-CD11b (1:200, BD, Cat. No. 552850), anti-CD45R/B220 (1:100, Invitrogen, Cat. No. 56-0452-80), anti-CD45R/B220 (1:100, BioLegend, Cat. No. 103241), anti-CD3ε (1:200, ThermoFischer, Cat. No. 61-0031-82), anti-FoxP3 (1:200, Invitrogen, Cat. No. 4340669), anti-CD11c (1:100, BioLegend, Cat. No. 117335), anti-Ki-67 (1:100, BioLegend, Cat. No. 563757). Using a LSR Fortessa II (BD Life Sciences) flow cytometry $0.3 \times 10^6$ leukocyte events were acquired with the Diva Software (v.9).

Data analyses were performed using FlowJo software (BD, Ashland, Oregon, USA) by excluding doublet cells, dead cells, and then following individual gating strategies. Fluorescence minus one (FMO) controls, which contain all antibodies of a staining panel except one, were included as controls to set gating thresholds for cell populations positive for the respective missing antibody.

## ScRNA-sequencing data alignment and pre-processing

10x Genomics raw sequencing data were processed using CellRanger software (version 3.0.2, 10x Genomics, Pleasanton, CA) and the 10x mouse genome 10 3.0.0 release as the reference (function cell ranger count). The matrices of cells and the unique molecular identifier (UMI) count were obtained and further processed by the R package Seurat (version 3.1.4)[61]. As a quality control (QC) step, we first filtered out genes detected in <3 cells and those cells in which <200 genes had non-zero counts. To remove potential doublets, cells with more than 5000 annotated genes were excluded. We further removed low-quality cells with more than 10% mitochondrial genes of all detected genes.

We aimed to verify the purity of the sorted and sequenced MMc sample by scRNA-seq, however, the utilized markers (CD45.1, CD45.2, H-2D^b/b^, H-2D^d/d^) had too small base pair differences. Also excluding fetal cells, contaminating the MMc sample by using paternal-specific SNP (Balb/c vs. C57BL/6) was not possible, due to limited sequencing depth.

## Clustering analysis and cell population identification

The Seurat R package (version 3.1.4) was used to perform unsupervised clustering analysis on scRNA-seq data. Gene counts for cells that passed QC were first normalized by library size and log-transformed (function NormalizeData, normalization.method = "LogNormalize", scale.factor = 10000). Then, highly variable genes were detected (function FindVariableFeatures, selection.method = "vst", nfeatures = 2000). Principal component (PC) analysis was performed on the scaled data (function RunPCA, npcs = 30) in order to reduce dimensionality. The selected PCs were then used to compute the KNN graph based on the euclidean distance (function FindNeighbors), which later generated cell clusters using the function FindClusters. The resolution of FindClusters function was set to 0.4 by exploration of top marker genes of each cluster. Uniform Manifold Approximation and Projection (UMAP) was used to visualize clustering results. The top differentially expressed genes in each cluster were found using the FindAllMarkers function (min.pct = 0.25, logfc.threshold = 0.25) that ran Wilcoxon rank sum tests. The top expressed genes were then used to determine the cell population of each cluster. Differential gene expression analysis of each microglia subpopulation within MMcs was computed by Model-based Analysis of Single-cell Transcriptomics (MAST) R package v.1.10.0[62].

## RNA sequencing and bioinformatic analysis

After isolation, the RNA integrity was analyzed with the RNA 6000 Pico Chip on an Agilent 2100 Bioanalyzer (Agilent Technologies). RNA-Seq libraries were generated using the SMART-Seq v4 Ultra Low Input RNA Kit (Clontech Laboratories) as per the manufacturer's recommendations. From cDNA, final libraries were generated utilizing the Nextera XT DNA Library Preparation Kit (Illumina). Concentrations of the final libraries were measured with a Qubit 2.0 Fluorometer (Thermo Fischer Scientific) and fragment lengths distribution was analyzed with the DNA High Sensitivity Chip on an Agilent 2100 Bioanalyzer (Agilent Technologies). All samples were normalized to 2 nM and pooled equimolar. The library pool was sequenced on the NextSeq500 (Illumina) with $1 \times 75$ bp, with ~20 Mio reads per sample. Sequence reads were aligned to the mouse reference assembly (GRCm38.98) with STAR (v.2.7.2d)[63]. Normalization and differential expression analysis were carried out with DESeq2[64]. Genes were considered to be differentially expressed if an absolute log2 fold change >1 and an FDR smaller than 0.05 were observed.

## Primary neuronal cultures

For primary neuron cultures, E15.5 cortices were isolated and freed from hippocampus and meninges[65]. Tissues were digested with 0.05% Trypsin-EDTA (1 ml/pup) for 5 min at 37 °C. Digestion was stopped with 10% FBS in HBSS and supernatant was removed. Tissue was dissociated and $2 \times 10^3$ neurons per well were seeded in coated (5 μg/ml poly-l-lysine) 96-well plate in BrainPhys Neuronal Medium containing penicillin/streptomycin (100 U/ml) and L-glutamine (2 mM) and incubated at 37 °C and 5% $CO_2$. Half medium was changed every 3 days using BrainPhys Neuronal Medium without glutamine.

## Microglia culture

Mixed glia cultures were prepared from embryos at E15.5. Cortices were isolated and freed from hippocampus and meninges. Tissues were minced and digested with 0.05% Trypsin-EDTA (1 ml/pup) and DNase I (5 μl/ml trypsin) for 30 min at 37 °C. Supernatant was removed and tissues were washed twice with HBSS. Tissues were dissociated in DMEM low glucose supplemented with FBS (10%) and penicillin/streptomycin (100 U/ml). Cell suspension was filtered (100 μm mesh) and centrifuged for 10 min at $200 \times g$ and 4 °C, followed by removal of the supernatant. Cells were resuspended in DMEM and counted. $4 \times 10^6$ cells were seeded in coated (40 μg/ml poly-l-lysine) 75 cm² culture flasks and incubated at 37 °C and 5% $CO_2$. After 24 h, the cultures were washed once with PBS, followed by incubation in DMEM. DMEM was changed at day 8 of culture. After 15 days, microglia cells were isolated using a rotary shaker. The microglia-containing medium was collected and centrifuged for 7 min at $200 \times g$ and 4 °C. Supernatant was removed except for 0.5 ml and cells were resuspended and counted using a hemocytometer.

## Neuron and microglia co-culture

Previously isolated microglia were added to the primary neuronal culture ($2 \times 10^3$ neurons and $3 \times 10^3$ microglia/well) and co-cultured at 37 °C and 5% $CO_2$ for 24 h. Cultures were fixed using 4% paraformaldehyde for 10 min, then washed in PBS.

## Immunohistochemistry

Perfused brains were postfixed in 4% paraformaldehyde for 24 h and then sectioned coronally at 50 μm. Slices were permeabilized and blocked with PBS containing 0.3% Triton X-100, 3% normal bovine serum, and 0.05% sodium azide. Subsequently, slices were processed by incubating them overnight at 4 °C with anti-Iba-1 (1:500, Wako Pure Chemical, Cat. No. 019-19741), anti-Vglut1 (1:1000, Millipore, Cat. No. AB5905), and anit-Vglut2 (1:500, Synaptic Systems, Cat. No. 135404), followed by 1 h incubation with goat-anti-guinea pig (1:500, AF488, Invitrogen, Cat. No. A-11073), donkey anti-rabbit (1:500, AF568, Invitrogen, Cat. No. A-10042) secondary antibodies and Hoechst33258 (1:5000, Sigma, Cat. No. 94403). Then, slices were transferred to glass slides and covered with Fluoromount-G. Brain sections were analyzed by Gram Stain[66].

Fixed neuron-microglia co-cultures were permeabilized with 0.3% Tween20 and 3% BSA in DPBS for 4 min. After washing with DPBS co-cultures were incubated with the primary antibody anti-MAP2 (1:1000, Sigma-Aldrich, Cat. No. M1406-100UL) for neuron and anti-Iba-1 (1:500, Wako Pure Chemical, Cat. No. 019-19741) for microglia detection at 4 °C overnight. Wash step was repeated, and co-cultures were blocked with donkey serum (5% in DPBS) for 1 h at room temperature. After washing, the secondary antibodies and nuclei stain were added to the co-cultures (donkey anti-rabbit, 1:500, AF568, Thermo Fischer Scientific, Cat. No. A-10042; donkey anti-mouse, 1:300, AF488, Dianova, Cat. No. 715-546-150; Hoechst33258, 1:5000, Sigma-Aldrich, Cat. No. 94403). After 2 h at room temperature, co-cultures were washed and kept in DPBS.

## In vivo electrophysiology and analysis

**Surgery.** Multisite extracellular recordings of local field potential (LFP) and multi-unit activity (MUA) were performed simultaneously from the prelimbic area (PL) of the medial PFC and the stratum pyramidale of CA1 area of the intermediate/ventral HP of P8 mice. For recordings in anesthetized state, mice received an intraperitoneal injection of urethane (1 mg/g body weight; Sigma-Aldrich) before surgery. For both groups, the surgery was performed under isoflurane anesthesia (induction: 5%; maintenance: 1.5%). Two plastic bars mounted on the nasal occipital bones were fixed with dental cement and used to enable the fixation of the head into a stereotaxic apparatus. The bone above the PFC (0.5 mm anterior to bregma, 0.1 mm right to the midline) was carefully removed by drilling a hole of <0.5 mm in diameter. Before recordings, mice were allowed to recover for 10–20 min on a heating blanket.

**Analysis.** Extracellular signals were acquired for 30 min, band-pass filtered (0.1–9000 Hz), and digitized (32 kHz) using a multichannel extracellular amplifier (Digital Lynx SX; Neuralynx, Bozeman, MO, USA) with the Cheetah acquisition software (v.6, Neuralynx). Afterward, pups were transcardially perfused and brains were sectioned coronally at 100 μm. After Nissl staining, the images of the sections were acquired by wide-field fluorescence images in order to confirm the position of the recording electrode defined by DiI (Lipophilic membrane stain) traces.

Active periods were detected with a custom-written algorithm[67]. The LFP was band-pass filtered (4–20 Hz) and downsampled to 100 Hz. The signal (raw and z-scored) was then passed through a boxcar square filter with a length of 500 ms and thresholded with a hysteresis approach. Active periods were detected as signal peaks exceeding an absolute or relative threshold (100 μV or 4 standard deviations, respectively) and subsequently extended to all neighboring time points that exceed a lower threshold (50 μV or 2 standard deviations, respectively). If the interval between two active periods was shorter than 1 s, the two were merged. Active periods whose duration was smaller than 300 ms were discarded. From the detected active periods, the following features were extracted: maximum amplitude (on the raw signal), average duration, number of active periods per minute, and proportion of time spent in active periods on the total recording length.

For power spectral density analysis, we used Welch's method with non-overlapping windows. For MUA analysis, data were band-pass filtered (500–5000 Hz). MUA was quantified as previously described[68]. Briefly, MUA was detected as the peak of negative deflections exceeding five times the standard deviation of the previously filtered signal and having a prominence larger than half the peak itself. Firing rate was computed by dividing the total number of spikes by the duration of the analyzed time window. The calculation of the imaginary coherence C over frequency ($f$) for the power spectral density, P of signals X and Y was performed according to the following formula: $C_{XY}(f) = |Im(\frac{P_{XY}(f)}{\sqrt{P_{XX}(f)P_{YY}(f)}})|$. Phase amplitude coupling (PAC) was computed according to Hartung et al.[69].

**Entropy and mutual information of/between spike trains.** Information theory-related parameters were computed on MUA spike trains of the entire recording of the recording sites in the PL and the CA1 stratum pyramidale (the reversal channel), binned at a 50 ms resolution and summed across electrodes. The entropy of such spike vectors and the mutual information between PL and CA1 were computed using *entropy* and *mutInfo* functions of the PRMLT Matlab toolbox: https://github.com/PRML/PRMLT. Given that there are no differences in the average number of MUA firing rates between the conditions that we investigated (Supplementary Fig. 9m), the results are unlikely to be biased by the number of detected spikes.

## Developmental and behavioral assessment

**Development.** On P3, 5, and 7 mouse pups were tested for somatic development according to a modified Fox battery[70,71]. Physical landmarks such as body weight but also surface righting, vibrissae reaction, cliff avoidance, and grasping reflex index were recorded.

**Maternal nest building quality.** On E16.5, the maternal nesting material was removed and replaced by three cotton nestlets (ZOONLAB GmbH, Castrop-Rauxel, Germany) were provided in order to enable the females to build a net using this non-familiar nesting material. After 24 h, the newly built nests were photo-documented and maternal nest building quality was scored by three independent observers, data are presented as mean of all three scores per mouse, according to published criteria[72]. In short, one point was given for a nest in which the cotton was not noticeably touched (more than 90% intact) (bad nest), five points were given for a perfect nest with more than 90% of the cotton torn, visible crater with walls (good nest).

*Mother-pup interaction:* The behavior of each dam was observed at P2, P4, and P6 in the second half of the light phase. All mice were housed in individually ventilated cages. In order to quantify mother-pup interaction, home cages were placed under a LAF cabin in the local animal facility. A Plexiglas cover replaced the filter top in order to improve visibility of the nest. A webcam (Logitech Europe S.A) connected to a notebook was mounted 60 cm above the cage and maternal behavior was filmed for 60 min using Logitech Capture Software. Time spent on the nest was defined as maternal caring behavior, whereas time spent off the nest as non-caring behavior. After 30 min of habituation, the behavior was scored in 90 s intervals using a scan-sampling approach. For this purpose, Solomon Coder software (Beta 19.08.02, ®András Péter) was used and 86 videos were scored by three independent observers. The percentage of time spent in the nest was calculated from 20 snapshots for the three days.

*Pup retrieval test:* At P6, maternal behavior was further assessed by a pup retrieval test[73]. The home cage was filmed using the same equipment as for mother-pup interaction and videos were analyzed by three independent observers using BORIS event logging software[74].

*Ultrasonic vocalization:* Recordings of ultrasonic vocalizations (USVs) were conducted in an acoustically isolated room at constant light intensity and temperature according to established protocols. The pup was isolated from the littermates and mother by placing it into a Plexiglas box (L: 34 cm, W: 17 cm, H: 19 cm) 19 cm below a condenser microphone (CM16/CMPA, Avisoft Bioacoustics, Berlin, Germany)[75] that is sensitive to frequencies from 10 to 200 kHz. The USVs were recorded for 600 s at a sampling rate of 250 kHz (16-bit format) using the ultrasound recording interface (Avisoft UltraSoundGate, Avisoft Bioacoustics). For training phase of utilized deep learning methods[76] in Matlab, a blind rater classified USVs into three groups (simple, frequency jumps, and complex).

For the behavioral tests described below, mice were tested from P19 onward using previously established protocols[68,77,78]. Briefly, all tests were conducted in a circular white arena, the size of which (D: 34 cm, H: 30 cm) maximized exploratory behavior, while minimizing incidental contact with testing objects[79]. A black and white CCD camera (Video Technical E. Hartig) was mounted 100 cm above the arena and connected to a computer via PCI interface serving as frame grabber for video tracking software (Video Mot2 software, TSE Systems). After every trial, the arena and the objects were cleaned with 0.1% acetic acid to remove all odors.

**Open field.** Pre-juvenile mice (P16–17) were investigated using a previously established protocol[68] that allowed them to freely explore the circular testing arena (D: 34 cm) for 10 min. The floor of the testing area was digitally sub-divided in 8 zones (4 center zones and 4 border zones) using the zone monitor mode of the VideoMot 2 analysis software (VideoMot 2,TSE Systems GmbH). The exploratory behavior of each mouse was quantified by the time spent in center and border zones as well as the running distance and velocity.

**Novelty recognition paradigms.** The protocols for assessing item recognition memory in pre-juvenile mice consisted of familiarization and testing trials as previously described[68]. During the familiarization trial, each mouse was placed into the circular testing arena that contained two identical objects (D: 1.5–3 cm; H: 3 cm) without resembling living stimuli (no eye spots, predator shape). The mice were released against the center of the opposite wall with the back to the objects. After 10 min of free exploration of objects, the mice were returned to a temporary holding cage. In the novel object recognition (NOR) task, tested in P17–P18 mice, the familiarization trial (5 min long) was followed by a test trial in which one object used in the familiarization and one new object were placed in the arena at the same location as during the familiarization trials. The mice were allowed to investigate the familiar and the novel object, with different shape and color, for 5 min. Object interaction during the first five minutes and the length of single interaction with the objects were monitored and compared between the groups. In the recency recognition (RR) task, tested at P19–22, mice experienced two 10 min familiarization trials with two different sets of identical objects (D: 1.5–3 cm; H: 3 cm) that were separated by an interval of 30 min. Five minutes after the second familiarization trial, a test trial followed in which one object from the first and one object from the second more recent familiarization attempt were placed in the arena in the same location as during the familiarization trials. Object interaction during the first five minutes and the length of single interaction with the objects were monitored and compared between the groups. All trials were video-tracked using the Video Mot2 analysis software. The object recognition module of the software was used and a 3-point tracking method identified the head, the rear end and the center of gravity of the mouse. A square zone was digitally created around each of those objects, and each entry of the headpoints into this area was considered an object interaction. Climbing or sitting on an object defined as having both head and center of gravity points in a square zone did not count as an interaction. Data were imported and analyzed offline using custom-written tools in MATLAB R2017b software (MathWorks, USA). Discrimination ratios were calculated as (Time spent interacting with novel object – time spent interacting with less recent object)/(Time spent interacting with novel object + time spent interacting with less recent object). Single interaction time ratios were analogously calculated.

**Working memory.** A custom build white opaque Y-shaped maze (L: 30 cm, W: 7 cm, H: 15 cm) with three identical arms in 120° angle to each other was used to assess the working memory via spontaneous alternation performance in P24 mice. The mouse is placed in the start arm (always arm A) facing the wall and can then freely explore the maze until 25 arm choices are made. Arm entries are defined as all four paws placed the arm. The mouse is recorded as described previously. The percentage of spontaneous alternation (i.e., sequential entries into all three arms) was calculated by dividing the number of alternations by the number of possible alternations [number of alternation/(number of total arm entries−2)]. In addition, percentages of alternate arm return and same arm return were calculated. After every trial, the maze was cleaned with 0.1% acetic acid to remove all odors.

**Repetitive-like behavior.** On P26 mice were examined in a marble burying essay using previously published protocols[80]. Clean cages (H: 37 cm, W: 21 cm, H: 15 cm) were filled with 4.5 cm aspen wood cage bedding (ABEDD Vertriebs GmbH), followed by arranging 20 blue glass marbles (13 mm diameter, Murmelein UG) equidistant in a 4 × 5 alignment. The exploration and digging behavior were recorded for 30 min as described before. The number of marbles buried was scored at 5, 10, 15, 20, and 30 min by three independent observers. A marble was

considered as buried if >75% of the volume is covered. Marble burying index shows the mean of counted buried marbles of all three observers.

## Adoptive transfer experiments

On E12.5, peripheral blood, uterus-draining inguinal and paraaortic lymph nodes, and spleen of CD45.1 Balb/c-mated pregnant wt C57BL/6 and tdTomato⁺ C57BL/6 females were collected. Lymph nodes and spleen were grinded and washed through a cell strainer and centrifuged. Spleen cell pellets were resuspended with 5 ml PBS and over-layered on 5 ml lympholyte®-M (CEDARLANE, Cat. No. CL5031). Red cell blood (RBC) lysis was performed on peripheral blood using RBC lysis buffer (eBioscience) according to manufacturer's instruction. Single-cell suspensions of all tissues were pooled, cell numbers were determined using a Neubauer chamber and readjusted to $1 \times 10^7$ cells/ml. Cell frequencies were determined in the cell suspensions and revealed a frequency of approx. 60% B, 30% T, and 10% myeloid cells, as well as low frequencies of NK cells, macrophages, lymphoid, and plasmacytoid dendritic cells (Supplementary Fig. 11c)

To restore the immune cell deficiency in $Rag2^{-/-} IL-2r\gamma^{-/-}$ females, CD45.1 Balb/c-mated pregnant $Rag2^{-/-} IL-2r\gamma^{-/-}$ females were anesthetized using $CO_2/O_2$ on E12.5. Then, $1 \times 10^7$ cells (0.2 ml of the above-described cell suspension) were injected intravenously.

## Statistical analysis

Statistical analyses were performed with GraphPad Prism (version 8.4.3) and R Statistical Software. A compilation of sample sizes of all behavioral experiments is provided in Supplementary Table 5 and all statistical results and exact $p$ values are included as Supplementary Table 6. Statistical parameters including sample sizes and dispersion are reported in the figures and their legends. Assumptions regarding normality and homogeneity of variances were formally tested before assessing statistical differences between groups. If data did not meet test prerequisites, non-parametric test equivalent and respective post hoc tests were used. Outlier were excluded if >90th and <10th percentile. Statistical differences between two groups were determined by unpaired, two-tailed Student's $t$-test. One-way analysis of variance (ANOVA) with Dunn's multiple comparisons post hoc test was used to statistically compare three groups. Data was defined as statistically significant when at least $p < 0.05$. Investigators of behavioral assessments and in vivo electrophysiology were not blinded to the groups of mice, but analyses were performed by at two independent observers of which one was blinded. In order to normalize litter, sex, and maternal caring effects on the pups, at day of birth (P0) pups were reduced to six littermates (3 females, 3 males) in total.

## Reporting summary

Further information on research design is available in the Nature Research Reporting Summary linked to this article.

# Data availability

The raw scRNA-sequencing data generated in this study have been deposited in the Gene Expression Omnibus database under accession code GSE207327. The RNA-sequencing data generated in this study have been deposited in the European Nucleotide Archive (ENA) under accession code PRJEB54823. The flow cytometric and LFP data used in this study are available in the Zenodo database under https://doi.org/10.5281/zenodo.6797300. The imaging data used in this study are available in the Zenodo database under https://doi.org/10.5281/zenodo.6797634 and https://doi.org/10.5281/zenodo.6797827. Source data are provided with this paper.

# Code availability

All custom written codes used in the current study are available on GitHub: https://github.com/OpatzLab/HanganuOpatzToolbox.

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

## Acknowledgements
We thank Agnes Wieczorek, Thomas Andreas, Annette Marquardt, Peggy Putthoff, Achim Dahlmann, Kristin Titze and Sandra Ehret for technical assistance. We thank Urte Matschl for support with Luminex measurements. The support from the staff members of the institutional core facilities (Microscopy Imaging Facility, FACS Sorting Core Unit, and Animal Housing Facility) and the NGS division of the Heinrich Pette Institute Hamburg (Dr. Daniela Indenbirken) is also appreciated. We thank Prof. Judith Eckert for support in yolk sac isolations. We thank Prof. Dr. Carlos A. Guzmán for providing B6.*129-Rag2*tm1CgnIl2rgtm1Cgn mice, Dr. Jan-Eric Turner for providing *Balb/c-Rag2*−/− *IL-2ryc*−/− mice, and Dr. Nicola Wanner for providing C57BL/6-*Gt(ROSA)26Sor*tm4(ACTB-tdTomato,-EGFP)Luo/J

mice. This work was supported by German Research Foundation (HA 4466/10-1, HA4466/12-1, HA4466/20-1: 178316478 CRC 936 B5 to I.L.H.-O.; KFO296: AR232/25-2, FOR5068: AR232/29-1 to P.C.A.; CRC1192 to V.G.P.); the Federal Ministry of Education and Research (BMBF: eMed Consortia Fibromap to V.G.P); the European Research Council (ERC-2015-CoG 681577, MSCA-ITN-H2020-860563 to I.L.H.-O.); and State Research Funding, Authority for Science, Research and Equality, Hamburg, Germany (LFF-FV73 and LFF-FV76 to P.C.A. and I.L.H.-O. respectively). S.S. received a fellowship from the *Forschungsförderungsfond* of the University Medical Center Hamburg-Eppendorf.

## Author contributions
S.S., I.L.H.-O., and P.C.A. designed and conceptualized the study. I.L.H.-O. and P.C.A. supervised all experiments and obtained funding. S.S., V.S., C.U., M.P., A.W., M.-T.T., D.Z., N.F., and J.H. carried out experiments. N.V., A.W., and V.G.P. performed brain clearing and imaging experiments. S.S., M.C., C.U., K.T., and M.P. analyzed data. T.S. and Y.Z. executed bioinformatic analyses for scRNA-seq under supervision of S.B. M.A. analyzed transcriptome data. S.S., M.C., K.T., T.S., Y.Z., M.P., S.B., N.G., I.L.H.-O., and P.C.A. interpreted the data. S.S., I.L.H.-O., and P.C.A. wrote the manuscript with input and edits from all authors.

## Funding

## Competing interests
The authors declare no competing interests.

## Additional information

**Peer review information** *Nature Communications* thanks Surendra Sharma and Other anonymous Reviewer(s) to the peer review of this work. Peer review Reports are available.

