## [Peer Review File · Nature Communications]

REVIEWER COMMENTS

Reviewer #1 (Remarks to the Author):

This manuscript focuses on the role of maternal microchimerism on neurodevelopmental and behavioral changes in the offspring. The authors claim that maternal microchimeric cells (MMc) of different lineage in the fetal brain provide regulatory help in shaping normal neurodevelopment and behavior. The authors have performed some clever experiments and presented their data in an informative manner. This is a complex theme and the data presented need to have a critical analysis and interpretation. Several concerns remain unaddressed.

Major comment:

1. The authors are analyzing tdTomato-positive MMc at E18.5. Did the authors look at E14, E16, or even earlier for their presence? It is important because some of the analyses the authors have presented may relate to neurodevelopmental issues such as autism and schizophrenia later in the offspring. Although the authors show in Fig. 1b that there is no difference in MMc content in males vs. females, it is surprising as even at the placenta level, sexual dimorphism makes an impact on the onset of fetal brain development. MMc significantly decrease as the offspring ages. Is it because some lineage cells of MMc do not renew themselves or are eliminated due to lack of their growth factors in the brain microenvironment. In humans, MMc can be detected after 27 years of birth (Bianchi et al). Did the authors analyze MMc at P60 using the methods described in Fig. 1e and g?
2. The authors state that MMc can be found in different organs. Is there any information available about their lifespan in these organs? Or does the brain support a different lifespan of MMc?
3. Pregnancy can be divided into three phases: 1. Implantation (inflammation), gestation (anti-inflammation), and parturition (inflammation). Since E18.5 is very close to the onset of parturition, is inflammation, lack of pregnancy hormones, or infection expected to impact the transport, and the content and/or function of MMc?
4. The authors have focused a lot on brain homeostasis and dismantling of synaptic connections. What does exactly brain homeostasis mean? In Fig. 2 and related Extended Figures, the authors have invoked DNA methylation mechanisms, fetal T and B cell distribution, and transcriptome choreography. It is suggested that genes such as Rab-7b were down-regulated in microglia of MMc low offspring. Are there scenarios where the brain is mainly populated with MMc low? Does this result in brain disorders? Is transcriptome profile different in Wt male vs. Wt female?
5. Figures 2 and 3 describe a solid experimental plan to rule out the contribution of immune cells in MMc-mediated brain development. The authors used allogeneic mating protocol involving Wt C57BL/6

female and Rag2IL-2 ryc deficient male mice. In another mating, they reversed the mating partners and analyzed the MMc content. They identified two types of neonates – MMcPos and MMc low. What dictates this very distinguished difference? Why some offspring become MMclow? Do MMclow offspring entail poor brain homeostasis? In Fig. 3, the authors employ ultrasonic vocalization as one of the tools to differentiate between MMcPos and MMclow offspring. Although the authors claim significant differences, the data may not support this claim strongly. Yes, there are clear cut differences in discrimination profiles, the number MMclow offspring used in the experiments is lower. A careful look at the ultrasonic vocalization suggest that MMc low offspring experience the same vocalization as that described autistic children, particularly male offspring. Do authors have any comment on this?

6. Adoptive transfer of MMc in MMclow offspring seems to restore microglial engulfment (?). How many MMc needed to be transferred? Is there a kinetic threshold of adoptive transfer ? What happens to the offspring at P60 after adoptive transfer?

Minor comments:

1. On line 118, it should be Fig. 2b-c not Fig. 3b-c. Also, on the same page line 132, it should be “to a lesser extent”.
2. Although the authors discuss why fetal microglia increase when MMc are low, it is not entirely clear why MMc become low and affect only microglia. Otherwise, Discussion is well written.

Reviewer #2 (Remarks to the Author):

This is a very interesting manuscript designed to determine the impact of maternal microchimeric (MMc) cells in fetal mouse brain on synaptic development, circuit function, and behavior. The authors report a significant number of the MMc within the fetal brain are microglia with high expression of homeostatic genes, and that these cells primarily localize to the PFC and HP. In the absence (or reduction) of MMc, there are more microglia in the fetal brain and they engulf more presynaptic markers. Moreover, offspring exhibit altered communication and cognitive behaviors and changes in network activity. Finally, adoptive transfer of leukocytes back to immunodeficient dams largely rescues these phenotypes. Overall there is a large amount of compelling data and the rigor and importance of the work seems high. There are some concerns which are outlined here.

1. Fig. 1 – panel f is confusing. It looks like according to the color coded dots for different cell types that endothelial cells also express high levels of the canonical microglial markers Cx3cr1, P2yr12, and Sparc? Similarly, for panel h, it says the heatmap of gene expression is based on genes that were more than 50% higher in MMc compared to fetal microglia, but the heatmap also shows genes with decreased fold

change? Finally, for panel g, these gene categories are subjective and context-dependent. The genes included in each category should be provided rather than assigning them to categories absent a functional readout. For instance, *tmem119* is often considered a “homeostatic” gene in microglia but its expression is not particularly high in the microglia clusters in panel f.

2. Fig. 2 - Determination of cell number using flow cytometry is not very reliable. There are too many variables regarding cell loss with the isolation and method of gating. Quantitative claims should be shown in the intact brain using IHC or similar.

3. Why was Vglut1 assessed for pruning by microglia? These label short range projections (e.g. intracortical) at P8, in contrast to Vglut2 which labels long-range projections. It is surprising given the claim that these synaptic changes underlie the circuit deficits in the mice, as the relevant long-range projection synapses were not assessed. It is therefore less convincing that these synaptic changes are linked to the network activity changes reported.

4. For fig 3d and f, it actually appears that the controls (MMcpos) mice show no discrimination as they are roughly at chance, whereas the MMclow do, and avoid the novel object. This is not a cognitive deficit but perhaps novelty avoidance. In any case they show better discrimination if I am interpreting this correctly. More concerning is the fact that the control group does not show normal discrimination so the data are difficult to interpret.

5. The authors make no comment on the mechanism by which immune cell deficiency in dams impacts microglial number and/or function. Are there changes in the yolk sac progenitors or only changes once they arrive into the CNS?

6. Why were cognitive behaviors tested at P19-24 in offspring? This is interesting because it is the time in which the hippocampal circuitry is just maturing, and any delay or alteration in this maturation could impact the behavioral phenotype. Do the behavioral phenotypes persist into the later life or are they transient to just this time window?

Reviewer #3 (Remarks to the Author):

The manuscript entitled "Pregnancy-induced maternal microchimerism shapes neurodevelopment and behavior" by Schepanski et al. described that subsets of maternal cells termed maternal microchimeric cells (MMc) contribute to fetal brain development by repressing fetal microglia cells and preventing excess synaptic pruning. Moreover, the authors observed that MMc ensured the establishment of the prefrontal-hippocampal circuit and normal learning and memory behaviors in offspring.

The observation is interesting and may significantly impact the understanding of brain development. However, there are some unignorable concerns about their data and interpretation.

Major points:

1. It is unclear why the authors ran mouse behavioral assay using only neonatal or (pre-)juvenile mice. The authors reasoned that MMc were not detected beyond P60. But if MMc played significant roles during early brain development, the behavioral alteration should sustain. How are the behavioral phenotypes in fully matured animals?
2. Gu's group showed that the mouse blood-brain barrier became functional at the embryonic day (E) 15.5 (Nature 509, 507-511, 2014), suggesting that MMc migration occurred before. Since fetal microglia cells also influenced the differentiation, proliferation, and migration of neural progenitor cells (NPCs) and newly differentiated neurons, why do the authors think that microglial defect is just synaptic pruning in MMc low mice that occurred much later? Do the authors observe any alteration in NPCs and newly differentiated neurons during earlier fetal brain development?
3. MMc were detected in different brain regions, including the cerebellum. Why only MMc in PFC and HP affect fetal microglia?
4. Is there any mechanistic understanding of how MMc suppress fetal microglia?
5. The authors defined that 5 clusters of MMc were microglia based on scRNA-seq. Does it mean that pregnant female mice shaded microglia in blood circulation? Or are these cells microglia-like cells? Also, their expression profile suggested that microglia-specific marker such as Tmem119 expression is low? How about Sall1 expression? Are these cells possibly border-associated macrophages (BAMs) or perivascular macrophages? If so, how about the expression of Pf4 and Lyve1 that are markers for BAMs?

Minor points:

1. Although lightsheet microscope imaging did not look like it, the Extended Figure 1f imaging looks like cells were in the tubular structure. Did the authors co-stain with a marker for the vasculature (e.g., CD31) and ensure that MMc were outside the vasculature?
2. It looks like MMc locate as clusters (by 2D and 3D images). Is there any discussion about it?
3. References are needed for Rab-7b is a negative regulator for inflammation. So far, the evidence is that Rab-7b is important for the degradation of TLR4.
4. Can the authors show the improvement of excess synaptic engulfment of microglia in vivo, such as sparse labeling of synapses? It is well known that microglia quickly changed the phenotype and gene expression.

5. A more detailed method is required for single-cell RNA-seq. For example, how many replicates did the authors use, and any batch effects were observed?

Point-by-point reply to reviewers' comments (reproduced verbatim) on revised manuscript entitled "Pregnancy-induced maternal microchimerism shapes neurodevelopment and behavior", NCOMMS-21-42047-T

Reviewer #1

General comment: This manuscript focuses on the role of maternal microchimerism on neurodevelopmental and behavioral changes in the offspring. The authors claim that maternal microchimeric cells (MMc) of different lineage in the fetal brain provide regulatory help in shaping normal neurodevelopment and behavior. The authors have performed some clever experiments and presented their data in an informative manner. This is a complex theme and the data presented need to have a critical analysis and interpretation. Several concerns remain unaddressed.

Authors' response to general comment: We are pleased to learn that Reviewer 1 generally appreciates our study and acknowledges our experimental approach and data presentation. We are grateful for his/her additional constructive comments, which helped us to improve our work, as outlined in the following.

Major comments:

#1. The authors are analyzing tdTomato-positive MMc at E18.5. Did the authors look at E14, E16, or even earlier for their presence? It is important because some of the analyses the authors have presented may relate to neurodevelopmental issues such as autism and schizophrenia later in the offspring. Although the authors show in Fig. 1b that there is no difference in MMc content in males vs. females, it is surprising as even at the placenta level, sexual dimorphism makes an impact on the onset of fetal brain development. MMc significantly decrease as the offspring ages. Is it because some lineage cells of MMc do not renew themselves or are eliminated due to lack of their growth factors in the brain microenvironment. In humans, MMc can be detected after 27 years of birth (Bianchi et al). Did the authors analyze MMc at P60 using the methods described in Fig. 1e and g?

Authors' response: In this comment, the reviewer raised several pivotal aspects. We have subdivided our response in three parts to give each aspect full credit.

Part I (Assessment of MMc at earlier gestational time points). We agree with the reviewer that kinetic analyses of MMc in fetal brain would be interesting, especially at critical neurodevelopmental time points. However, MMc transfer from mother to fetus during gestation is not only a physiological phenomenon, but has also been described as a continuous flow of cells which commences upon completion of placentation at gestational days 9.5/10.5. Therefore, in our present study, we focused on the analysis of MMc presence in the fetal brain at the last time point possible during gestation (E18) in wild-type mice as well as in mice with experimental MMc reduction. This allowed us to characterize MMc at the cellular and molecular level at the end of fetal development, when they could accumulate the longest. Additionally, this enabled us to assess the impact of MMc on fetal microglia phenotype and function prior to birth, hence, prior to the onset of early life environmental stimuli. However, we agree with Reviewer 1 that this approach precludes us from detecting possible fluctuations of MMc at various stages of neurodevelopment. From our perspective, such focus on distinct time points during neurodevelopment is highly relevant when assessing the impact of prenatal adverse events, which may occur at certain days of fetal development, interfere with neurodevelopment and also affect MMc phenotype and function. The integration of adverse events into the experimental setting of our present study would have been beyond its scope, as it was our aim to primarily identify the yet unknown physiological role of MMc. However, we anticipate that our study will now foster the analysis of adverse events on MMc at various gestational time points in future studies. We have revised the discussion as follows to highlight the need for such approach:

'The experimental approach we here chose was primarily geared towards the identification of the yet unknown physiological role of MMc on fetal brain development and later function. However, our focus on E18.5, the time point during fetal development closest to birth, precludes us from detecting possible fluctuations of MMc in fetal brain at various stages of neurodevelopment throughout gestation, once MMc transfer occurs upon completion of placentation. Based on the insights presented here on the relevance of MMc for offspring's neurodevelopment and behavior, a focus on distinct time points during neurodevelopment will likely be highly relevant when assessing the impact of prenatal adverse events. These events can occur at certain days of fetal development, are well known to interfere with neurodevelopment and hence, likely also interfere with MMc phenotype and function^{6,8,37}.'

Part II (No difference in MMc content in males vs. females). We agree with the reviewer that fetal brain development shows a high degree of sexual dimorphism, which is especially obvious in the context of adverse prenatal events. When identifying the number of MMc in brain in pregnancies unchallenged by adverse events, we did not observe significant differences between male and female fetuses in wild type mice. These observations suggest that the vertical transfer of MMc is not affected by the sexual dimorphism on the placental level.

When aiming to assess the consequences of MMc reduction in mice, we used offspring from immunodeficient $Rag2^{-/-}\gamma c^{-/-}$ C57BL/6 females, which had been mated to wild-type Balb/c males. Due to their immunodeficiency, these $Rag2^{-/-}\gamma c^{-/-}$ C57BL/6 females are only capable of transferring a very limited number of immune cells to their fetuses, which we could confirm by the low number of MMc in fetal brain. Therefore, we termed these offspring (which genotype is $Rag2^{+/-}\gamma c^{+/-}$) as MMc^{low}. *Vice versa*, mating wildtype C57BL/6 females to $Rag2^{-/-}\gamma c^{-/-}$ Balb/c males also yields to offspring with a $Rag2^{+/-}\gamma c^{+/-}$ genotype, but the female are fully immunocompetent and hence, transfer physiological levels of MMc to the fetal brain. We termed these offspring MMc⁺ (see also Fig. 2 a,b). However, the γc gene is encoded by the X-chromosome and hence, male offspring born to $Rag2^{-/-}\gamma c^{-/-}$ females are γc deficient, while male offspring born to wild-type females (termed MMc⁺) carry one copy of the γc gene. To control for this hemizyosity, we exclusively included female offspring in the respective experiments, which precludes us from the identification of a possible sexual dimorphisms this this MMc reduction model. To cover this aspect in the revised manuscript, we revised the text as follows:

In Results

'Noteworthy, since the γc gene is encoded by the X-chromosome, male offspring born to $Rag2^{-/-}IL-2\gamma c^{-/-}$ females are γc deficient, while male offspring born to wild-type females carry one copy of the γc gene. To control for this hemizyosity, only female offspring were included in the respective experiments.'

In Discussion

'Fetal brain development shows a high degree of sexual dimorphism, which is especially obvious in the context of adverse prenatal events^{38, 39}. Surprisingly, when identifying the number of MMc in brain in pregnancies unchallenged by adverse events, we did not observe significant differences between male and female fetuses in wild type mice, which suggests that the vertical transfer of MMc is not affected by the sexual dimorphism at the placental level. Due to hemizyosity of the γc gene in male MMc^{low} offspring, we excluded all male offspring from the assessments in the MMc^{pos}/MMc^{low} model. Future investigations will assess the possible sex-specific MMc effects in the MMc reduction model.'

Part III (MMc decrease with increasing offspring age). Similar to reports in other organs, we observed a decrease of MMc with increasing age, whilst MMc were still detectable at low numbers in offspring's brain at P60. In our present study, we did not aim to identify the mechanisms underlying the MMc decline over time. Clearly, the reviewer suggests pivotal pathways that may explain the observed MMc decline with increasing age, e.g., no potential for self-renewal, or elimination due to the absence of growth factors in the microenvironment of the offspring's brain. We have amended the discussion to cover these possibilities, as outlined below. Additionally, the reviewer queries whether we performed scRNA-Seq on MMc isolated on P60. We solely performed this elaborate (and costly) analyses in order to assess MMc on E18 (as shown in Fig. 1e, g). We agree with the reviewer that it would certainly be desirable to survey the gene expression in brain MMc throughout life. However, due to the significant decline in MMc numbers, this approach will likely have to wait until technology allows to isolate and assess cells at extremely low numbers at reasonable costs. Therefore, we prioritized to subject MMc isolated on P60 to flow cytometry-based analysis first, as this allowed us to evaluate overall

numbers along with MMc phenotypes. To address this point in the revised manuscript, we have revised the results and discussion section as follows:

In Discussion:

'Similar to reports in other organs^{17,42}, we observed a decrease of MMc with increasing age, whereby MMc were still detectable at low numbers in offspring's brain at P60. To date, insights on pathways supporting such longevity of MMc – including organ-specific longevity of MMc – as well as the mechanisms leading to the decline of MMc with increasing offspring's age are still mostly unknown. Possible pathways that may explain the observed MMc decline over time may include a limited potential for self-renewal, or the death of MMc due to cellular exhaustion or absence of growth factors in the organ-specific microenvironment, e.g. the offspring's brain. Remarkably, MMc longevity may also be explained by the different MMc phenotypes that can be detected in offspring's fetal and adult organs. E.g., in mice, a large number of MMc in bone marrow are T cells¹⁷, whereas we here show that the largest MMc population in the brain are microglia. These observations suggest either a preferential recruitment of these MMc subsets to the different fetal organs, or a disparate differentiation of progenitor-like MMc, dependent of the tissue microenvironment in which they seeded.'

'Another limitation is the monitoring of MMc by scRNA-seq solely on E18.5. Clearly, it would have been desirable to survey the gene expression in brain MMc throughout life. However, technologies enabling to isolate and assess cells at extremely low numbers at reasonable costs are still missing.'

#2. The authors state that MMc can be found in different organs. Is there any information available about their lifespan in these organs? Or does the brain support a different lifespan of MMc?

Authors' response: Based on the issue raised here, we felt that we needed to amend the introduction by some state-of-the art details and have revised the manuscripts as follows:

In Introduction

'The transfer of MMc commences with maturing placentation, hence, with the onset of the second trimester in humans and around mid-gestation in mice⁹. Remarkably, MMc are not rejected by the fetal immune system¹⁴. In fact, the genetically discordant MMc can even show a long-term persistence in offspring's organs until adulthood¹⁴⁻¹⁶. During fetal development, MMc seed into a number of fetal organs, including primary and secondary immune organs as well as non-immune organs^{17,18}. MMc have also been detected in the offspring's brain^{9,19}, yet their phenotype, location and impact on brain-resident immune and non-immune cells in the fetus and brain function is still unknown.'

Furthermore, the reviewer's query pertaining to the lifespan of MMc in various organs, or possible differential organ (brain)-specific support of MMc lifespans, are in line with Part III of point #1, which is why we had included the details here marked in **bold** in the newly inserted text in the revised discussion. We apologize for the lengthy repeat of these amendments, but feel that this is needed for clarity.

From #1, Part III 'To date, insights on pathways supporting such longevity of MMc – **including organ-specific longevity of MMc** – as well as the mechanisms leading to the decline of MMc with increasing offspring's age are still mostly unknown. Possible pathways that may explain the observed MMc decline over time may include a limited potential for self-renewal, or the death of MMc due to cellular exhaustion or absence of growth factors in the **organ-specific microenvironment**, e.g. the offspring's brain. **Remarkable, MMc longevity may also be explained by the different MMc phenotypes that can be detected in offspring's fetal and adult organs. E.g., in mice, a large number of MMc in bone marrow are T cells¹⁷, whereas we here show that the largest MMc population in the brain are microglia. These observations suggest either a preferential recruitment of these MMc subsets to the different fetal organs, or a disparate differentiation of progenitor-like MMc, dependent of the tissue microenvironment in which they seeded. This may subsequently also affect the lifespan of MMc in the different offspring's organs.**

#3. Pregnancy can be divided into three phases: 1. Implantation (inflammation), gestation (anti-inflammation), and parturition (inflammation). Since E18.5 is very close to the onset of parturition, is inflammation, lack of pregnancy hormones, or infection expected to impact the transport, and the content and/or function of MMc?

Authors' response: The reviewer raises an important point, and we seem to have neglected to address this aspect appropriately in our original submission. We now amended the discussion as follows in order to highlight the importance of the immune trajectory during pregnancy:

'The course of pregnancy can be divided into immunologically distinct stages, including a brief inflammatory surge around the time of blastocysts implantation, followed by the long gestational period of anti-inflammation and immune tolerance to ensure that the fetus is not rejected. Parturition is then initiated by progesterone withdrawal and inflammation⁵⁵. Especially the inflammatory period related to the onset of parturition may affect the transfer of MMc from mother to fetus. Moreover, adverse events occurring during the period of anti-inflammation during pregnancy in mice and humans, e.g., infection or related proxy, as well as trauma, skew maternal cells towards a pro-inflammatory phenotype. This has been shown to enhance the transfer rate of MMc from mother to fetus⁵⁶⁻⁵⁹ and may possibly alter the function of MMc in various fetal organs. The here presorted data on the functional role of MMc during normally progressing pregnancies will enable to address the inflammation-induced alterations of MMc upon adverse prenatal events and the related consequences for offspring's brain and other offspring's organ development and function.'

#4. The authors have focused a lot on brain homeostasis and dismantling of synaptic connections. What does exactly brain homeostasis mean? In Fig. 2 and related Extended Figures, the authors have invoked DNA methylation mechanisms, fetal T and B cell distribution, and transcriptome choreography. It is suggested that genes such as *Rab-7b* were down-regulated in microglia of MMc low offspring. Are there scenarios where the brain is mainly populated with MMc low? Does this result in brain disorders? Is transcriptome profile different in Wt male vs. Wt female?

Author's response: We feel that the description of our findings may have not been comprehensive, otherwise Reviewer 1 would likely not have asked if there are scenarios where the brain is populated with low numbers of MMc or whether we detected sex-specific effects. As shown in Figure 2 and following, we had integrated a model of low MMc in our experimental design. In order to increase comprehension of our data, we have now inserted a new subheading in order to introduce this model of low MMc first, before we describe our findings in fetal brain using this model. This new section reads as follows:

'Mouse model of experimental reduction of MMc in fetal brain

'... , we used a mouse model in which MMc in offspring's brain were experimentally reduced in order to gain insights into the functional role of MMc in the offspring's brain. This reduction of MMc was achieved by reciprocal mating of *Rag2^{-/-}IL-2ryc^{-/-}* female or male mice with wild type (wt) mice. *Rag2^{-/-}IL-2ryc^{-/-}* mice are immunodeficient and lack T, B and – to a lesser extent – innate lymphoid cells³⁰. The offspring of these reciprocal mating combinations all expressed a *Rag2^{+/-}IL-2ryc^{+/-}* genotype (Fig. 2a). Noteworthy, since the *yc* gene is encoded by the X-chromosome, male offspring born to *Rag2^{-/-}IL-2ryc^{-/-}* females are *yc* deficient, while male offspring born to wild-type females carry one copy of the *yc* gene. To control for this hemizyosity, only female offspring were included in the respective experiments.'

'... , we analyzed MMc numbers in offspring's brain at E18.5 and P8. Offspring born to *Rag2^{-/-}IL-2ryc^{-/-}* dams harbored significantly fewer MMc in the brain and were termed 'MMc^{low}', compared to offspring from wt dams (termed 'MMc^{pos}') (Fig. 2b-d).'

In fact, in the MMc low offspring, genes suppressing inflammation, e.g., *Rab-7b*, were down-regulated in microglia, which is mentioned later in the results section as flows:

In Results:

'Here, genes suppressing inflammation, such as ras-related protein (*Rab-7b*), responsible for suppressing tumor necrosis factor (*Tnf*), interleukin-6 (*Il-6*), and interferon β (*Inf- β*) production in macrophages³¹ were down-regulated in microglia from MMc^{low} offspring (Fig. 2h).'

As for possible sex-specific effects, we wish to highlight again that these are not WT offspring, but *Rag2^{+/-} IL-2ryc^{+/-}* offspring. Hence, we could only focus on female offspring (kindly see our reply to #1, part II).

#5. Figures 2 and 3 describe a solid experimental plan to rule out the contribution of immune cells in MMc-mediated brain development. The authors used allogeneic mating protocol involving Wt C57BL/6 female and *Rag2IL-2 ryc* deficient male mice. In another mating, they reversed the mating partners and analyzed the MMc content. They identified two types of neonates – MMcPos and MMc low. What dictates this very distinguished difference? Why some offspring become MMclow? Do MMclow offspring entail poor brain homeostasis? In Fig. 3, the authors employ ultrasonic vocalization as one of the tools to differentiate between MMcPos

and MMclow offspring. Although the authors claim significant differences, the data may not support this claim strongly. Yes, there are clear-cut differences in discrimination profiles; the number MMclow offspring used in the experiments is lower. A careful look at the ultrasonic vocalization suggest that MMc low offspring experience the same vocalization as that described autistic children, particularly male offspring. Do authors have any comment on this?

Author's response: We also subdivided our response to this point into two parts and first address the query related to MMc^{pos}/MMc^{low} animal model.

Part I (Animal model): This comment also indicates that the description of our experimental approach, especially the MMc^{low} vs MMc⁺ model has been too superficial and may leave the reader confused. As outlined above under #4, we have significantly amended the description of our experimental design. Here, we now explain in greater detail why some offspring become MMc^{low}. This facilitates comprehension of the distinguished differences between the Rag2^{+/-} IL-2r γ ^{+/-} offspring, which have either been born from Rag2^{+/+} IL-2r γ ^{+/+} (termed MMc^{low}) or wt mothers (termed MMc⁺). In fact, the comparison between MMc^{low} and MMc⁺ allowed us to identify the hyperactivation and altered function of fetal microglia in MMc^{low} offspring, resulting in enhanced disruption of brain homeostasis, and dismantling of synaptic connections by phagocytosis of presynaptic vesicles. In the context of the present revision, we carefully went over the description of our specific findings in order to ensure rapid comprehension. We refrain from copying the entire section in this reply and kindly refer to the results section of the revised manuscript with the subheading 'MMc maintain fetal microglia homeostasis and suppress excessive presynaptic elimination'.

Part II (Ultrasonic vocalization): Here, Reviewer 1 suggests that our data may not support strong differences. The reviewer correctly noted that the overall number of MMc^{low} pups is lower compared to the number of MMc⁺ pups that were available for the vocalization analysis (8 vs 11). Irrespective of these differences in group size, we observed significant differences with regard to the quality of the vocalizations between groups. As shown in **Fig. 4b**, simple calls were lower in the MMc^{low} offspring, whilst the frequency jumps and the complex calls are higher.

Furthermore, we were very intrigued about the cross-reference made by the reviewer with regard to the similarities between MMc^{low} offspring and autistic children, in which higher fundamental frequencies were noted (Esposito and Venuti, 2010) – similar to the cries we recorded in the MMc^{low} pups. In order to convey the importance of our observations in mice more clearly, along with the intriguing link to autism in humans, we amended the manuscript as follows:

In Results:

'We observed significant differences with regard to the quality of the vocalizations between groups, as the length of simple calls was lower in the MMc^{low} offspring, whereas the frequency jumps and the complex calls lasted longer (**Fig. 4a-b, Extended Data Fig. 10a-c**). These behavioral features might indicate emotional distress and disruption of social communication between mother and MMc^{low} pups. Intriguingly, a similar cry pattern has been observed in autistic children³⁶.'

#6. Adoptive transfer of MMc in MMclow offspring seems to restore microglial engulfment (?). How many MMc needed to be transferred? Is there a kinetic threshold of adoptive transfer? What happens to the offspring at P60 after adoptive transfer?

Author's response: We adoptively transferred 1×10^7 immune cells into the immunodeficient Rag2^{-/-}IL-2r γ ^{-/-} pregnant mice on E12.5. The cell suspension used for the transfer has been obtained by harvesting leukocytes from blood, lymph nodes and spleen of pregnant wild-type mice. The phenotypic assessment of the isolated cells revealed a frequency of approx. 60% B, 30% T, and 10% myeloid cells, as well as low frequencies of NK cells, macrophages, lymphoid, and plasmacytoid dendritic cells, as depicted in **Extended Data Fig. 11c**. Noteworthy, 1×10^7 immune cells is the maximum cell number allowed by our institutional ethical guidelines

to be injected, as higher numbers may cause thromboembolism. Since this number was just sufficient to restore the MMc in fetal brain to the levels seen in MMc⁺ offspring, we refrained from testing fewer cells or evaluating kinetic thresholds.

We did not include the day P60 in our experimental design, as our observations in wild type mice had informed us that MMc in brain have significantly waned at this offspring's age. Therefore, the experimental burden of the intervention for the mice would not have been justified by the outcome, as this would have likely confirmed MMc numbers close to detection limit.

In order to convey the findings in the MMc^{low+AT} offspring born from the *Rag2^{-/-}IL-2 γ c^{-/-}* pregnant mice upon adoptive transfer, we reworded this section in the results section as follows:

'In the MMc^{low+AT} offspring, AT restored the absolute number of MMc in fetal and neonatal brain (**Fig. 5b-d**) and MMc subset populations on E18 and P8 were similarly distributed as observed in MMc^{pos} offspring (**Extended Data Fig. 11d-e**). Moreover, the number of fetal and neonatal microglia was restored in MMc^{low+AT} offspring to the frequencies observed in MMc^{pos} offspring (**Fig. 5e-f**). Similarly, in the MMc^{low+AT} offspring, the enhanced presynaptic terminal elimination detected in MMc^{low} offspring was restored to levels comparable to those seen in MMc^{pos} offspring (**Fig. 5g-h**).'

Minor comments:

#1. On line 118, it should be Fig. 2b-c not Fig. 3b-c. Also, on the same page line 132, it should be "to a lesser extent".

Author's response: We corrected the text accordingly.

#2. Although the authors discuss why fetal microglia increase when MMc are low, it is not entirely clear why MMc become low and affect only microglia. Otherwise, Discussion is well written.

Author's response: In our response to point #4, we have explained why MMc become low in the experimental model we developed and revised the manuscript accordingly. Here, the reviewer additionally queries why only microglia are affected by the decrease of MMc. We respectfully disagree with the statement reducing our findings to 'only' microglia changes. Besides the overall behavioral changes we observed, we also screened for additional changes of fetal brain immune cells and observed reduced numbers of T and B cells in fetal brain (**Extended Data Fig. 6h-f**). Given the importance of microglia, we focused on the consequences of the altered fetal microglia number, phenotype and function, e.g. in their interaction with neurons. Admittedly, we did not follow up on the consequences related to the decrease of T and B cells. In order to address the reviewer's point, we included this now in the discussion as follows:

'Given the importance of microglia for brain wiring, the present study focuses on the identification and functional consequences of the altered fetal microglia number seen in offspring with reduced or restored MMc. The observed decrease of T and B cells in brain of fetal MMc^{low} offspring will be subject of future investigations, especially taking into account the functional role of T cells in autoimmune diseases affecting the brain.'

Reviewer #2

Reviewer #2 (Remarks to the Author):

This is a very interesting manuscript designed to determine the impact of maternal microchimeric (MMc) cells in fetal mouse brain on synaptic development, circuit function, and behavior. The authors report a significant number of the MMc within the fetal brain are microglia with high expression of homeostatic genes, and that these cells primarily localize to the PFC and HP. In the absence (or reduction) of MMc, there are more microglia in the fetal brain and they engulf more presynaptic markers. Moreover, offspring exhibit altered communication and cognitive behaviors and changes in network activity. Finally, adoptive transfer of leukocytes back to immunodeficient dams largely rescues these phenotypes. Overall, there is a large amount of compelling data and the rigor and importance of the work seems high. There are some concerns, which are outlined here.

Author's response to general comment: We are pleased to learn that the reviewer generally appreciates our work and describes our data as compelling.

#1. Fig. 1 – panel f is confusing. It looks like according to the color-coded dots for different cell types that endothelial cells also express high levels of the canonical microglial markers *Cx3cr1*, *P2yr12*, and *Sparc*? Similarly, for panel h, it says the heatmap of gene expression is based on genes that were more than 50% higher in MMc compared to fetal microglia, but the heatmap also shows genes with decreased fold change? Finally, for panel g, these gene categories are subjective and context-dependent. The genes included in each category should be provided rather than assigning them to categories absent a functional readout. For instance, *tmem119* is often considered a “homeostatic” gene in microglia but its expression is not particularly high in the microglia clusters in panel f.

Authors' response: We have to apologize for an oversight that has resulted into the wrong insertion of genes in the graph shown in Fig. 1f, which has been noticed by the reviewer (and unfortunately not by us prior to submission!). The faulty graph indicated the expression of genes (*Cx3cr1*, *P2yr12*, *Sparc*, *Tmem119*), which are not regularly expressed in the non-microglia cells we evaluated (T, B, endothelial, and neuron-like cells). Our systems biologists repeated the analysis pipeline, which resulted in the corrected and now inserted graph 1f, in which *Cx3cr1*, *P2yr12*, *Sparc* and *Tmem119* are no longer upregulated in the non-microglia cells.

Next, the reviewer commented that the gene categories we had used in graph 1g are based on subjective and context-dependent criteria. We agree with the reviewer that such categorization of genes underlies a certain degree of subjectivity. On the other hand, such categorization is a common approach used in many scientific publications in order to facilitate comprehension of the complex findings resulting from scRNA-Seq. However, since it was our primary intention to characterize MMc on a cellular and molecular level - as shown in **Fig. 1e-f, Extended Data 2a-b** - we decided to omit the somewhat redundant graph 1g (and related graph 1h), in which we had introduced the gene categories.

#2. Fig. 2 - Determination of cell number using flow cytometry is not very reliable. There are too many variables regarding cell loss with the isolation and method of gating. Quantitative claims should be shown in the intact brain using IHC or similar.

Authors' response: It appears that two schools of thought may be colliding here, whereas the reviewer seems to favor immunohistological (IHC)/in situ approaches, and our approach is largely based on flow cytometry. Therefore, we respectfully disagree with Reviewer 2's statement that determination of cell number is not very reliable when using flow cytometry as a quantitative approach. To justify and strengthen our approach, we wish to summarize the experimental methods we implemented in order to exclude technical limitations related to flow cytometry, e.g., the loss of cells when isolating and gating brain cell, as mentioned by the reviewer. First, we transcardially perfused the mice prior to sacrificing in order to exclude that

cells isolated from brain are contaminated by cells from the peripheral vasculature supplying the brain. The risk of cell loss when isolating cells from the brain was reduced by our established flow cytometry protocol, which includes i) careful handling of cell pellets when washing the cells, ii) no enzymatic digestion of the brain tissue, as this would have affected cell numbers and surface marker expression. In fact, our protocol allows to isolated brain cell by careful manual homogenizing of the organ through a cell mesh. Taken together, this protocol is well-established and widely used internationally and has resulted in important insights on microglia function and development (Hammond et al., 2019; Kim et al., 2021; Marsh et al., 2022; Matcovitch-Natan et al., 2016). Similarly, the method of gating we (and many others) chose to identify distinct cell subsets in the cell suspensions isolated from the brain is unambiguous due to the well-balanced number of antibodies and conjugated fluorochromes (Filipello et al., 2018; Hammond et al., 2019; Masuda et al., 2019). Our protocol also ensures that the gated cells are indeed the cells of interest by including the so-called FMO control (fluorescence minus one), in which we sequentially omit one fluorochrome-conjugated antibody from the complete antibody panel in order to create a negative control for this specific marker. The high quality of our experimental approach with regard to cell isolation and gating is also reflected by the similar number of parent populations we detected in the respective experimental groups. This quantification step underpins that analysis within a similar parent population yields to sound cellular data in the brain. Second, the quantification of MMC in fetal brain requires the detection of four cell surface markers alone (CD45.2⁺, CD45.1^{neg}, H-2D^{b/lb} +, H-2D^{d/lb} neg), plus additional markers to identify the phenotype of MMC. Here, we additionally included CD11b for the identification of microglia, CD3 for T cells, B220 for B cells and others. In total, 14 markers are needed to unambiguously quantify MMC in fetal brain. Therefore, a histological approach using 14 cells surface markers (antibodies) simultaneously would have been associated with significant technical limitations. Nonetheless, the reviewer is certainly correct in favoring IHC-based approaches, as is enables to localize cells in situ. This was also our goal in the context of MMC detection in the fetal brain, which is why we had included the mouse model in which tdTomato^{+/-} females mated to wild-type males were used and order to generate litter with 50% of tdTomato^{+/-} and 50% of tdTomato^{-/-} offspring. The tdTomato^{-/-} offspring then enable localization of tdTomato-positive cells in the brain, which can only be maternally derived and hence, MMC. These data are shown in Suppl. Fig. 1f and – following the CUBIC clearing of fetal brain - in Movie 1.

We hope that our explanations have convinced Reviewer 2 that the determination of cell number using flow cytometry in our manuscript was sound and has yielded to reliable data.

#3. Why was Vglut1 assessed for pruning by microglia? These label short-range projections (e.g. intra-cortical) at P8, in contrast to Vglut2, which labels long-range projections. It is surprising given the claim that these synaptic changes underlie the circuit deficits in the mice, as the relevant long-range projection synapses were not assessed. It is therefore less convincing that these synaptic changes are linked to the network activity changes reported.

Authors' response: We agree with the reviewer that the assessment of microglia pruning solely based on Vglut1 expression had its limitation, as the Vglut1 staining only labels short-range projection. In the context of the revision, we performed additional experiments and now included data showing the Vglut2-based long-range projections. Here, similar to Vglut1, we also observed a higher Vglut2 engulfment by microglia from MMC^{low} compared to MMC^{pos} offspring. However, opposed to the observations with regard to Vglut1, this increase did not reach levels of significance for Vglut2 ($P = 0.07$).

We amended the results section of the revised manuscript as follows:

'In order to provide evidence for this notion, we investigated microglia engulfment of synaptic terminals in the pre-limbic subdivision (PL) of PFC and HP, since these areas are the core of neuronal networks accounting for complex cognitive abilities, such as memory, learning, and flexibility³⁴. The engulfment of terminals from short-range projections stained by Vglut1⁺ and from long-range projections stained by Vglut2⁺ augmented in MMC^{low} offspring *ex vivo*, yet the increase reached significance level only for Vglut1⁺ (Fig. 2i-l, Extended Data Fig. 7a-n).'

In Methods/Imaging and image analysis:

'For Iba-1, Vglut1, and Vglut2 positive cell expression, microscopic stacks were acquired as 1024x1024 pixels images with 750nm Z-steps capturing 8 microglial cells within the mPFC and CA1 region of the HP, using a 63X objective.'

In Methods/Immunohistochemistry:

'Subsequently, slices were processed by incubating them overnight at 4°C with anti-Iba-1 (1:500, Wako Pure Chemical, Cat. No. 019-19741), anti-Vglut1 (1:1000, Millipore, Cat. No. AB5905), and anti-Vglut2 (1:500, Synaptic Systems, Cat. No. 135404), followed by 1 h incubation with goat-anti-guinea pig (1:500, AF488, Invitrogen, Cat. No. A-11073), donkey-anti-rabbit (1:500, AF568, Invitrogen, Cat. No. A-10042) secondary antibodies and Hoechst33258 (1:5000, Sigma, Cat. No. 94403).'

#4. For fig 3d and f, it actually appears that the controls (MMc^{pos}) mice show no discrimination as they are roughly at chance, whereas the MMc^{low} do, and avoid the novel object. This is not a cognitive deficit but perhaps novelty avoidance. In any case, they show better discrimination if I am interpreting this correctly. More concerning is the fact that the control group does not show normal discrimination so the data are difficult to interpret.

Authors' response: We agree with the reviewer that the behavioral performance of the control offspring (MMc^{pos}) is roughly at chance, whereas the behavioral performance of the MMc^{low} offspring is well below the threshold. We have made similar observations in previous studies, where the control animals also did not all perform above chance (Chini et al., 2020), whilst overall better compared to the data presented here. The poor behavioral performance observed in our present study may be explained by the heterozygous genetic background of the MMc^{pos} and MMc^{low} offspring. These offspring resulted from the allogenic mating combination of two different mouse strains, C57BL/6 females and Balb/c males, hence have a mixed strain background. Published evidence showed that Balb/c and C57BL/6 mice perform differently in behavioral experiments, for instance Balb/c mice are more anxious and less explorative (Depino and Gross, 2007; Garcia and Esquivel, 2018). In addition to the mixed C57BL/6xBalb/c strain background of the MMc^{pos} and MMc^{low} offspring, these offspring are also all heterozygous for the *Rag2/IL-2ryc* genes. It was reported that the genetic manipulation of *IL-2ryc* also results in impaired behavioral competence (Petitto et al., 1999), which may further explain the overall poor performance we observed. We here directly compare offspring that are all of mixed strain background and heterozygous for the *Rag2/IL-2ryc* genes and thus feel, that the significant reduction of behavioral performance in MMc^{low} compared to MMc^{pos} animals is a valid observation. Clearly, the option to extrapolate or compare the overall performance we observe in our unique experimental setting to other studies where e.g., pure strains without transgenic manipulation were used, is very limited. To highlight this aspect in the revised manuscript, we specified the text as follows:

In Discussion:

'Of note, the performance of pre-juvenile MMc^{pos} offspring in recognition memory tasks was often below chance level. This may be explained by the mixed strain background of the MMc^{pos}/MMc^{low} offspring, which resulted from the allogenic mating combination of C57BL/6 females and Balb/c males. In fact, Balb/c mice have been shown to be more anxious and less explorative^{50,51}. Additionally, the MMc^{pos} and MMc^{low} offspring are also heterozygous for the *Rag2/IL-2ryc* genes. The genetic manipulation of *IL-2ryc* may result in an impaired behavioral competence⁵². However, these deficits of individual groups do not bias the robust differences observed between MMc^{low} and MMc^{pos} animals, since they all shared mixed strain background and are heterozygous for the *Rag2/IL-2ryc* genes.'

#5. The authors make no comment on the mechanism by which immune cell deficiency in dams impacts microglial number and/or function. Are there changes in the yolk sac progenitors or only changes once they arrive into the CNS?

Authors' response: In order to address this question, we performed additional experiments in which we isolated the individual yolk sacs from female MMc^{pos} and MMc^{low} offspring and subsequently quantified the leukocytes, erythromyeloid progenitor cells (EMPs) and pre-macrophages at E9.5 by means of flow cytometry. Since the total number of EMP (A) and pre-

macrophages (B) is very low, differences between groups of offspring do not seem to be reliable. However, the reviewer raised an intriguing notion, which we would like to convey to a future readership. We therefore amended the discussion as follows:

'Tissue-resident macrophages like microglia in brain originate from erythromyeloid progenitors (EMP), which develop in the yolk sac, then migrate and seed into the fetal liver and subsequently colonize embryonic organs as EMP-derived macrophages. In the brain, these progenitors complete their differentiation into microglia, which are self-renewing throughout life and are only minimally replenished by circulating macrophages^{43,44}. In our study, the number of progenitor cells did not differ in yolk sac of MMc^{pos} and MMc^{low} offspring at E9.5 (**Extended Data Fig. 12a-b**). However, since the overall number of such progenitor cells was extremely low, the biological significance of these observations may be limited. Future studies should aim at in-depth investigations of the microglia progenitor cells and

their development in presence and absence of MMc infiltration in order to determine their potential interaction already in the yolk sac.'

#6. Why were cognitive behaviors tested at P19-24 in offspring? This is interesting because it is the time in which the hippocampal circuitry is just maturing, and any delay or alteration in this maturation could impact the behavioral phenotype. Do the behavioral phenotypes persist into the later life or are they transient to just this time window?

Authors' response: The reviewer raises an important question. The cognitive behavior has been tested at the time point when the abilities emerge. For this age, we developed the optimally fitting experimental paradigms and acquired a large data set for mice and rats (Chini et al., 2020; Kruger et al., 2012; Xu et al., 2021). The relationship between maturational dynamics and cognitive behavior is of high interest. Studies in rodents showed that specific developmental time windows are of particular relevance for adult behavioral performance. For example, we previously identified the beginning of the second postnatal week as critical for cognitive abilities of adults, since manipulation of neuronal activity at this neonatal age caused memory deficits later in life (Bitzenhofer et al., 2021). On the other hand, environmental triggers or activity manipulation may cause deficits that reverse along postnatal life, become milder in their characteristics or even entirely disappear. For instance, individuals with autism spectrum disorder have been closely studied for their autistic traits in several longitudinal and retrospective studies (Billstedt et al., 2005; Eaves and Ho, 2008; Farley et al., 2009; Helles et al., 2015; Lord et al., 2015; Magiati et al., 2014; Seltzer et al., 2004), showing clear improvements in cognitive and behavioral symptoms over time. The experiments necessary to properly address question #6 are complex and exceed the framework of this already very dense study. They will be the core of a future investigation focusing on the long-term effects. In the present manuscript, we addressed the reviewer's concern by specifying:

In Discussion:

'In the present study, we focused on the emergence of cognitive abilities along neonatal and pre-juvenile development and did not extend the investigation of behavioral phenotype until adult age. The long-term effects of reduced number of MMc might be either milder and (partially) compensated or persistent, leading to life-long deficits. We recently identified critical time windows of cognitive development during which transient manipulation of electrical activity causes permanent reduction of network function and behavioral performance in memory tasks⁵³. Similar processes may occur also in MMc^{low} mice. The number of retained MMc declines with age, although low numbers are still detectable in mature offspring.'

Reviewer #3

Reviewer #3 (Remarks to the Author):

The manuscript entitled "Pregnancy-induced maternal microchimerism shapes neurodevelopment and behavior" by Schepanski et al. described that subsets of maternal cells termed maternal microchimeric cells (MMc) contribute to fetal brain development by repressing fetal microglia cells and preventing excess synaptic pruning. Moreover, the authors observed that MMc ensured the establishment of the prefrontal-hippocampal circuit and normal learning and memory behaviors in offspring. The observation is interesting and may significantly impact the understanding of brain development. However, there are some unignorable concerns about their data and interpretation.

Major points:

#1. It is unclear why the authors ran mouse behavioral assay using only neonatal or (pre-) juvenile mice. The authors reasoned that MMc were not detected beyond P60. But if MMc played significant roles during early brain development, the behavioral alteration should sustain. How are the behavioral phenotypes in fully matured animals?

Authors' response: As specified to query #6 of Reviewer #2, the impact of MMc on cognitive performance along the entire development is a highly important aspect, yet the in-depth experimental investigation exceeds the framework and main aims of the present study. The cognitive behavior has been tested at the time point when the abilities emerge. For this age, we developed the optimally fitting experimental paradigms and acquired a large data set for mice and rats (Chini et al., 2020; Kruger et al., 2012; Xu et al., 2021). The relationship between maturational dynamics and cognitive behavior is of high interest. Studies in rodents showed that specific developmental time windows are of particular relevance for adult behavioral performance. For example, we previously identified the beginning of the second postnatal week as critical for cognitive abilities of adults, since manipulation of neuronal activity at this neonatal age caused memory deficits later in life (Bitzenhofer et al., 2021). On the other hand, environmental triggers or activity manipulation may cause deficits that reverse along postnatal life, become milder in their characteristics or even entirely disappear. Since most structural and functional investigations focused on early and midterm development, we monitored the behavioral correlates during similar time windows. Since neonatal mice, e.g. at P8, are not yet able to perform complex tasks, we focused on basic behavioral experiments such as recording their vocalizations upon separation from the mother to assess the short-term consequences of reduced numbers of MMc during fetal development. Here, we observed significant differences with regard to the quality of the vocalizations between MMc^{pos} and MMc^{low} offspring. The medium-term consequences of reduced numbers of MMc during fetal development were assessed at pre-juvenile age (between P19-24). At this age, more complex behavioral patterns emerge. We observed significantly shorter interaction time and fewer interactions with novel or less-recent objects in MMc^{low} offspring. The number of retained MMc declines with age, although low numbers are still detectable in mature offspring (P60). The experiments necessary to uncover the long-term impact of MMc on adult cognitive behavior (long- and short-term memory, attention, decision-making and working-memory) will be the core of a future study. In the revised manuscript we specified the focus on short- and mid-term effects as follows:

'Next, we tested whether the dysfunction of PFC-HP circuits in neonatal MMc^{low} mice leads to behavioral deficits already at this early developmental stage.'

'Second, we monitored the emergence of cognitive abilities requiring prefrontal-hippocampal communication that can be tested starting from the second-third postnatal week.'

'In the present study, we focused on the emergence of cognitive abilities along neonatal and pre-juvenile development and did not extend the investigation of behavioral phenotype until adult age. The long-term effects of reduced number of MMc might be either milder and (partially) compensated or persistent, leading to life-long deficits. We recently identified critical time windows of cognitive development during which transient manipulation of electrical activity causes permanent reduction of network function and behavioral performance in memory tasks⁵³. Similar processes may occur also in MMc^{low} mice. The number of retained MMc declines with age, although low numbers are still detectable in mature offspring.'

#2. Gu's group showed that the mouse blood-brain barrier became functional at the embryonic day (E) 15.5 (Nature 509, 507-511, 2014), suggesting that MMc migration occurred before. Since fetal microglia cells also influenced the differentiation, proliferation, and migration of neural progenitor cells (NPCs) and newly differentiated neurons, why do the authors think that microglial defect is just synaptic pruning in MMc low mice that occurred much later? Do the authors observe any alteration in NPCs and newly differentiated neurons during earlier fetal brain development?

Authors' response: We share the reviewer's appreciation for the work by Gu's group (Ben-Zvi et al., 2014) on the functional establishment of the blood-brain barrier at the embryonic day 15.5 in mice. However, from our point of view, this functional establishment of the blood-brain barrier does not necessarily exclude continuous MMc migration across the blood-brain barrier. In fact, published evidence reveals that breast milk-derived MMc can be detected in the offspring's brain (Aydin et al., 2018), supporting an MMc migration after blood-brain barrier establishment.

As stated by the reviewer, fetal microglia can influence a wealth of processes in the developing brain, e.g. differentiation, proliferation, and migration of neural progenitor cells as well as the refinement of synaptic connectivity. We here analyzed their influence on refining synaptic connectivity and neuronal branching, because synaptic pruning is the result of a variety of neuronal molecular interactions, such target recognition, as well as phagocytosis, and has been shown to be an essential step in the cascade of impairing neurodevelopment (Faust et al., 2021; Neniskyte and Gross, 2017). In the present study, we did not include assessments of different neuronal types, specific receptor functions, and dendritic arborization earlier during fetal brain development, as it was our primary focus to assess the effect of MMc of fetal microglia towards the end of fetal development, when they could accumulate the longest. In addition, this allowed for the investigation of the influence of MMc on fetal microglia almost entirely devoid of (early) postnatal environmental stimuli, which have been shown to alter microglia function dramatically (Hanamsagar and Bilbo, 2017). To convey our rationale more clearly, we have revised the discussion as follows:

'Since the fetal blood-brain barrier becomes functional during fetal development at E15.5⁴⁰, one may assume that MMc migration into the fetal brain discontinues as of the milestone. Interestingly, published evidence reveals that MMc derived from breast milk can also be detected in the offspring's brain⁴¹, which strongly supports a continuous MMc migration upon blood-brain barrier establishment.'

'The microglia engulfment of pre-synaptic terminals was used as readout of diverse neuronal interactions that control the development of circuits.'

#3. MMc were detected in different brain regions, including the cerebellum. Why only MMc in PFC and HP affect fetal microglia?

Authors' response: As noticed by the reviewer, the initial screening assays led to the detection of MMc in different brain regions, including the cerebellum. In the subsequent functional analyses, we focused on the consequences of MMc in PFC and HP, since these areas are the core of the limbic circuit accounting for cognitive behavior (learning, memory, flexibility). Correspondingly, structural and function alterations of PFC-HP communication have been detected as substrate of cognitive impairment in several neurological and neuropsychiatric disorders (Böhner et al., 2015; Herweg et al., 2016; Milad et al., 2007; O'Neill et al., 2013; Spellman et al., 2015). To mirror these aspects, we modified the text:

In Results:

'In order to provide evidence for this notion, we investigated microglia engulfment of synaptic terminals in the pre-limbic subdivision (PL) of PFC and HP, since these areas are the core of neuronal networks accounting for complex cognitive abilities, such as memory, learning, and flexibility³⁴. The engulfment of terminals from short-range projections stained by Vglut1⁺ and from long-range projections stained by Vglut2⁺ augmented in MMc^{low} offspring *ex vivo*, yet the increase reached significance level only for Vglut1⁺ (Fig. 2i-k, Extended Data Fig. 7a-n).'

In Discussion:

'In the present study, we were able to detect MMc in different brain regions, including the cerebellum. The in-depth functional investigation focused only on MMc in PFC and HP due to the role of these areas for cognitive processing in health and abnormal memory and cognitive flexibility in neurological and neuropsychiatric disorders⁴⁵⁻⁴⁹.'

#4. Is there any mechanistic understanding of how MMc suppress fetal microglia?

Authors' response: The reviewer raises a valid question about the interaction of MMc and fetal microglia in the developing brain, as our results strongly support an MMc-dependent suppression of fetal microglia activation and related synaptic pruning. From our point of view, the outcome of our scRNASeq analyses provides a number of pivotal hints to understand how the interaction between MMc and fetal microglia may be operational. Here, we identified an upregulation of sensome genes in MMc, including *Cd47*, *Selplg*, *Cd37* and *Il-6ra*, along with a downregulation of inflammatory genes (data are shown in Fig. 1). In fact, it has been shown that the microglia sensome conveys neuroprotection and is involved in host defense (Hickman et al., 2013). More specifically, CD47 protects synapses from excess microglia-mediated pruning during development (Lehrman et al., 2018). This provide an explanation for the observation we made in MMc^{low} offspring, where the reduction of MMc and hence, *Cd47*, was linked to an increased microglia-dependent pruning. Another gene expressed by MMc, the *Il-6r*, has similar beneficial functions, as repopulation of the brain with microglia is dependent on IL-6r pathways (Willis et al., 2020). Additionally, only a very low number of MMc expressed inflammatory genes, which may skew the microenvironment in the fetal brain towards homeostatic balance. Indeed, when MMc are low, we observed an upregulation of inflammatory genes in fetal microglia, e.g., *Tnf- α* , and *Ifn- β* , along with the downregulation of *Rab-7b*, which suppresses inflammation. In our present work, we did not provide causal proof to confirm each of these possible pathways underlying the cross-talk between MMc and fetal microglia, as functional evaluation of each pathway is likely 'a paper in itself'. We see the strength of our findings on a larger scale, as we identified a broad spectrum of possible pathways for interaction between MMc and fetal microglia and possible also the entire microenvironment in the developing brain. In order to emphasize on this more clearly, we now amended the discussion as follows:

'Since our results strongly support an MMc-dependent suppression of fetal microglia activation and related synaptic pruning, they raise the question of how MMc may interact with fetal microglia in the developing brain. The outcome of our scRNA-Seq analyses provides pivotal hints towards understanding how the interaction between MMc and fetal microglia is operational. We identified an upregulation of sensome genes in MMc, including *Cd47*, *Selplg*, *Cd37* and *Il-6ra*, along with a down-regulation of inflammatory genes. In fact, it has been shown that the microglia sensome conveys neuroprotection and is involved in host defense²⁷. More specifically, CD47 protects synapses from excess microglia-mediated pruning during development³³. This provide an explanation for the observation we made in MMc^{low} offspring, where the reduction of MMc and hence, *Cd47*, was linked to an increased microglia-dependent pruning. Another gene expressed by MMc, the *Il-6r*, has similar beneficial functions, as repopulation of the brain with microglia is dependent on IL-6r pathways⁵⁴. Additionally, only a very low number of MMc expressed inflammatory genes, which may skew the microenvironment in the fetal brain towards homeostatic balance. Indeed, when MMc are low, we observed an upregulation of inflammatory genes in fetal microglia, e.g., *Tnf- α* , and *Ifn- β* , along with the downregulation of *Rab-7b*, which suppresses inflammation. Taken together, the data suggest a broad spectrum of possible pathways for interaction between MMc and fetal microglia, and, very likely, the entire micro-environment in the developing brain. Causal proof to confirm these pathways should develop from future studies for which our present findings provide a solid rationale.'

#5. The authors defined that 5 clusters of MMc were microglia based on scRNA-seq. Does it mean that pregnant female mice shaded microglia in blood circulation? Or are these cells microglia-like cells? Also, their expression profile suggested that microglia-specific marker such as Tmem119 expression is low? How about Sall1 expression? Are these cells possibly border-associated macrophages (BAMs) or perivascular macrophages? If so, how about the expression of Pf4 and Lyve1 that are markers for BAMs?

Authors' response: We divided our response to this point into two parts.

Part 1: The suggestion made by the reviewer that pregnant females shaded microglia in blood circulation is certainly intriguing. Our approach when assessing MMc in the fetal brain does not provide insights on the origin or history of differentiation of the MMc prior to entering the fetal brain. However, in a previous study, we focused on the role of MMc in fetal bone marrow,

where we observed that a high frequency of MMc were T cells. Thus, it may be more likely that maternal progenitor cells enter the fetal circulation and may then undergo further differentiation once entered or recruited into distinct fetal organs. Although speculative, we have now included this concept as follows in the revised discussion as follows:

‘Clearly, our assessment of MMc in the fetal brain does not provide insights on the origin or differentiation fate of MMc prior to entering the fetal brain. We here observed that a large number of MMc in the fetal brain are microglia, but also T and B cells were present. In a previous study, we focused on the role of MMc in fetal bone marrow and could identify that the high frequency of MMc are T cells, whilst the frequency of microglia-like cells, e.g., macrophages, was low¹⁷.’

Part 2: Here, the reviewer enquires about the low expression of microglia-specific marker, such as Tmem119. As explained in our response to comment #1 from reviewer #2, we apologize for the wrong insertion of genes in the graph shown in Fig. 1f. After re-analyses performed by our systems biologists, we now confirm an elevated expression of Tmem119 in MMc microglia. Moreover, the reviewer comments on the Sall1 expression, possible border-associated macrophages (BAM), and perivascular macrophages among the pool of MMc. In order to address this aspect, we performed additional analyses of the MMc. As now shown in Supplementary Fig. 2b, all microglia clusters showed an elevated expression of Sall1, which supports the idea of MMc expressing microglia check-point genes. Further, by these analyses we excluded that MMc may be border-associated macrophages (BAMs) or perivascular macrophages, since their expression of Pf4 and Lyve1 is low.

In order to provide these aspects, we amended the discussion as follows:

‘We excluded that the MMc microglia were border-associated or perivascular macrophages due to their low expression of platelet factor 4 (*Pf4*) and lymphatic vessel endothelial hyaluronan receptor 1 (*Lyve1*) (**Extended Data Fig. 2b, Supplementary Table 2**).’

Minor points:

#1. Although lightsheet microscope imaging did not look like it, the Extended Figure 1f imaging looks like cells were in the tubular structure. Did the authors co-stain with a marker for the vasculature (e.g., CD31) and ensure that MMc were outside the vasculature?

Author’s response: We thank the reviewer for the comment on **Extended Figure 1f**. As suggested by the reviewer, one might recognize that the structure appearing in the center may be vasculature. In our imaging experiments, we transcardially perfused the offspring’s brains in order to exclude any false-positive signals arising from cells within the cerebral vasculature lumen. In order to only target the td-Tomato-expressing MMc, we did not stain for any other marker. The presence of MMc in the vasculature suggests that the MMc in this image was captured when crossing the blood brain barrier. We amended the figure legend as follows:

‘Noteworthy, tdTomato⁺ MMc can be found in the cerebral vasculature lumen.’

#2. It looks like MMc locate as clusters (by 2D and 3D images). Is there any discussion about it?

Author’s response: As suggested, we mentioned the clustering of MMc:

‘According to the present data, MMc seemed to cluster in the PFC. This unique profile of homeostatic and sensory genes may account for the observed neuronal refinement in specific brain areas.’

#3. References are needed for Rab-7b is a negative regulator for inflammation. So far, the evidence is that Rab-7b is important for the degradation of TLR4.

Author’s response: We provide the requested reference (Yao et al., 2009).

#4. Can the authors show the improvement of excess synaptic engulfment of microglia in vivo, such as sparse labeling of synapses? It is well known that microglia quickly changed the phenotype and gene expression.

Author's response: Reviewer 3 queries about the excessive synaptic engulfment of microglia in vivo mirrored by sparse labeling of synapses. However, an in vivo approach would require the UV ablation of selective neurons in order to visualize microglia activity using live imaging. However, UV ablation would influence the phenotype and gene expression of microglia. In order to provide some of the information requested by the reviewer on the sparse labeling of synapses, we performed additional experiments that included histological examinations of brain sections. Here, we now show significantly less Vglut-1 and Vglut-2 puncta in the PFC and HP of MMc^{low} offspring when compared to MMc^{pos} pups at P8.

We amended the manuscript as follows:

In Results:

'In order to provide evidence for this notion, we investigated microglia engulfment of synaptic terminals in the pre-limbic subdivision (PL) of PFC and HP, since these areas are the core of neuronal networks accounting for complex cognitive abilities, such as memory, learning, and flexibility³⁴. The engulfment of terminals from short-range projections stained by Vglut1⁺ and from long-range projections stained by Vglut2⁺ augmented in MMc^{low} offspring *ex vivo*, yet the increase reached significance level only for Vglut1⁺ (Fig. 2i-k, Extended Data Fig. 7a-n).'

In Methods:

'Microglia numbers and labeled synapses (Vglut1, Vglut2) were determined using the particle analyzer plugin for the ImageJ software. The threshold for all signals was set to acquire optimal representation and kept constant during image analyses. Afterwards, the number was normalized to mm².'

#5. A more detailed method is required for single-cell RNA-seq. For example, how many replicates did the authors use, and any batch effects were observed?

Author's response: We added the requested information:

In Methods/Cell sorting:

'For scRNA-seq, 16 fetal brains from 4 litters were pooled in order to collect 1x10⁴ MMc, which simultaneously minimized potential batch effects and integrates cells from 16 biological replicates.'

References (exclusively cited in the point-by-point reply)

- Aydin, M.S., Yigit, E.N., Vatandaslar, E., Erdogan, E., and Ozturk, G. (2018). Transfer and Integration of Breast Milk Stem Cells to the Brain of Suckling Pups. *Sci Rep* 8, 14289.
- Bähner, F., Demanuele, C., Schweiger, J., Gerchen, M.F., Zamoscik, V., Ueltzhöffer, K., Hahn, T., Meyer, P., Flor, H., Durstewitz, D., *et al.* (2015). Hippocampal–Dorsolateral Prefrontal Coupling as a Species-Conserved Cognitive Mechanism: A Human Translational Imaging Study. *Neuropsychopharmacol* 40, 1674-1681.
- Ben-Zvi, A., Lacoste, B., Kur, E., Andreone, B.J., Mayshar, Y., Yan, H., and Gu, C. (2014). Mfsd2a is critical for the formation and function of the blood–brain barrier. *Nature* 509, 507-511.
- Billstedt, E., Gillberg, C., and Gillberg, C. (2005). Autism after adolescence: population-based 13-to 22-year follow-up study of 120 individuals with autism diagnosed in childhood. *Journal of autism and developmental disorders* 35, 351-360.
- Bitzenhofer, S.H., Pöplau, J.A., Chini, M., Marquardt, A., and Hanganu-Opatz, I.L. (2021). A transient developmental increase in prefrontal activity alters network maturation and causes cognitive dysfunction in adult mice. *Neuron* 109, 1350-1364.e1356.
- Chini, M., Popplau, J.A., Lindemann, C., Carol-Perdiguer, L., Hnida, M., Oberlander, V., Xu, X., Ahlbeck, J., Bitzenhofer, S.H., Mulert, C., and Hanganu-Opatz, I.L. (2020). Resolving and Rescuing Developmental Miswiring in a Mouse Model of Cognitive Impairment. *Neuron* 105, 60-74 e67.
- Depino, A.M., and Gross, C. (2007). Simultaneous assessment of autonomic function and anxiety-related behavior in BALB/c and C57BL/6 mice. *Behav Brain Res* 177, 254-260.
- Eaves, L.C., and Ho, H.H. (2008). Young adult outcome of autism spectrum disorders. *Journal of autism and developmental disorders* 38, 739-747.
- Esposito, G., and Venuti, P. (2010). Understanding early communication signals in autism: a study of the perception of infants' cry. *Journal of Intellectual Disability Research* 54, 216-223.
- Farley, M.A., McMahon, W.M., Fombonne, E., Jenson, W.R., Miller, J., Gardner, M., Block, H., Pingree, C.B., Ritvo, E.R., and Ritvo, R.A. (2009). Twenty-year outcome for individuals with autism and average or near-average cognitive abilities. *Autism Research* 2, 109-118.
- Faust, T.E., Gunner, G., and Schafer, D.P. (2021). Mechanisms governing activity-dependent synaptic pruning in the developing mammalian CNS. *Nat Rev Neurosci* 22, 657-673.
- Filipello, F., Morini, R., Corradini, I., Zerbi, V., Canzi, A., Michalski, B., Erreni, M., Markicevic, M., Starvaggi-Cucuzza, C., Otero, K., *et al.* (2018). The Microglial Innate Immune Receptor TREM2 Is Required for Synapse Elimination and Normal Brain Connectivity. *Immunity* 48, 979-991 e978.
- Garcia, Y., and Esquivel, N. (2018). Comparison of the response of male BALB/c and C57BL/6 mice in behavioral tasks to evaluate cognitive function. *Behavioral Sciences* 8, 14.
- Hammond, T.R., Dufort, C., Dissing-Olesen, L., Giera, S., Young, A., Wysoker, A., Walker, A.J., Gergits, F., Segel, M., Nemesh, J., *et al.* (2019). Single-Cell RNA Sequencing of Microglia throughout the Mouse Lifespan and in the Injured Brain Reveals Complex Cell-State Changes. *Immunity* 50, 253-271 e256.
- Hanamsagar, R., and Bilbo, S.D. (2017). Environment matters: microglia function and dysfunction in a changing world. *Current opinion in neurobiology* 47, 146-155.
- Helles, A., Gillberg, C.I., Gillberg, C., and Billstedt, E. (2015). Asperger syndrome in males over two decades: stability and predictors of diagnosis. *Journal of Child Psychology and Psychiatry* 56, 711-718.
- Herweg, N.A., Apitz, T., Leicht, G., Mulert, C., Fuentemilla, L., and Bunzeck, N. (2016). Theta-Alpha Oscillations Bind the Hippocampus, Prefrontal Cortex, and Striatum during Recollection: Evidence from Simultaneous EEG–fMRI. *The Journal of Neuroscience* 36, 3579-3587.
- Hickman, S.E., Kingery, N.D., Ohsumi, T.K., Borowsky, M.L., Wang, L.C., Means, T.K., and El Khoury, J. (2013). The microglial sensome revealed by direct RNA sequencing. *Nat Neurosci* 16, 1896-1905.
- Kim, J.-S., Kolesnikov, M., Peled-Hajaj, S., Scheyltjens, I., Xia, Y., Trzebanski, S., Haimon, Z., Shemer, A., Lubart, A., and Van Hove, H. (2021). A binary Cre transgenic approach dissects microglia and CNS border-associated macrophages. *Immunity* 54, 176-190. e177.

Kruger, H.S., Brockmann, M.D., Salamon, J., Ittrich, H., and Hanganu-Opatz, I.L. (2012). Neonatal hippocampal lesion alters the functional maturation of the prefrontal cortex and the early cognitive development in pre-juvenile rats. *Neurobiol Learn Mem* 97, 470-481.

Lehrman, E.K., Wilton, D.K., Litvina, E.Y., Welsh, C.A., Chang, S.T., Frouin, A., Walker, A.J., Heller, M.D., Umemori, H., Chen, C., and Stevens, B. (2018). CD47 Protects Synapses from Excess Microglia-Mediated Pruning during Development. *Neuron* 100, 120-134 e126.

Lord, C., Bishop, S., and Anderson, D. (2015). Developmental trajectories as autism phenotypes. In *American Journal of Medical Genetics Part C: Seminars in Medical Genetics* (Wiley Online Library), pp. 198-208.

Magiati, I., Tay, X.W., and Howlin, P. (2014). Cognitive, language, social and behavioural outcomes in adults with autism spectrum disorders: A systematic review of longitudinal follow-up studies in adulthood. *Clinical psychology review* 34, 73-86.

Marsh, S.E., Walker, A.J., Kamath, T., Dissing-Olesen, L., Hammond, T.R., de Soysa, T.Y., Young, A.M., Murphy, S., Abdulraouf, A., and Nadaf, N. (2022). Dissection of artifactual and confounding glial signatures by single-cell sequencing of mouse and human brain. *Nat Neurosci* 25, 306-316.

Masuda, T., Sankowski, R., Staszewski, O., Böttcher, C., Amann, L., Scheiwe, C., Nessler, S., Kunz, P., van Loo, G., and Coenen, V.A. (2019). Spatial and temporal heterogeneity of mouse and human microglia at single-cell resolution. *Nature* 566, 388-392.

Matcovitch-Natan, O., Winter, D.R., Giladi, A., Vargas Aguilar, S., Spinrad, A., Sarrazin, S., Ben-Yehuda, H., David, E., Zelada Gonzalez, F., Perrin, P., *et al.* (2016). Microglia development follows a stepwise program to regulate brain homeostasis. *Science* 353, aad8670.

Milad, M.R., Wright, C.I., Orr, S.P., Pitman, R.K., Quirk, G.J., and Rauch, S.L. (2007). Recall of fear extinction in humans activates the ventromedial prefrontal cortex and hippocampus in concert. *Biological psychiatry* 62, 446-454.

Neniskyte, U., and Gross, C.T. (2017). Errant gardeners: glial-cell-dependent synaptic pruning and neurodevelopmental disorders. *Nature Reviews Neuroscience* 18, 658-670.

O'Neill, P.-K., Gordon, J.A., and Sigurdsson, T. (2013). Theta oscillations in the medial prefrontal cortex are modulated by spatial working memory and synchronize with the hippocampus through its ventral subregion. *Journal of Neuroscience* 33, 14211-14224.

Petitto, J.M., McNamara, R.K., Gendreau, P.L., Huang, Z., and Jackson, A.J. (1999). Impaired learning and memory and altered hippocampal neurodevelopment resulting from interleukin-2 gene deletion. *J Neurosci Res* 56, 441-446.

Seltzer, M.M., Shattuck, P., Abbeduto, L., and Greenberg, J.S. (2004). Trajectory of development in adolescents and adults with autism. *Mental retardation and developmental disabilities research reviews* 10, 234-247.

Spellman, T., Rigotti, M., Ahmari, S.E., Fusi, S., Gogos, J.A., and Gordon, J.A. (2015). Hippocampal-prefrontal input supports spatial encoding in working memory. *Nature* 522, 309-314.

Willis, E.F., MacDonald, K.P.A., Nguyen, Q.H., Garrido, A.L., Gillespie, E.R., Harley, S.B.R., Bartlett, P.F., Schroder, W.A., Yates, A.G., Anthony, D.C., *et al.* (2020). Repopulating Microglia Promote Brain Repair in an IL-6-Dependent Manner. *Cell* 180, 833-846 e816.

Xu, X., Song, L., Kringel, R., and Hanganu-Opatz, I.L. (2021). Developmental decrease of entorhinal-hippocampal communication in immune-challenged DISC1 knockdown mice. *Nature Communications* 12, 6810.

Yao, M., Liu, X., Li, D., Chen, T., Cai, Z., and Cao, X. (2009). Late endosome/lysosome-localized Rab7b suppresses TLR9-initiated proinflammatory cytokine and type I IFN production in macrophages. *The Journal of Immunology* 183, 1751-1758.

REVIEWERS' COMMENTS

Reviewer #1 (Remarks to the Author):

The authors have adequately responded to this reviewer's concerns. The revised manuscript is now suitable for publication.

Reviewer #2 (Remarks to the Author):

The authors have addressed the majority of my comments. However, there is still some issue with the interpretation of the behavioral data in what is now Fig 4. The controls are performing at chance, whereas the MMcPOS mice are performing with a discrimination index that is almost certainly different from chance. Did the authors perform this statistical test (to determine if performance is different from 50% chance in either direction?). If so, the MMc are exhibiting discrimination, which is not a deficit/disturbance. It is a change in behavior for sure, but it is not accurate to say that it is a deficit when they are showing greater discrimination than the controls. This may be somewhat semantic but it is important for the interpretation of the rest of the study and supports the growing literature that microglial pruning is not always maladaptive or may be context-dependent.

Reviewer #3 (Remarks to the Author):

The revised manuscript by Schepanski et al. entitled "Pregnancy-induced maternal microchimerism shapes neurodevelopment and behavior" addressed major concerns, and the manuscript showed significant improvement. Since the manuscript contained a new concept of how maternal cells suppressed fetal microglia and endured normal brain development and behavior, this manuscript should be published in Nature Communications.

Point-by-point reply to reviewers' comments (reproduced verbatim) on manuscript entitled "Pregnancy-induced maternal microchimerism shapes neurodevelopment and behavior", NCOMMS-21-42047-T, April 2022

Reviewer #1

General comment: This manuscript focuses on the role of maternal microchimerism on neurodevelopmental and behavioral changes in the offspring. The authors claim that maternal microchimeric cells (MMc) of different lineage in the fetal brain provide regulatory help in shaping normal neurodevelopment and behavior. The authors have performed some clever experiments and presented their data in an informative manner. This is a complex theme and the data presented need to have a critical analysis and interpretation. Several concerns remain unaddressed.

Authors' response to general comment: We are pleased to learn that Reviewer 1 generally appreciates our study and acknowledges our experimental approach and data presentation. We are grateful for his/her additional constructive comments, which helped us to improve our work, as outlined in the following.

Major comments:

#1. The authors are analyzing tdTomato-positive MMc at E18.5. Did the authors look at E14, E16, or even earlier for their presence? It is important because some of the analyses the authors have presented may relate to neurodevelopmental issues such as autism and schizophrenia later in the offspring. Although the authors show in Fig. 1b that there is no difference in MMc content in males vs. females, it is surprising as even at the placenta level, sexual dimorphism makes an impact on the onset of fetal brain development. MMc significantly decrease as the offspring ages. Is it because some lineage cells of MMc do not renew themselves or are eliminated due to lack of their growth factors in the brain microenvironment. In humans, MMc can be detected after 27 years of birth (Bianchi et al). Did the authors analyze MMc at P60 using the methods described in Fig. 1e and g?

Authors' response: In this comment, the reviewer raised several pivotal aspects. We have subdivided our response in three parts to give each aspect full credit.

Part I (Assessment of MMc at earlier gestational time points). We agree with the reviewer that kinetic analyses of MMc in fetal brain would be interesting, especially at critical neurodevelopmental time points. However, MMc transfer from mother to fetus during gestation is not only a physiological phenomenon, but has also been described as a continuous flow of cells which commences upon completion of placentation at gestational days 9.5/10.5. Therefore, in our present study, we focused on the analysis of MMc presence in the fetal brain at the last time point possible during gestation (E18) in wild-type mice as well as in mice with experimental MMc reduction. This allowed us to characterize MMc at the cellular and molecular level at the end of fetal development, when they could accumulate the longest. Additionally, this enabled us to assess the impact of MMc on fetal microglia phenotype and function prior to birth, hence, prior to the onset of early life environmental stimuli. However, we agree with Reviewer 1 that this approach precludes us from detecting possible fluctuations of MMc at various stages of neurodevelopment. From our perspective, such focus on distinct time points during neurodevelopment is highly relevant when assessing the impact of prenatal adverse events, which may occur at certain days of fetal development, interfere with neurodevelopment and also affect MMc phenotype and function. The integration of adverse events into the experimental setting of our present study would have been beyond its scope, as it was our aim to primarily identify the yet unknown physiological role of MMc. However, we anticipate that our study will now foster the analysis of adverse events on MMc at various gestational time points in future studies. We have revised the discussion as follows to highlight the need for such approach:

'The experimental approach we here chose was primarily geared towards the identification of the yet unknown physiological role of MMc on fetal brain development and later function. However, our focus on E18.5, the time point during fetal development closest to birth, precludes us from detecting possible fluctuations of MMc in fetal brain at various stages of neurodevelopment throughout gestation, once MMc transfer occurs upon completion of placentation. Based on the insights presented here on the relevance of MMc for offspring's neurodevelopment and behavior, a focus on distinct time points during neurodevelopment will likely be highly relevant when assessing the impact of prenatal adverse events. These events can occur at certain days of fetal development, are well known to interfere with neurodevelopment and hence, likely also interfere with MMc phenotype and function^{6,8,37}.'

Part II (No difference in MMc content in males vs. females). We agree with the reviewer that fetal brain development shows a high degree of sexual dimorphism, which is especially obvious in the context of adverse prenatal events. When identifying the number of MMc in brain in pregnancies unchallenged by adverse events, we did not observe significant differences between male and female fetuses in wild type mice. These observations suggest that the vertical transfer of MMc is not affected by the sexual dimorphism on the placental level.

When aiming to assess the consequences of MMc reduction in mice, we used offspring from immunodeficient $Rag2^{-/-}\gamma c^{-/-}$ C57BL/6 females, which had been mated to wild-type Balb/c males. Due to their immunodeficiency, these $Rag2^{-/-}\gamma c^{-/-}$ C57BL/6 females are only capable of transferring a very limited number of immune cells to their fetuses, which we could confirm by the low number of MMc in fetal brain. Therefore, we termed these offspring (which genotype is $Rag2^{+/-}\gamma c^{+/-}$) as MMc^{low}. *Vice versa*, mating wildtype C57BL/6 females to $Rag2^{-/-}\gamma c^{-/-}$ Balb/c males also yields to offspring with a $Rag2^{+/-}\gamma c^{+/-}$ genotype, but the female are fully immunocompetent and hence, transfer physiological levels of MMc to the fetal brain. We termed these offspring MMc⁺ (see also Fig. 2 a,b). However, the γc gene is encoded by the X-chromosome and hence, male offspring born to $Rag2^{-/-}\gamma c^{-/-}$ females are γc deficient, while male offspring born to wild-type females (termed MMc⁺) carry one copy of the γc gene. To control for this hemizyosity, we exclusively included female offspring in the respective experiments, which precludes us from the identification of a possible sexual dimorphisms this this MMc reduction model. To cover this aspect in the revised manuscript, we revised the text as follows:

In Results

'Noteworthy, since the γc gene is encoded by the X-chromosome, male offspring born to $Rag2^{-/-}IL-2\gamma c^{-/-}$ females are γc deficient, while male offspring born to wild-type females carry one copy of the γc gene. To control for this hemizyosity, only female offspring were included in the respective experiments.'

In Discussion

'Fetal brain development shows a high degree of sexual dimorphism, which is especially obvious in the context of adverse prenatal events^{38, 39}. Surprisingly, when identifying the number of MMc in brain in pregnancies unchallenged by adverse events, we did not observe significant differences between male and female fetuses in wild type mice, which suggests that the vertical transfer of MMc is not affected by the sexual dimorphism at the placental level. Due to hemizyosity of the γc gene in male MMc^{low} offspring, we excluded all male offspring from the assessments in the MMc^{pos}/MMc^{low} model. Future investigations will assess the possible sex-specific MMc effects in the MMc reduction model.'

Part III (MMc decrease with increasing offspring age). Similar to reports in other organs, we observed a decrease of MMc with increasing age, whilst MMc were still detectable at low numbers in offspring's brain at P60. In our present study, we did not aim to identify the mechanisms underlying the MMc decline over time. Clearly, the reviewer suggests pivotal pathways that may explain the observed MMc decline with increasing age, e.g., no potential for self-renewal, or elimination due to the absence of growth factors in the microenvironment of the offspring's brain. We have amended the discussion to cover these possibilities, as outlined below. Additionally, the reviewer queries whether we performed scRNA-Seq on MMc isolated on P60. We solely performed this elaborate (and costly) analyses in order to assess MMc on E18 (as shown in Fig. 1e, g). We agree with the reviewer that it would certainly be desirable to survey the gene expression in brain MMc throughout life. However, due to the significant decline in MMc numbers, this approach will likely have to wait until technology allows to isolate and assess cells at extremely low numbers at reasonable costs. Therefore, we prioritized to subject MMc isolated on P60 to flow cytometry-based analysis first, as this allowed us to evaluate overall

numbers along with MMc phenotypes. To address this point in the revised manuscript, we have revised the results and discussion section as follows:

In Discussion:

'Similar to reports in other organs^{17,42}, we observed a decrease of MMc with increasing age, whereby MMc were still detectable at low numbers in offspring's brain at P60. To date, insights on pathways supporting such longevity of MMc – including organ-specific longevity of MMc – as well as the mechanisms leading to the decline of MMc with increasing offspring's age are still mostly unknown. Possible pathways that may explain the observed MMc decline over time may include a limited potential for self-renewal, or the death of MMc due to cellular exhaustion or absence of growth factors in the organ-specific microenvironment, e.g. the offspring's brain. Remarkably, MMc longevity may also be explained by the different MMc phenotypes that can be detected in offspring's fetal and adult organs. E.g., in mice, a large number of MMc in bone marrow are T cells¹⁷, whereas we here show that the largest MMc population in the brain are microglia. These observations suggest either a preferential recruitment of these MMc subsets to the different fetal organs, or a disparate differentiation of progenitor-like MMc, dependent of the tissue microenvironment in which they seeded.'

'Another limitation is the monitoring of MMc by scRNA-seq solely on E18.5. Clearly, it would have been desirable to survey the gene expression in brain MMc throughout life. However, technologies enabling to isolate and assess cells at extremely low numbers at reasonable costs are still missing.'

#2. The authors state that MMc can be found in different organs. Is there any information available about their lifespan in these organs? Or does the brain support a different lifespan of MMc?

Authors' response: Based on the issue raised here, we felt that we needed to amend the introduction by some state-of-the art details and have revised the manuscripts as follows:

In Introduction

'The transfer of MMc commences with maturing placentation, hence, with the onset of the second trimester in humans and around mid-gestation in mice⁹. Remarkably, MMc are not rejected by the fetal immune system¹⁴. In fact, the genetically discordant MMc can even show a long-term persistence in offspring's organs until adulthood¹⁴⁻¹⁶. During fetal development, MMc seed into a number of fetal organs, including primary and secondary immune organs as well as non-immune organs^{17,18}. MMc have also been detected in the offspring's brain^{9,19}, yet their phenotype, location and impact on brain-resident immune and non-immune cells in the fetus and brain function is still unknown.'

Furthermore, the reviewer's query pertaining to the lifespan of MMc in various organs, or possible differential organ (brain)-specific support of MMc lifespans, are in line with Part III of point #1, which is why we had included the details here marked in **bold** in the newly inserted text in the revised discussion. We apologize for the lengthy repeat of these amendments, but feel that this is needed for clarity.

From #1, Part III 'To date, insights on pathways supporting such longevity of MMc – **including organ-specific longevity of MMc** – as well as the mechanisms leading to the decline of MMc with increasing offspring's age are still mostly unknown. Possible pathways that may explain the observed MMc decline over time may include a limited potential for self-renewal, or the death of MMc due to cellular exhaustion or absence of growth factors in the **organ-specific microenvironment**, e.g. the offspring's brain. **Remarkable, MMc longevity may also be explained by the different MMc phenotypes that can be detected in offspring's fetal and adult organs. E.g., in mice, a large number of MMc in bone marrow are T cells¹⁷, whereas we here show that the largest MMc population in the brain are microglia. These observations suggest either a preferential recruitment of these MMc subsets to the different fetal organs, or a disparate differentiation of progenitor-like MMc, dependent of the tissue microenvironment in which they seeded. This may subsequently also affect the lifespan of MMc in the different offspring's organs.**

#3. Pregnancy can be divided into three phases: 1. Implantation (inflammation), gestation (anti-inflammation), and parturition (inflammation). Since E18.5 is very close to the onset of parturition, is inflammation, lack of pregnancy hormones, or infection expected to impact the transport, and the content and/or function of MMc?

Authors' response: The reviewer raises an important point, and we seem to have neglected to address this aspect appropriately in our original submission. We now amended the discussion as follows in order to highlight the importance of the immune trajectory during pregnancy:

'The course of pregnancy can be divided into immunologically distinct stages, including a brief inflammatory surge around the time of blastocysts implantation, followed by the long gestational period of anti-inflammation and immune tolerance to ensure that the fetus is not rejected. Parturition is then initiated by progesterone withdrawal and inflammation⁵⁵. Especially the inflammatory period related to the onset of parturition may affect the transfer of MMc from mother to fetus. Moreover, adverse events occurring during the period of anti-inflammation during pregnancy in mice and humans, e.g., infection or related proxy, as well as trauma, skew maternal cells towards a pro-inflammatory phenotype. This has been shown to enhance the transfer rate of MMc from mother to fetus⁵⁶⁻⁵⁹ and may possibly alter the function of MMc in various fetal organs. The here presorted data on the functional role of MMc during normally progressing pregnancies will enable to address the inflammation-induced alterations of MMc upon adverse prenatal events and the related consequences for offspring's brain and other offspring's organ development and function.'

#4. The authors have focused a lot on brain homeostasis and dismantling of synaptic connections. What does exactly brain homeostasis mean? In Fig. 2 and related Extended Figures, the authors have invoked DNA methylation mechanisms, fetal T and B cell distribution, and transcriptome choreography. It is suggested that genes such as *Rab-7b* were down-regulated in microglia of MMc low offspring. Are there scenarios where the brain is mainly populated with MMc low? Does this result in brain disorders? Is transcriptome profile different in Wt male vs. Wt female?

Author's response: We feel that the description of our findings may have not been comprehensive, otherwise Reviewer 1 would likely not have asked if there are scenarios where the brain is populated with low numbers of MMc or whether we detected sex-specific effects. As shown in Figure 2 and following, we had integrated a model of low MMc in our experimental design. In order to increase comprehension of our data, we have now inserted a new subheading in order to introduce this model of low MMc first, before we describe our findings in fetal brain using this model. This new section reads as follows:

'Mouse model of experimental reduction of MMc in fetal brain

'... , we used a mouse model in which MMc in offspring's brain were experimentally reduced in order to gain insights into the functional role of MMc in the offspring's brain. This reduction of MMc was achieved by reciprocal mating of *Rag2^{-/-}IL-2ryc^{-/-}* female or male mice with wild type (wt) mice. *Rag2^{-/-}IL-2ryc^{-/-}* mice are immunodeficient and lack T, B and – to a lesser extent – innate lymphoid cells³⁰. The offspring of these reciprocal mating combinations all expressed a *Rag2^{+/-}IL-2ryc^{+/-}* genotype (Fig. 2a). Noteworthy, since the *yc* gene is encoded by the X-chromosome, male offspring born to *Rag2^{-/-}IL-2ryc^{-/-}* females are *yc* deficient, while male offspring born to wild-type females carry one copy of the *yc* gene. To control for this hemizyosity, only female offspring were included in the respective experiments.'

'... , we analyzed MMc numbers in offspring's brain at E18.5 and P8. Offspring born to *Rag2^{-/-}IL-2ryc^{-/-}* dams harbored significantly fewer MMc in the brain and were termed 'MMc^{low}', compared to offspring from wt dams (termed 'MMc^{pos}') (Fig. 2b-d).'

In fact, in the MMc low offspring, genes suppressing inflammation, e.g., *Rab-7b*, were down-regulated in microglia, which is mentioned later in the results section as flows:

In Results:

'Here, genes suppressing inflammation, such as ras-related protein (*Rab-7b*), responsible for suppressing tumor necrosis factor (*Tnf*), interleukin-6 (*Il-6*), and interferon β (*Inf- β*) production in macrophages³¹ were down-regulated in microglia from MMc^{low} offspring (Fig. 2h).'

As for possible sex-specific effects, we wish to highlight again that these are not WT offspring, but *Rag2^{+/-} IL-2ryc^{+/-}* offspring. Hence, we could only focus on female offspring (kindly see our reply to #1, part II).

#5. Figures 2 and 3 describe a solid experimental plan to rule out the contribution of immune cells in MMc-mediated brain development. The authors used allogeneic mating protocol involving Wt C57BL/6 female and *Rag2IL-2 ryc* deficient male mice. In another mating, they reversed the mating partners and analyzed the MMc content. They identified two types of neonates – MMcPos and MMc low. What dictates this very distinguished difference? Why some offspring become MMclow? Do MMclow offspring entail poor brain homeostasis? In Fig. 3, the authors employ ultrasonic vocalization as one of the tools to differentiate between MMcPos

and MMclow offspring. Although the authors claim significant differences, the data may not support this claim strongly. Yes, there are clear-cut differences in discrimination profiles; the number MMclow offspring used in the experiments is lower. A careful look at the ultrasonic vocalization suggest that MMc low offspring experience the same vocalization as that described autistic children, particularly male offspring. Do authors have any comment on this?

Author's response: We also subdivided our response to this point into two parts and first address the query related to MMc^{pos}/MMc^{low} animal model.

Part I (Animal model): This comment also indicates that the description of our experimental approach, especially the MMc^{low} vs MMc⁺ model has been too superficial and may leave the reader confused. As outlined above under #4, we have significantly amended the description of our experimental design. Here, we now explain in greater detail why some offspring become MMc^{low}. This facilitates comprehension of the distinguished differences between the Rag2^{+/-} IL-2r γ ^{+/-} offspring, which have either been born from Rag2^{+/+} IL-2r γ ^{+/+} (termed MMc^{low}) or wt mothers (termed MMc⁺). In fact, the comparison between MMc^{low} and MMc⁺ allowed us to identify the hyperactivation and altered function of fetal microglia in MMc^{low} offspring, resulting in enhanced disruption of brain homeostasis, and dismantling of synaptic connections by phagocytosis of presynaptic vesicles. In the context of the present revision, we carefully went over the description of our specific findings in order to ensure rapid comprehension. We refrain from copying the entire section in this reply and kindly refer to the results section of the revised manuscript with the subheading 'MMc maintain fetal microglia homeostasis and suppress excessive presynaptic elimination'.

Part II (Ultrasonic vocalization): Here, Reviewer 1 suggests that our data may not support strong differences. The reviewer correctly noted that the overall number of MMc^{low} pups is lower compared to the number of MMc⁺ pups that were available for the vocalization analysis (8 vs 11). Irrespective of these differences in group size, we observed significant differences with regard to the quality of the vocalizations between groups. As shown in **Fig. 4b**, simple calls were lower in the MMc^{low} offspring, whilst the frequency jumps and the complex calls are higher.

Furthermore, we were very intrigued about the cross-reference made by the reviewer with regard to the similarities between MMc^{low} offspring and autistic children, in which higher fundamental frequencies were noted (Esposito and Venuti, 2010) – similar to the cries we recorded in the MMc^{low} pups. In order to convey the importance of our observations in mice more clearly, along with the intriguing link to autism in humans, we amended the manuscript as follows:

In Results:

'We observed significant differences with regard to the quality of the vocalizations between groups, as the length of simple calls was lower in the MMc^{low} offspring, whereas the frequency jumps and the complex calls lasted longer (**Fig. 4a-b, Extended Data Fig. 10a-c**). These behavioral features might indicate emotional distress and disruption of social communication between mother and MMc^{low} pups. Intriguingly, a similar cry pattern has been observed in autistic children³⁶.'

#6. Adoptive transfer of MMc in MMclow offspring seems to restore microglial engulfment (?). How many MMc needed to be transferred? Is there a kinetic threshold of adoptive transfer? What happens to the offspring at P60 after adoptive transfer?

Author's response: We adoptively transferred 1×10^7 immune cells into the immunodeficient Rag2^{-/-}IL-2r γ ^{-/-} pregnant mice on E12.5. The cell suspension used for the transfer has been obtained by harvesting leukocytes from blood, lymph nodes and spleen of pregnant wild-type mice. The phenotypic assessment of the isolated cells revealed a frequency of approx. 60% B, 30% T, and 10% myeloid cells, as well as low frequencies of NK cells, macrophages, lymphoid, and plasmacytoid dendritic cells, as depicted in **Extended Data Fig. 11c**. Noteworthy, 1×10^7 immune cells is the maximum cell number allowed by our institutional ethical guidelines

to be injected, as higher numbers may cause thromboembolism. Since this number was just sufficient to restore the MMc in fetal brain to the levels seen in MMc⁺ offspring, we refrained from testing fewer cells or evaluating kinetic thresholds.

We did not include the day P60 in our experimental design, as our observations in wild type mice had informed us that MMc in brain have significantly waned at this offspring's age. Therefore, the experimental burden of the intervention for the mice would not have been justified by the outcome, as this would have likely confirmed MMc numbers close to detection limit.

In order to convey the findings in the MMc^{low+AT} offspring born from the *Rag2^{-/-}IL-2 γ c^{-/-}* pregnant mice upon adoptive transfer, we reworded this section in the results section as follows:

'In the MMc^{low+AT} offspring, AT restored the absolute number of MMc in fetal and neonatal brain (**Fig. 5b-d**) and MMc subset populations on E18 and P8 were similarly distributed as observed in MMc^{pos} offspring (**Extended Data Fig. 11d-e**). Moreover, the number of fetal and neonatal microglia was restored in MMc^{low+AT} offspring to the frequencies observed in MMc^{pos} offspring (**Fig. 5e-f**). Similarly, in the MMc^{low+AT} offspring, the enhanced presynaptic terminal elimination detected in MMc^{low} offspring was restored to levels comparable to those seen in MMc^{pos} offspring (**Fig. 5g-h**).'

Minor comments:

#1. On line 118, it should be Fig. 2b-c not Fig. 3b-c. Also, on the same page line 132, it should be "to a lesser extent".

Author's response: We corrected the text accordingly.

#2. Although the authors discuss why fetal microglia increase when MMc are low, it is not entirely clear why MMc become low and affect only microglia. Otherwise, Discussion is well written.

Author's response: In our response to point #4, we have explained why MMc become low in the experimental model we developed and revised the manuscript accordingly. Here, the reviewer additionally queries why only microglia are affected by the decrease of MMc. We respectfully disagree with the statement reducing our findings to 'only' microglia changes. Besides the overall behavioral changes we observed, we also screened for additional changes of fetal brain immune cells and observed reduced numbers of T and B cells in fetal brain (**Extended Data Fig. 6h-f**). Given the importance of microglia, we focused on the consequences of the altered fetal microglia number, phenotype and function, e.g. in their interaction with neurons. Admittedly, we did not follow up on the consequences related to the decrease of T and B cells. In order to address the reviewer's point, we included this now in the discussion as follows:

'Given the importance of microglia for brain wiring, the present study focuses on the identification and functional consequences of the altered fetal microglia number seen in offspring with reduced or restored MMc. The observed decrease of T and B cells in brain of fetal MMc^{low} offspring will be subject of future investigations, especially taking into account the functional role of T cells in autoimmune diseases affecting the brain.'

Reviewer #2

Reviewer #2 (Remarks to the Author):

This is a very interesting manuscript designed to determine the impact of maternal microchimeric (MMc) cells in fetal mouse brain on synaptic development, circuit function, and behavior. The authors report a significant number of the MMc within the fetal brain are microglia with high expression of homeostatic genes, and that these cells primarily localize to the PFC and HP. In the absence (or reduction) of MMc, there are more microglia in the fetal brain and they engulf more presynaptic markers. Moreover, offspring exhibit altered communication and cognitive behaviors and changes in network activity. Finally, adoptive transfer of leukocytes back to immunodeficient dams largely rescues these phenotypes. Overall, there is a large amount of compelling data and the rigor and importance of the work seems high. There are some concerns, which are outlined here.

Author's response to general comment: We are pleased to learn that the reviewer generally appreciates our work and describes our data as compelling.

#1. Fig. 1 – panel f is confusing. It looks like according to the color-coded dots for different cell types that endothelial cells also express high levels of the canonical microglial markers *Cx3cr1*, *P2yr12*, and *Sparc*? Similarly, for panel h, it says the heatmap of gene expression is based on genes that were more than 50% higher in MMc compared to fetal microglia, but the heatmap also shows genes with decreased fold change? Finally, for panel g, these gene categories are subjective and context-dependent. The genes included in each category should be provided rather than assigning them to categories absent a functional readout. For instance, *tmem119* is often considered a “homeostatic” gene in microglia but its expression is not particularly high in the microglia clusters in panel f.

Authors' response: We have to apologize for an oversight that has resulted into the wrong insertion of genes in the graph shown in Fig. 1f, which has been noticed by the reviewer (and unfortunately not by us prior to submission!). The faulty graph indicated the expression of genes (*Cx3cr1*, *P2yr12*, *Sparc*, *Tmem119*), which are not regularly expressed in the non-microglia cells we evaluated (T, B, endothelial, and neuron-like cells). Our systems biologists repeated the analysis pipeline, which resulted in the corrected and now inserted graph 1f, in which *Cx3cr1*, *P2yr12*, *Sparc* and *Tmem119* are no longer upregulated in the non-microglia cells.

Next, the reviewer commented that the gene categories we had used in graph 1g are based on subjective and context-dependent criteria. We agree with the reviewer that such categorization of genes underlies a certain degree of subjectivity. On the other hand, such categorization is a common approach used in many scientific publications in order to facilitate comprehension of the complex findings resulting from scRNA-Seq. However, since it was our primary intention to characterize MMc on a cellular and molecular level - as shown in **Fig. 1e-f, Extended Data 2a-b** - we decided to omit the somewhat redundant graph 1g (and related graph 1h), in which we had introduced the gene categories.

#2. Fig. 2 - Determination of cell number using flow cytometry is not very reliable. There are too many variables regarding cell loss with the isolation and method of gating. Quantitative claims should be shown in the intact brain using IHC or similar.

Authors' response: It appears that two schools of thought may be colliding here, whereas the reviewer seems to favor immunohistological (IHC)/in situ approaches, and our approach is largely based on flow cytometry. Therefore, we respectfully disagree with Reviewer 2's statement that determination of cell number is not very reliable when using flow cytometry as a quantitative approach. To justify and strengthen our approach, we wish to summarize the experimental methods we implemented in order to exclude technical limitations related to flow cytometry, e.g., the loss of cells when isolating and gating brain cell, as mentioned by the reviewer. First, we transcardially perfused the mice prior to sacrificing in order to exclude that

cells isolated from brain are contaminated by cells from the peripheral vasculature supplying the brain. The risk of cell loss when isolating cells from the brain was reduced by our established flow cytometry protocol, which includes i) careful handling of cell pellets when washing the cells, ii) no enzymatic digestion of the brain tissue, as this would have affected cell numbers and surface marker expression. In fact, our protocol allows to isolated brain cell by careful manual homogenizing of the organ through a cell mesh. Taken together, this protocol is well-established and widely used internationally and has resulted in important insights on microglia function and development (Hammond et al., 2019; Kim et al., 2021; Marsh et al., 2022; Matcovitch-Natan et al., 2016). Similarly, the method of gating we (and many others) chose to identify distinct cell subsets in the cell suspensions isolated from the brain is unambiguous due to the well-balanced number of antibodies and conjugated fluorochromes (Filipello et al., 2018; Hammond et al., 2019; Masuda et al., 2019). Our protocol also ensures that the gated cells are indeed the cells of interest by including the so-called FMO control (fluorescence minus one), in which we sequentially omit one fluorochrome-conjugated antibody from the complete antibody panel in order to create a negative control for this specific marker. The high quality of our experimental approach with regard to cell isolation and gating is also reflected by the similar number of parent populations we detected in the respective experimental groups. This quantification step underpins that analysis within a similar parent population yields to sound cellular data in the brain. Second, the quantification of MMC in fetal brain requires the detection of four cell surface markers alone (CD45.2⁺, CD45.1^{neg}, H-2D^{b/lb} +, H-2D^{d/lb} neg), plus additional markers to identify the phenotype of MMC. Here, we additionally included CD11b for the identification of microglia, CD3 for T cells, B220 for B cells and others. In total, 14 markers are needed to unambiguously quantify MMC in fetal brain. Therefore, a histological approach using 14 cells surface markers (antibodies) simultaneously would have been associated with significant technical limitations. Nonetheless, the reviewer is certainly correct in favoring IHC-based approaches, as is enables to localize cells in situ. This was also our goal in the context of MMC detection in the fetal brain, which is why we had included the mouse model in which tdTomato^{+/-} females mated to wild-type males were used and order to generate litter with 50% of tdTomato^{+/-} and 50% of tdTomato^{-/-} offspring. The tdTomato^{-/-} offspring then enable localization of tdTomato-positive cells in the brain, which can only be maternally derived and hence, MMC. These data are shown in Suppl. Fig. 1f and – following the CUBIC clearing of fetal brain - in Movie 1.

We hope that our explanations have convinced Reviewer 2 that the determination of cell number using flow cytometry in our manuscript was sound and has yielded to reliable data.

#3. Why was Vglut1 assessed for pruning by microglia? These label short-range projections (e.g. intra-cortical) at P8, in contrast to Vglut2, which labels long-range projections. It is surprising given the claim that these synaptic changes underlie the circuit deficits in the mice, as the relevant long-range projection synapses were not assessed. It is therefore less convincing that these synaptic changes are linked to the network activity changes reported.

Authors' response: We agree with the reviewer that the assessment of microglia pruning solely based on Vglut1 expression had its limitation, as the Vglut1 staining only labels short-range projection. In the context of the revision, we performed additional experiments and now included data showing the Vglut2-based long-range projections. Here, similar to Vglut1, we also observed a higher Vglut2 engulfment by microglia from MMC^{low} compared to MMC^{pos} offspring. However, opposed to the observations with regard to Vglut1, this increase did not reach levels of significance for Vglut2 ($P = 0.07$).

We amended the results section of the revised manuscript as follows:

'In order to provide evidence for this notion, we investigated microglia engulfment of synaptic terminals in the pre-limbic subdivision (PL) of PFC and HP, since these areas are the core of neuronal networks accounting for complex cognitive abilities, such as memory, learning, and flexibility³⁴. The engulfment of terminals from short-range projections stained by Vglut1⁺ and from long-range projections stained by Vglut2⁺ augmented in MMC^{low} offspring *ex vivo*, yet the increase reached significance level only for Vglut1⁺ (Fig. 2i-k, Extended Data Fig. 7a-n).'

In Methods/Imaging and image analysis:

'For Iba-1, Vglut1, and Vglut2 positive cell expression, microscopic stacks were acquired as 1024x1024 pixels images with 750nm Z-steps capturing 8 microglial cells within the mPFC and CA1 region of the HP, using a 63X objective.'

In Methods/Immunohistochemistry:

'Subsequently, slices were processed by incubating them overnight at 4°C with anti-Iba-1 (1:500, Wako Pure Chemical, Cat. No. 019-19741), anti-Vglut1 (1:1000, Millipore, Cat. No. AB5905), and anti-Vglut2 (1:500, Synaptic Systems, Cat. No. 135404), followed by 1 h incubation with goat-anti-guinea pig (1:500, AF488, Invitrogen, Cat. No. A-11073), donkey-anti-rabbit (1:500, AF568, Invitrogen, Cat. No. A-10042) secondary antibodies and Hoechst33258 (1:5000, Sigma, Cat. No. 94403).'

#4. For fig 3d and f, it actually appears that the controls (MMc^{pos}) mice show no discrimination as they are roughly at chance, whereas the MMc^{low} do, and avoid the novel object. This is not a cognitive deficit but perhaps novelty avoidance. In any case, they show better discrimination if I am interpreting this correctly. More concerning is the fact that the control group does not show normal discrimination so the data are difficult to interpret.

Authors' response: We agree with the reviewer that the behavioral performance of the control offspring (MMc^{pos}) is roughly at chance, whereas the behavioral performance of the MMc^{low} offspring is well below the threshold. We have made similar observations in previous studies, where the control animals also did not all perform above chance (Chini et al., 2020), whilst overall better compared to the data presented here. The poor behavioral performance observed in our present study may be explained by the heterozygous genetic background of the MMc^{pos} and MMc^{low} offspring. These offspring resulted from the allogenic mating combination of two different mouse strains, C57BL/6 females and Balb/c males, hence have a mixed strain background. Published evidence showed that Balb/c and C57BL/6 mice perform differently in behavioral experiments, for instance Balb/c mice are more anxious and less explorative (Depino and Gross, 2007; Garcia and Esquivel, 2018). In addition to the mixed C57BL/6xBalb/c strain background of the MMc^{pos} and MMc^{low} offspring, these offspring are also all heterozygous for the *Rag2/IL-2ryc* genes. It was reported that the genetic manipulation of *IL-2ryc* also results in impaired behavioral competence (Petitto et al., 1999), which may further explain the overall poor performance we observed. We here directly compare offspring that are all of mixed strain background and heterozygous for the *Rag2/IL-2ryc* genes and thus feel, that the significant reduction of behavioral performance in MMc^{low} compared to MMc^{pos} animals is a valid observation. Clearly, the option to extrapolate or compare the overall performance we observe in our unique experimental setting to other studies where e.g., pure strains without transgenic manipulation were used, is very limited. To highlight this aspect in the revised manuscript, we specified the text as follows:

In Discussion:

'Of note, the performance of pre-juvenile MMc^{pos} offspring in recognition memory tasks was often below chance level. This may be explained by the mixed strain background of the MMc^{pos}/MMc^{low} offspring, which resulted from the allogenic mating combination of C57BL/6 females and Balb/c males. In fact, Balb/c mice have been shown to be more anxious and less explorative^{50,51}. Additionally, the MMc^{pos} and MMc^{low} offspring are also heterozygous for the *Rag2/IL-2ryc* genes. The genetic manipulation of *IL-2ryc* may result in an impaired behavioral competence⁵². However, these deficits of individual groups do not bias the robust differences observed between MMc^{low} and MMc^{pos} animals, since they all shared mixed strain background and are heterozygous for the *Rag2/IL-2ryc* genes.'

#5. The authors make no comment on the mechanism by which immune cell deficiency in dams impacts microglial number and/or function. Are there changes in the yolk sac progenitors or only changes once they arrive into the CNS?

Authors' response: In order to address this question, we performed additional experiments in which we isolated the individual yolk sacs from female MMc^{pos} and MMc^{low} offspring and subsequently quantified the leukocytes, erythromyeloid progenitor cells (EMPs) and pre-macrophages at E9.5 by means of flow cytometry. Since the total number of EMP (A) and pre-

macrophages (B) is very low, differences between groups of offspring do not seem to be reliable. However, the reviewer raised an intriguing notion, which we would like to convey to a future readership. We therefore amended the discussion as follows:

'Tissue-resident macrophages like microglia in brain originate from erythromyeloid progenitors (EMP), which develop in the yolk sac, then migrate and seed into the fetal liver and subsequently colonize embryonic organs as EMP-derived macrophages. In the brain, these progenitors complete their differentiation into microglia, which are self-renewing throughout life and are only minimally replenished by circulating macrophages^{43,44}. In our study, the number of progenitor cells did not differ in yolk sac of MMc^{pos} and MMc^{low} offspring at E9.5 (**Extended Data Fig. 12a-b**). However, since the overall number of such progenitor cells was extremely low, the biological significance of these observations may be limited. Future studies should aim at in-depth investigations of the microglia progenitor cells and

their development in presence and absence of MMc infiltration in order to determine their potential interaction already in the yolk sac.'

#6. Why were cognitive behaviors tested at P19-24 in offspring? This is interesting because it is the time in which the hippocampal circuitry is just maturing, and any delay or alteration in this maturation could impact the behavioral phenotype. Do the behavioral phenotypes persist into the later life or are they transient to just this time window?

Authors' response: The reviewer raises an important question. The cognitive behavior has been tested at the time point when the abilities emerge. For this age, we developed the optimally fitting experimental paradigms and acquired a large data set for mice and rats (Chini et al., 2020; Kruger et al., 2012; Xu et al., 2021). The relationship between maturational dynamics and cognitive behavior is of high interest. Studies in rodents showed that specific developmental time windows are of particular relevance for adult behavioral performance. For example, we previously identified the beginning of the second postnatal week as critical for cognitive abilities of adults, since manipulation of neuronal activity at this neonatal age caused memory deficits later in life (Bitzenhofer et al., 2021). On the other hand, environmental triggers or activity manipulation may cause deficits that reverse along postnatal life, become milder in their characteristics or even entirely disappear. For instance, individuals with autism spectrum disorder have been closely studied for their autistic traits in several longitudinal and retrospective studies (Billstedt et al., 2005; Eaves and Ho, 2008; Farley et al., 2009; Helles et al., 2015; Lord et al., 2015; Magiati et al., 2014; Seltzer et al., 2004), showing clear improvements in cognitive and behavioral symptoms over time. The experiments necessary to properly address question #6 are complex and exceed the framework of this already very dense study. They will be the core of a future investigation focusing on the long-term effects. In the present manuscript, we addressed the reviewer's concern by specifying:

In Discussion:

'In the present study, we focused on the emergence of cognitive abilities along neonatal and pre-juvenile development and did not extend the investigation of behavioral phenotype until adult age. The long-term effects of reduced number of MMc might be either milder and (partially) compensated or persistent, leading to life-long deficits. We recently identified critical time windows of cognitive development during which transient manipulation of electrical activity causes permanent reduction of network function and behavioral performance in memory tasks⁵³. Similar processes may occur also in MMc^{low} mice. The number of retained MMc declines with age, although low numbers are still detectable in mature offspring.'

Reviewer #3

Reviewer #3 (Remarks to the Author):

The manuscript entitled "Pregnancy-induced maternal microchimerism shapes neurodevelopment and behavior" by Schepanski et al. described that subsets of maternal cells termed maternal microchimeric cells (MMc) contribute to fetal brain development by repressing fetal microglia cells and preventing excess synaptic pruning. Moreover, the authors observed that MMc ensured the establishment of the prefrontal-hippocampal circuit and normal learning and memory behaviors in offspring. The observation is interesting and may significantly impact the understanding of brain development. However, there are some unignorable concerns about their data and interpretation.

Major points:

#1. It is unclear why the authors ran mouse behavioral assay using only neonatal or (pre-) juvenile mice. The authors reasoned that MMc were not detected beyond P60. But if MMc played significant roles during early brain development, the behavioral alteration should sustain. How are the behavioral phenotypes in fully matured animals?

Authors' response: As specified to query #6 of Reviewer #2, the impact of MMc on cognitive performance along the entire development is a highly important aspect, yet the in-depth experimental investigation exceeds the framework and main aims of the present study. The cognitive behavior has been tested at the time point when the abilities emerge. For this age, we developed the optimally fitting experimental paradigms and acquired a large data set for mice and rats (Chini et al., 2020; Kruger et al., 2012; Xu et al., 2021). The relationship between maturational dynamics and cognitive behavior is of high interest. Studies in rodents showed that specific developmental time windows are of particular relevance for adult behavioral performance. For example, we previously identified the beginning of the second postnatal week as critical for cognitive abilities of adults, since manipulation of neuronal activity at this neonatal age caused memory deficits later in life (Bitzenhofer et al., 2021). On the other hand, environmental triggers or activity manipulation may cause deficits that reverse along postnatal life, become milder in their characteristics or even entirely disappear. Since most structural and functional investigations focused on early and midterm development, we monitored the behavioral correlates during similar time windows. Since neonatal mice, e.g. at P8, are not yet able to perform complex tasks, we focused on basic behavioral experiments such as recording their vocalizations upon separation from the mother to assess the short-term consequences of reduced numbers of MMc during fetal development. Here, we observed significant differences with regard to the quality of the vocalizations between MMc^{pos} and MMc^{low} offspring. The medium-term consequences of reduced numbers of MMc during fetal development were assessed at pre-juvenile age (between P19-24). At this age, more complex behavioral patterns emerge. We observed significantly shorter interaction time and fewer interactions with novel or less-recent objects in MMc^{low} offspring. The number of retained MMc declines with age, although low numbers are still detectable in mature offspring (P60). The experiments necessary to uncover the long-term impact of MMc on adult cognitive behavior (long- and short-term memory, attention, decision-making and working-memory) will be the core of a future study. In the revised manuscript we specified the focus on short- and mid-term effects as follows:

'Next, we tested whether the dysfunction of PFC-HP circuits in neonatal MMc^{low} mice leads to behavioral deficits already at this early developmental stage.'

'Second, we monitored the emergence of cognitive abilities requiring prefrontal-hippocampal communication that can be tested starting from the second-third postnatal week.'

'In the present study, we focused on the emergence of cognitive abilities along neonatal and pre-juvenile development and did not extend the investigation of behavioral phenotype until adult age. The long-term effects of reduced number of MMc might be either milder and (partially) compensated or persistent, leading to life-long deficits. We recently identified critical time windows of cognitive development during which transient manipulation of electrical activity causes permanent reduction of network function and behavioral performance in memory tasks⁵³. Similar processes may occur also in MMc^{low} mice. The number of retained MMc declines with age, although low numbers are still detectable in mature offspring.'

#2. Gu's group showed that the mouse blood-brain barrier became functional at the embryonic day (E) 15.5 (Nature 509, 507-511, 2014), suggesting that MMc migration occurred before. Since fetal microglia cells also influenced the differentiation, proliferation, and migration of neural progenitor cells (NPCs) and newly differentiated neurons, why do the authors think that microglial defect is just synaptic pruning in MMc low mice that occurred much later? Do the authors observe any alteration in NPCs and newly differentiated neurons during earlier fetal brain development?

Authors' response: We share the reviewer's appreciation for the work by Gu's group (Ben-Zvi et al., 2014) on the functional establishment of the blood-brain barrier at the embryonic day 15.5 in mice. However, from our point of view, this functional establishment of the blood-brain barrier does not necessarily exclude continuous MMc migration across the blood-brain barrier. In fact, published evidence reveals that breast milk-derived MMc can be detected in the offspring's brain (Aydin et al., 2018), supporting an MMc migration after blood-brain barrier establishment.

As stated by the reviewer, fetal microglia can influence a wealth of processes in the developing brain, e.g. differentiation, proliferation, and migration of neural progenitor cells as well as the refinement of synaptic connectivity. We here analyzed their influence on refining synaptic connectivity and neuronal branching, because synaptic pruning is the result of a variety of neuronal molecular interactions, such target recognition, as well as phagocytosis, and has been shown to be an essential step in the cascade of impairing neurodevelopment (Faust et al., 2021; Neniskyte and Gross, 2017). In the present study, we did not include assessments of different neuronal types, specific receptor functions, and dendritic arborization earlier during fetal brain development, as it was our primary focus to assess the effect of MMc of fetal microglia towards the end of fetal development, when they could accumulate the longest. In addition, this allowed for the investigation of the influence of MMc on fetal microglia almost entirely devoid of (early) postnatal environmental stimuli, which have been shown to alter microglia function dramatically (Hanamsagar and Bilbo, 2017). To convey our rationale more clearly, we have revised the discussion as follows:

'Since the fetal blood-brain barrier becomes functional during fetal development at E15.5⁴⁰, one may assume that MMc migration into the fetal brain discontinues as of the milestone. Interestingly, published evidence reveals that MMc derived from breast milk can also be detected in the offspring's brain⁴¹, which strongly supports a continuous MMc migration upon blood-brain barrier establishment.'

'The microglia engulfment of pre-synaptic terminals was used as readout of diverse neuronal interactions that control the development of circuits.'

#3. MMc were detected in different brain regions, including the cerebellum. Why only MMc in PFC and HP affect fetal microglia?

Authors' response: As noticed by the reviewer, the initial screening assays led to the detection of MMc in different brain regions, including the cerebellum. In the subsequent functional analyses, we focused on the consequences of MMc in PFC and HP, since these areas are the core of the limbic circuit accounting for cognitive behavior (learning, memory, flexibility). Correspondingly, structural and function alterations of PFC-HP communication have been detected as substrate of cognitive impairment in several neurological and neuropsychiatric disorders (Böhner et al., 2015; Herweg et al., 2016; Milad et al., 2007; O'Neill et al., 2013; Spellman et al., 2015). To mirror these aspects, we modified the text:

In Results:

'In order to provide evidence for this notion, we investigated microglia engulfment of synaptic terminals in the pre-limbic subdivision (PL) of PFC and HP, since these areas are the core of neuronal networks accounting for complex cognitive abilities, such as memory, learning, and flexibility³⁴. The engulfment of terminals from short-range projections stained by Vglut1⁺ and from long-range projections stained by Vglut2⁺ augmented in MMc^{low} offspring *ex vivo*, yet the increase reached significance level only for Vglut1⁺ (Fig. 2i-k, Extended Data Fig. 7a-n).'

In Discussion:

'In the present study, we were able to detect MMc in different brain regions, including the cerebellum. The in-depth functional investigation focused only on MMc in PFC and HP due to the role of these areas for cognitive processing in health and abnormal memory and cognitive flexibility in neurological and neuropsychiatric disorders⁴⁵⁻⁴⁹.'

#4. Is there any mechanistic understanding of how MMc suppress fetal microglia?

Authors' response: The reviewer raises a valid question about the interaction of MMc and fetal microglia in the developing brain, as our results strongly support an MMc-dependent suppression of fetal microglia activation and related synaptic pruning. From our point of view, the outcome of our scRNASeq analyses provides a number of pivotal hints to understand how the interaction between MMc and fetal microglia may be operational. Here, we identified an upregulation of sensome genes in MMc, including *Cd47*, *Selplg*, *Cd37* and *Il-6ra*, along with a downregulation of inflammatory genes (data are shown in Fig. 1). In fact, it has been shown that the microglia sensome conveys neuroprotection and is involved in host defense (Hickman et al., 2013). More specifically, CD47 protects synapses from excess microglia-mediated pruning during development (Lehrman et al., 2018). This provide an explanation for the observation we made in MMc^{low} offspring, where the reduction of MMc and hence, *Cd47*, was linked to an increased microglia-dependent pruning. Another gene expressed by MMc, the *Il-6r*, has similar beneficial functions, as repopulation of the brain with microglia is dependent on IL-6r pathways (Willis et al., 2020). Additionally, only a very low number of MMc expressed inflammatory genes, which may skew the microenvironment in the fetal brain towards homeostatic balance. Indeed, when MMc are low, we observed an upregulation of inflammatory genes in fetal microglia, e.g., *Tnf-α*, and *Ifn-β*, along with the downregulation of *Rab-7b*, which suppresses inflammation. In our present work, we did not provide causal proof to confirm each of these possible pathways underlying the cross-talk between MMc and fetal microglia, as functional evaluation of each pathway is likely 'a paper in itself'. We see the strength of our findings on a larger scale, as we identified a broad spectrum of possible pathways for interaction between MMc and fetal microglia and possible also the entire microenvironment in the developing brain. In order to emphasize on this more clearly, we now amended the discussion as follows:

'Since our results strongly support an MMc-dependent suppression of fetal microglia activation and related synaptic pruning, they raise the question of how MMc may interact with fetal microglia in the developing brain. The outcome of our scRNA-Seq analyses provides pivotal hints towards understanding how the interaction between MMc and fetal microglia is operational. We identified an upregulation of sensome genes in MMc, including *Cd47*, *Selplg*, *Cd37* and *Il-6ra*, along with a down-regulation of inflammatory genes. In fact, it has been shown that the microglia sensome conveys neuroprotection and is involved in host defense²⁷. More specifically, CD47 protects synapses from excess microglia-mediated pruning during development³³. This provide an explanation for the observation we made in MMc^{low} offspring, where the reduction of MMc and hence, *Cd47*, was linked to an increased microglia-dependent pruning. Another gene expressed by MMc, the *Il-6r*, has similar beneficial functions, as repopulation of the brain with microglia is dependent on IL-6r pathways⁵⁴. Additionally, only a very low number of MMc expressed inflammatory genes, which may skew the microenvironment in the fetal brain towards homeostatic balance. Indeed, when MMc are low, we observed an upregulation of inflammatory genes in fetal microglia, e.g., *Tnf-α*, and *Ifn-β*, along with the downregulation of *Rab-7b*, which suppresses inflammation. Taken together, the data suggest a broad spectrum of possible pathways for interaction between MMc and fetal microglia, and, very likely, the entire micro-environment in the developing brain. Causal proof to confirm these pathways should develop from future studies for which our present findings provide a solid rationale.'

#5. The authors defined that 5 clusters of MMc were microglia based on scRNA-seq. Does it mean that pregnant female mice shaded microglia in blood circulation? Or are these cells microglia-like cells? Also, their expression profile suggested that microglia-specific marker such as Tmem119 expression is low? How about Sall1 expression? Are these cells possibly border-associated macrophages (BAMs) or perivascular macrophages? If so, how about the expression of Pf4 and Lyve1 that are markers for BAMs?

Authors' response: We divided our response to this point into two parts.

Part 1: The suggestion made by the reviewer that pregnant females shaded microglia in blood circulation is certainly intriguing. Our approach when assessing MMc in the fetal brain does not provide insights on the origin or history of differentiation of the MMc prior to entering the fetal brain. However, in a previous study, we focused on the role of MMc in fetal bone marrow,

where we observed that a high frequency of MMc were T cells. Thus, it may be more likely that maternal progenitor cells enter the fetal circulation and may then undergo further differentiation once entered or recruited into distinct fetal organs. Although speculative, we have now included this concept as follows in the revised discussion as follows:

'Clearly, our assessment of MMc in the fetal brain does not provide insights on the origin or differentiation fate of MMc prior to entering the fetal brain. We here observed that a large number of MMc in the fetal brain are microglia, but also T and B cells were present. In a previous study, we focused on the role of MMc in fetal bone marrow and could identify that the high frequency of MMc are T cells, whilst the frequency of microglia-like cells, e.g., macrophages, was low¹⁷.'

Part 2: Here, the reviewer enquires about the low expression of microglia-specific marker, such as Tmem119. As explained in our response to comment #1 from reviewer #2, we apologize for the wrong insertion of genes in the graph shown in Fig. 1f. After re-analyses performed by our systems biologists, we now confirm an elevated expression of Tmem119 in MMc microglia. Moreover, the reviewer comments on the Sall1 expression, possible border-associated macrophages (BAM), and perivascular macrophages among the pool of MMc. In order to address this aspect, we performed additional analyses of the MMc. As now shown in Supplementary Fig. 2b, all microglia clusters showed an elevated expression of Sall1, which supports the idea of MMc expressing microglia check-point genes. Further, by these analyses we excluded that MMc may be border-associated macrophages (BAMs) or perivascular macrophages, since their expression of Pf4 and Lyve1 is low.

In order to provide these aspects, we amended the discussion as follows:

'We excluded that the MMc microglia were border-associated or perivascular macrophages due to their low expression of platelet factor 4 (*Pf4*) and lymphatic vessel endothelial hyaluronan receptor 1 (*Lyve1*) (**Extended Data Fig. 2b, Supplementary Table 2**).'

Minor points:

#1. Although lightsheet microscope imaging did not look like it, the Extended Figure 1f imaging looks like cells were in the tubular structure. Did the authors co-stain with a marker for the vasculature (e.g., CD31) and ensure that MMc were outside the vasculature?

Author's response: We thank the reviewer for the comment on **Extended Figure 1f**. As suggested by the reviewer, one might recognize that the structure appearing in the center may be vasculature. In our imaging experiments, we transcidentally perfused the offspring's brains in order to exclude any false-positive signals arising from cells within the cerebral vasculature lumen. In order to only target the td-Tomato-expressing MMc, we did not stain for any other marker. The presence of MMc in the vasculature suggests that the MMc in this image was captured when crossing the blood brain barrier. We amended the figure legend as follows:

'Noteworthy, tdTomato⁺ MMc can be found in the cerebral vasculature lumen.'

#2. It looks like MMc locate as clusters (by 2D and 3D images). Is there any discussion about it?

Author's response: As suggested, we mentioned the clustering of MMc:

'According to the present data, MMc seemed to cluster in the PFC. This unique profile of homeostatic and sensory genes may account for the observed neuronal refinement in specific brain areas.'

#3. References are needed for Rab-7b is a negative regulator for inflammation. So far, the evidence is that Rab-7b is important for the degradation of TLR4.

Author's response: We provide the requested reference (Yao et al., 2009).

#4. Can the authors show the improvement of excess synaptic engulfment of microglia in vivo, such as sparse labeling of synapses? It is well known that microglia quickly changed the phenotype and gene expression.

Author's response: Reviewer 3 queries about the excessive synaptic engulfment of microglia in vivo mirrored by sparse labeling of synapses. However, an in vivo approach would require the UV ablation of selective neurons in order to visualize microglia activity using live imaging. However, UV ablation would influence the phenotype and gene expression of microglia. In order to provide some of the information requested by the reviewer on the sparse labeling of synapses, we performed additional experiments that included histological examinations of brain sections. Here, we now show significantly less Vglut-1 and Vglut-2 puncta in the PFC and HP of MMc^{low} offspring when compared to MMc^{pos} pups at P8.

We amended the manuscript as follows:

In Results:

'In order to provide evidence for this notion, we investigated microglia engulfment of synaptic terminals in the pre-limbic subdivision (PL) of PFC and HP, since these areas are the core of neuronal networks accounting for complex cognitive abilities, such as memory, learning, and flexibility³⁴. The engulfment of terminals from short-range projections stained by Vglut1⁺ and from long-range projections stained by Vglut2⁺ augmented in MMc^{low} offspring *ex vivo*, yet the increase reached significance level only for Vglut1⁺ (Fig. 2i-k, Extended Data Fig. 7a-n).'

In Methods:

'Microglia numbers and labeled synapses (Vglut1, Vglut2) were determined using the particle analyzer plugin for the ImageJ software. The threshold for all signals was set to acquire optimal representation and kept constant during image analyses. Afterwards, the number was normalized to mm².'

#5. A more detailed method is required for single-cell RNA-seq. For example, how many replicates did the authors use, and any batch effects were observed?

Author's response: We added the requested information:

In Methods/Cell sorting:

'For scRNA-seq, 16 fetal brains from 4 litters were pooled in order to collect 1x10⁴ MMc, which simultaneously minimized potential batch effects and integrates cells from 16 biological replicates.'

References (exclusively cited in the point-by-point reply)

- Aydin, M.S., Yigit, E.N., Vatandaslar, E., Erdogan, E., and Ozturk, G. (2018). Transfer and Integration of Breast Milk Stem Cells to the Brain of Suckling Pups. *Sci Rep* 8, 14289.
- Bähner, F., Demanuele, C., Schweiger, J., Gerchen, M.F., Zamoscik, V., Ueltzhöffer, K., Hahn, T., Meyer, P., Flor, H., Durstewitz, D., *et al.* (2015). Hippocampal–Dorsolateral Prefrontal Coupling as a Species-Conserved Cognitive Mechanism: A Human Translational Imaging Study. *Neuropsychopharmacol* 40, 1674-1681.
- Ben-Zvi, A., Lacoste, B., Kur, E., Andreone, B.J., Mayshar, Y., Yan, H., and Gu, C. (2014). Mfsd2a is critical for the formation and function of the blood–brain barrier. *Nature* 509, 507-511.
- Billstedt, E., Gillberg, C., and Gillberg, C. (2005). Autism after adolescence: population-based 13-to 22-year follow-up study of 120 individuals with autism diagnosed in childhood. *Journal of autism and developmental disorders* 35, 351-360.
- Bitzenhofer, S.H., Pöplau, J.A., Chini, M., Marquardt, A., and Hanganu-Opatz, I.L. (2021). A transient developmental increase in prefrontal activity alters network maturation and causes cognitive dysfunction in adult mice. *Neuron* 109, 1350-1364.e1356.
- Chini, M., Popplau, J.A., Lindemann, C., Carol-Perdiguer, L., Hnida, M., Oberlander, V., Xu, X., Ahlbeck, J., Bitzenhofer, S.H., Mulert, C., and Hanganu-Opatz, I.L. (2020). Resolving and Rescuing Developmental Miswiring in a Mouse Model of Cognitive Impairment. *Neuron* 105, 60-74 e67.
- Depino, A.M., and Gross, C. (2007). Simultaneous assessment of autonomic function and anxiety-related behavior in BALB/c and C57BL/6 mice. *Behav Brain Res* 177, 254-260.
- Eaves, L.C., and Ho, H.H. (2008). Young adult outcome of autism spectrum disorders. *Journal of autism and developmental disorders* 38, 739-747.
- Esposito, G., and Venuti, P. (2010). Understanding early communication signals in autism: a study of the perception of infants' cry. *Journal of Intellectual Disability Research* 54, 216-223.
- Farley, M.A., McMahon, W.M., Fombonne, E., Jenson, W.R., Miller, J., Gardner, M., Block, H., Pingree, C.B., Ritvo, E.R., and Ritvo, R.A. (2009). Twenty-year outcome for individuals with autism and average or near-average cognitive abilities. *Autism Research* 2, 109-118.
- Faust, T.E., Gunner, G., and Schafer, D.P. (2021). Mechanisms governing activity-dependent synaptic pruning in the developing mammalian CNS. *Nat Rev Neurosci* 22, 657-673.
- Filipello, F., Morini, R., Corradini, I., Zerbi, V., Canzi, A., Michalski, B., Erreni, M., Markicevic, M., Starvaggi-Cucuzza, C., Otero, K., *et al.* (2018). The Microglial Innate Immune Receptor TREM2 Is Required for Synapse Elimination and Normal Brain Connectivity. *Immunity* 48, 979-991 e978.
- Garcia, Y., and Esquivel, N. (2018). Comparison of the response of male BALB/c and C57BL/6 mice in behavioral tasks to evaluate cognitive function. *Behavioral Sciences* 8, 14.
- Hammond, T.R., Dufort, C., Dissing-Olesen, L., Giera, S., Young, A., Wysoker, A., Walker, A.J., Gergits, F., Segel, M., Nemesh, J., *et al.* (2019). Single-Cell RNA Sequencing of Microglia throughout the Mouse Lifespan and in the Injured Brain Reveals Complex Cell-State Changes. *Immunity* 50, 253-271 e256.
- Hanamsagar, R., and Bilbo, S.D. (2017). Environment matters: microglia function and dysfunction in a changing world. *Current opinion in neurobiology* 47, 146-155.
- Helles, A., Gillberg, C.I., Gillberg, C., and Billstedt, E. (2015). Asperger syndrome in males over two decades: stability and predictors of diagnosis. *Journal of Child Psychology and Psychiatry* 56, 711-718.
- Herweg, N.A., Apitz, T., Leicht, G., Mulert, C., Fuentemilla, L., and Bunzeck, N. (2016). Theta-Alpha Oscillations Bind the Hippocampus, Prefrontal Cortex, and Striatum during Recollection: Evidence from Simultaneous EEG–fMRI. *The Journal of Neuroscience* 36, 3579-3587.
- Hickman, S.E., Kingery, N.D., Ohsumi, T.K., Borowsky, M.L., Wang, L.C., Means, T.K., and El Khoury, J. (2013). The microglial sensome revealed by direct RNA sequencing. *Nat Neurosci* 16, 1896-1905.
- Kim, J.-S., Kolesnikov, M., Peled-Hajaj, S., Scheyltjens, I., Xia, Y., Trzebanski, S., Haimon, Z., Shemer, A., Lubart, A., and Van Hove, H. (2021). A binary Cre transgenic approach dissects microglia and CNS border-associated macrophages. *Immunity* 54, 176-190. e177.

Kruger, H.S., Brockmann, M.D., Salamon, J., Ittrich, H., and Hanganu-Opatz, I.L. (2012). Neonatal hippocampal lesion alters the functional maturation of the prefrontal cortex and the early cognitive development in pre-juvenile rats. *Neurobiol Learn Mem* 97, 470-481.

Lehrman, E.K., Wilton, D.K., Litvina, E.Y., Welsh, C.A., Chang, S.T., Frouin, A., Walker, A.J., Heller, M.D., Umemori, H., Chen, C., and Stevens, B. (2018). CD47 Protects Synapses from Excess Microglia-Mediated Pruning during Development. *Neuron* 100, 120-134 e126.

Lord, C., Bishop, S., and Anderson, D. (2015). Developmental trajectories as autism phenotypes. In *American Journal of Medical Genetics Part C: Seminars in Medical Genetics* (Wiley Online Library), pp. 198-208.

Magiati, I., Tay, X.W., and Howlin, P. (2014). Cognitive, language, social and behavioural outcomes in adults with autism spectrum disorders: A systematic review of longitudinal follow-up studies in adulthood. *Clinical psychology review* 34, 73-86.

Marsh, S.E., Walker, A.J., Kamath, T., Dissing-Olesen, L., Hammond, T.R., de Soysa, T.Y., Young, A.M., Murphy, S., Abdulraouf, A., and Nadaf, N. (2022). Dissection of artifactual and confounding glial signatures by single-cell sequencing of mouse and human brain. *Nat Neurosci* 25, 306-316.

Masuda, T., Sankowski, R., Staszewski, O., Böttcher, C., Amann, L., Scheiwe, C., Nessler, S., Kunz, P., van Loo, G., and Coenen, V.A. (2019). Spatial and temporal heterogeneity of mouse and human microglia at single-cell resolution. *Nature* 566, 388-392.

Matcovitch-Natan, O., Winter, D.R., Giladi, A., Vargas Aguilar, S., Spinrad, A., Sarrazin, S., Ben-Yehuda, H., David, E., Zelada Gonzalez, F., Perrin, P., *et al.* (2016). Microglia development follows a stepwise program to regulate brain homeostasis. *Science* 353, aad8670.

Milad, M.R., Wright, C.I., Orr, S.P., Pitman, R.K., Quirk, G.J., and Rauch, S.L. (2007). Recall of fear extinction in humans activates the ventromedial prefrontal cortex and hippocampus in concert. *Biological psychiatry* 62, 446-454.

Neniskyte, U., and Gross, C.T. (2017). Errant gardeners: glial-cell-dependent synaptic pruning and neurodevelopmental disorders. *Nature Reviews Neuroscience* 18, 658-670.

O'Neill, P.-K., Gordon, J.A., and Sigurdsson, T. (2013). Theta oscillations in the medial prefrontal cortex are modulated by spatial working memory and synchronize with the hippocampus through its ventral subregion. *Journal of Neuroscience* 33, 14211-14224.

Petitto, J.M., McNamara, R.K., Gendreau, P.L., Huang, Z., and Jackson, A.J. (1999). Impaired learning and memory and altered hippocampal neurodevelopment resulting from interleukin-2 gene deletion. *J Neurosci Res* 56, 441-446.

Seltzer, M.M., Shattuck, P., Abbeduto, L., and Greenberg, J.S. (2004). Trajectory of development in adolescents and adults with autism. *Mental retardation and developmental disabilities research reviews* 10, 234-247.

Spellman, T., Rigotti, M., Ahmari, S.E., Fusi, S., Gogos, J.A., and Gordon, J.A. (2015). Hippocampal-prefrontal input supports spatial encoding in working memory. *Nature* 522, 309-314.

Willis, E.F., MacDonald, K.P.A., Nguyen, Q.H., Garrido, A.L., Gillespie, E.R., Harley, S.B.R., Bartlett, P.F., Schroder, W.A., Yates, A.G., Anthony, D.C., *et al.* (2020). Repopulating Microglia Promote Brain Repair in an IL-6-Dependent Manner. *Cell* 180, 833-846 e816.

Xu, X., Song, L., Kringel, R., and Hanganu-Opatz, I.L. (2021). Developmental decrease of entorhinal-hippocampal communication in immune-challenged DISC1 knockdown mice. *Nature Communications* 12, 6810.

Yao, M., Liu, X., Li, D., Chen, T., Cai, Z., and Cao, X. (2009). Late endosome/lysosome-localized Rab7b suppresses TLR9-initiated proinflammatory cytokine and type I IFN production in macrophages. *The Journal of Immunology* 183, 1751-1758.

Point-by-point reply to reviewers' feedback or remaining comments (reproduced verbatim) on revised manuscript entitled "Pregnancy-induced maternal microchimerism shapes neurodevelopment and behavior", NCOMMS-21-42047-A, July 2022

Reviewer #1 (Remarks to the Author)

The authors have adequately responded to this reviewer's concerns. The revised manuscript is now suitable for publication.

Author's response: We thank the reviewer for the positive feedback.

Reviewer #2 (Remarks to the Author)

The authors have addressed the majority of my comments. However, there is still some issue with the interpretation of the behavioral data in what is now Fig 4. The controls are performing at chance, whereas the MMcPOS mice are performing with a discrimination index that is almost certainly different from chance. Did the authors perform this statistical test (to determine if performance is different from 50% chance in either direction?). If so, the MMc are exhibiting discrimination, which is not a deficit/disturbance. It is a change in behavior for sure, but it is not accurate to say that it is a deficit when they are showing greater discrimination than the controls. This may be somewhat semantic but it is important for the interpretation of the rest of the study and supports the growing literature that microglial pruning is not always maladaptive or may be context-dependent.

Author's response: As suggested, we performed the additional statistical testing to decide whether the performance is different from 50% chance in either direction. In line with the result, we revised the interpretation of the behavioral data related to the discrimination ratio and re-phrased the sentences as following:

In Results:

'MMc^{low} offspring showed significantly shorter interaction time and fewer interactions with novel or less-recent objects when compared to MMc^{pos} (Fig. 4c-f), which indicates greater discrimination compared to MMc^{pos} animals.'

Reviewer #3 (Remarks to the Author)

The revised manuscript by Schepanski et al. entitled "Pregnancy-induced maternal microchimerism shapes neurodevelopment and behavior" addressed major concerns, and the manuscript showed significant improvement. Since the manuscript contained a new concept of how maternal cells suppressed fetal microglia and endured normal brain development and behavior, this manuscript should be published in Nature Communications.

Author's response: We thank the reviewer for the feedback and helpful comments.